# Alteration of the size distributions and mixing states of black carbon through transport in the boundary layer in East Asia

Takuma Miyakawa[1,2], Naga Oshima[3], Fumikazu Taketani[1,2], Yuichi Komazaki[1], Ayako Yoshino[4], Akinori Takami[4], Yutaka Kondo[5], and Yugo Kanaya[1,2]

[1]Department of Environmental Geochemical Cycle Research, Japan Agency for Marine-Earth Science and Technology, 3173-25 Showa-machi, Kanazawa-ku, Yokohama, Kanagawa, 236-0001, Japan.

[2]Institute for Arctic Climate Environment Research, Japan Agency for Marine-Earth Science and Technology, 3173-25 Showa-machi, Kanazawa-ku, Yokohama, Kanagawa, 236-0001, Japan.

[3]Meteorological Research Institute, 1-1 Nagamine, Tsukuba, Ibaraki, 305-0052, Japan

[4]Center for Regional Environmental Research, National Institute for Environmental Studies, 16-2 Onogawa, Tsukuba, Ibaraki, 305-8506, Japan

[5]National Institute for Polar Research, 10-3 Midori-cho, Tachikawa, Tokyo, 190-8518, Japan

*Correspondence to*: Takuma Miyakawa (miyakawat@jamstec.go.jp)

**Abstract.** Ground-based measurements of black carbon (BC) were performed near an industrial source region in the early summer of 2014 and at a remote island in Japan in the spring of 2015. Here, we report the temporal variations in the transport, size distributions, and mixing states of the BC-containing particles. These particles were characterized using a continuous soot monitoring system, a single particle soot

photometer, and an aerosol chemical speciation monitor. The effects of aging on the growth of BC-containing particles were examined by comparing the ground-based observations between the near-source and remote island sites. Secondary formation of sulfate and organic aerosols strongly affected the increases in BC coating (i.e., enhancement of cloud condensation nuclei activity) with air mass aging from the source to the outflow regions. The effects of wet removal on BC microphysics were elucidated by classifying the continental outflow air masses depending on the enhancement ratios of BC to CO ($\Delta BC/\Delta CO$) ratios, which was used as an indicator of the transport efficiency of BC. It was found that $\Delta BC/\Delta CO$ ratios were controlled mainly by the wet removal during transport in the planetary boundary layer (PBL) on the timescale of 1-2 days. The meteorological conditions and backward trajectory analyses suggested that air masses strongly affected by wet removal originated mainly from a region in South China (20º-35ºN) in the spring of 2015. Removal of large and thickly-coated BC-containing particles was detected in the air masses that were substantially affected by the wet removal in the PBL, as predicted by Köhler theory. The size and water-solubility of BC-containing particles in the PBL can be altered by the wet removal as well as the condensation of non-BC materials.

**1. Introduction**

Black carbon (BC)-containing particles in atmosphere can significantly affect the radiative budget of the Earth through two effects; direct (light absorption and scattering) and indirect (aerosol-cloud interactions) effects (Bond et al., 2013; references therein). The difficulty in the estimation of these effects in the atmosphere results from both the short lifetime relative to other greenhouse gases and the variable physicochemical properties of BC-containing particles. The BC itself is water-insoluble immediately

after emission, but it subsequently exhibits increased hygroscopicity (McMeeking et al., 2011) and cloud condensation nuclei (CCN) activity (Kuwata et al., 2007) through atmospheric transport and aging. Only small amounts of water-soluble materials on BC particles are needed to cause their activation to form cloud droplets under moderate supersaturation conditions (Kuwata et al., 2007; 2009). It is considered that BC-containing particles are removed from the atmosphere mainly by wet deposition (Seinfeld and Pandis, 2006).

The horizontal and vertical distributions of aerosols can be substantially altered by their atmospheric lifetimes (e.g., Lawrence et al., 2007). Moreover, their studies suggested that the removal processes of BC such as dry deposition, below-cloud (i.e., washout), and in-cloud (i.e., rainout) can greatly change the atmospheric lifetimes. The in-cloud processes include nucleation scavenging and scavenging by the preexisting cloud droplets. Precipitation followed by in-cloud processes leads to the irreversible removal of BC-containing particles. Samset et al. (2014), using multiple global model data sets constrained by aircraft observations, suggested that the atmospheric lifetime of BC largely affects its distribution, especially in the northern hemisphere, and this results in significant variations in global direct radiative forcing values. The removal of BC has been considered as an important issue for the geochemical carbon cycle as well as for climate science. The BC-containing particles deposited onto the ocean surface can affect ocean surface particles, dissolved organic carbon (DOC), and microbial processes, by absorbing DOC, stimulating particle aggregation, and changing the size distribution of suspended particles (Mari et al., 2014).

Previous modeling studies have dealt with BC aging processes (condensational growth and coagulation) in box and regional-scale models, and parameterized timescales for the

conversion of BC-containing particles from water-insoluble to -soluble in global models
(Oshima et al., 2009; Liu et al., 2011; Oshima and Koike, 2013).   However, quantitative
knowledge of the variability of microphysical parameters of BC-containing particles and
the timescale of their aging processes is still limited, and thus more investigation are
needed for near-source and remote regions (Samset et al., 2014).   Moteki et al. (2012)
reported the first observational evidence of the size-dependent activation of BC in air
masses uplifting from the planetary boundary layer (PBL) to the free troposphere (FT) in
East Asia in the spring of 2009, as the part of the Aerosol Radiative Forcing in East Asia
(A-FORCE) aircraft campaigns (Oshima et al., 2012).   A similar altitude dependence of
the BC size distribution and similarity in the BC mixing state were observed in other
aircraft measurements conducted in East Asia in winter (Kondo et al., 2016).   Selective
removal of larger BC-containing particles though cloud processing, which is predicted by
Köhler theory, was qualitatively observed in the atmosphere.   This observational
evidence indicates that the size distributions and mixing states of BC-containing particles
have a large impact on the global- and regional-scale spatial distributions of BC through
their upward transport from the PBL to the FT associated with cloud processes.   Despite
the importance of the size distributions and mixing states of BC-containing particles in
the PBL, the measurements of their microphysical properties are still limited around the
source regions in East Asia.
Kanaya et al. (2016) have conducted long-term measurements of BC for 6 years (2009-
2015) at Fukue Island, and they reported the emission and removal of BC in East Asia
using these data sets.   This study determined that the transport efficiency of BC aerosol
particles through the PBL was substantially reduced by wet removal.   Here we examine
the effects of aging and wet removal during transport on the changes in BC size
distributions and mixing state, as well as concentrations, based on ground-based
measurements conducted at the same site in the spring of 2015 using a single particle soot
photometer (SP2) and an Aerosol Chemical Speciation Monitor (ACSM).    We first
describe the meteorological characteristics of the East Asian region in the spring of 2015.
Then, we discuss the relative importance of the below-cloud (i.e., washout) and in-cloud
scavenging (i.e., rainout) processes for the removal of BC as well as the transport patterns
of the East Asian outflow air masses in spring.    The loss of BC-containing particles for
that period is investigated using a similar approach to that used by Kanaya et al. (2016),
and this is performed in connection with the associated changes in BC microphysics and
their relevance to the transport pathways.

**2.    Experimental and data analysis**
**2.1. Atmospheric observations**
Continuous measurements of $PM_{2.5}$ and BC aerosols have been conducted at a remote
island, Fukue Island, since February 2009 (Kanaya et al., 2013; Ikeda et al., 2014).    The
observation site is located at the Fukue Island Atmospheric Environment Monitoring
Station (32.75°N, 128.68°E, **Fig. 1**).    The site is located in the northwest portion of
Fukue Island, approximately 20 km from the main residential area in the southeast.    The
$PM_{2.5}$ aerosols sampled at the site are mostly transported from areas beyond the island.
The enhanced concentrations of BC aerosols in Fukue Island can be mainly attributed to
long-range transport from the Asian continent, according to a previous study (Shiraiwa et
al., 2008) and an emission inventory work (**Fig. 1**, REAS ver. 2.1, Kurokawa et al., 2013).
We deployed an SP2 (Droplet Measurement Technologies, Inc., USA) for the analysis
of microphysical parameters of refractory BC (rBC, Petzold et al., 2013) from March 26,
2015 to April 14, 2015.    The SP2 was calibrated before starting the ambient
measurements.    The calibration protocol for our SP2 is described in Miyakawa et al.
(2016).    Fullerene soot (FS, stock 40971, lot L20W054, Alfa Aesar, USA) particles were
used as a calibration standard for the SP2.    A differential mobility analyzer (Model 3081,
TSI Inc., USA) was used for preparing the monodisperse FS particles.    The analysis of
the calibration results suggests that the full width of half maxima (FWHM) was typically
30% of the modal incandescence signal intensity ($S_{LII}$) for the diameter range studied.
Note that the FWHM can be regarded as an upper limit to describe the resolving power
of rBC mass per particle using our SP2, because the combination of polydisperse size
distribution of FS particles and the transfer function of the DMA can broaden the
distributions of $S_{LII}$ for the prepared FS particles.    The variations in the laser power were
within ±3% during the observation period, thus indicating that the fluctuations of laser
power did not largely affect the lower limit of the detectable rBC size using the SP2.
Mass equivalent diameter ($D_{core}$) of an rBC particle was derived from the rBC mass per
particle ($m_{pp}$) with an assumed particle density for BC (1800 kg m$^{-3}$, Bond and Bergstrom,
2006).    A large diameter Nafion dryer (MD-700, Perma Pure, Inc., USA) was placed in
front of the SP2 for drying the sample air without significant loss of the aerosol particles
greater than 50 nm.    The dry air for MD-700 was generated by a heatless dryer (HD-
2000, Perma Pure, Inc., USA) and a compressor (2AH-23-M222X, MFG Corp., USA).
The relative humidity of the sample air was less than 20% during the observation period.
The hourly number/mass size distributions and hourly median values of shell ($D_S$) to $D_{core}$
ratios ($D_S/D_{core}$) for the selected $D_{core}$ ranges were calculated.    The retrievals of $D_S$ from
the light scattering signals measured by an avalanche photodiode and a position sensitive
detector (Gao et al., 2007) were performed using a time-resolved scattering cross section
method given by Laborde et al. (2012).    In this study, we quantified the $D_S/D_{core}$ ratios
with a $D_{core}$ range between 0.15 and 0.35 µm. The upper limit of the estimation of
$D_S/D_{core}$ ratios is 4 in this study. Maximum levels of $D_S/D_{core}$ ratios retrieved were ~2.5
at $D_{core}$ of 0.2 µm. We also analyzed the microphysical parameters of rBC particles
measured using the SP2 in the early summer of 2014 at Yokosuka (35.32°N, 139.65°E,
**Fig. 1**), located near industrial sources along Tokyo Bay (Miyakawa et al., 2016). These
data sets were used as a reference for the BC-containing particles in air masses strongly
affected by combustion sources.
Equivalent BC (EBC, Petzold et al., 2013) mass concentrations are continuously
measured at Fukue Island using two instruments; a continuous soot-monitoring system
(COSMOS; model 3130, Kanomax, Japan), and a multi-angle absorption photometer
(MAAP; MAAP5012, Thermo Scientific, Inc., USA). The details of the air sampling
and intercomparisons for EBC measurements at Fukue Island have been described
elsewhere (Kanaya et al., 2013; 2016). In this study, mass concentrations of EBC
measured using the COSMOS were evaluated by comparison with those of SP2-derived
rBC. The intercomparison between SP2 and COSMOS will be briefly discussed below.
**Figure 2** depicts the correlation between COSMOS-EBC and SP2-rBC hourly mass
concentrations. The unmeasured fraction of the rBC mass was corrected by
extrapolation of the lognormal fit for the measured mass size distributions outside the
measurable $D_{core}$ range (0.08-0.5 µm). Note that the unmeasured fraction of rBC mass
was minor (<5%) in this study. The linear regression slope of the correlation between
EBC and rBC was 0.88 (±0.03). Uncertainty with respect to the calibration was
examined in an industrial region and found to be within around 3% (Miyakawa et al.,
2016). The average discrepancy between EBC and rBC was beyond the uncertainty of
the calibration and was comparable to the uncertainty of COSMOS (10%) as evaluated
by Kondo et al. (2009).   While the validity of the calibration standard, FS particles, has
been evaluated only near source regions (Moteki and Kondo, 2011; Miyakawa et al.,
2016), the discrepancy may be partly attributed to the differences in physicochemical
properties between ambient BC in remote air and FS particles.   Onsite calibration of the
SP2 using ambient BC particles prepared by a thermal denuder and particle mass classifier,
such as an aerosol particle mass analyzer (APM), is desirable for better quantification of
the rBC mass based on the laser-induced incandescence technique in remote areas.
Although we need to make further attempts to evaluate SP2 in remote areas, this study
indicated that SP2-rBC mass concentrations agreed well with COSMOS-EBC.
Therefore we simply use "BC", instead of the EBC and rBC defined depending upon the
measurement techniques.   We analyzed the COSMOS data for the BC mass
concentrations, and the SP2 data for the BC microphysics.

The chemical composition of non-refractory submicron aerosols was measured using

an Aerodyne Aerosol Chemical Speciation Monitor (ACSM, Aerodyne, Inc., USA.)
placed in an observatory container at Fukue Island during the observation period.   The
details of the ACSM at Fukue Island have been described in Irei et al. (2014).   The
collection efficiency (CE) of the ACSM was assumed to be 0.5 for this period (Yoshino
et al., 2016).   We considered sulfate ($SO_4^{2-}$) ions as a major non-BC material and one of
the most important secondary aerosols in East Asia (Takami et al., 2007) for the data
interpretation.   The fact that $SO_4^{2-}$ is produced in the cloud phase as well as in the gas
phase is beneficial for interpreting temporal changes in $SO_4^{2-}$ concentration associated
with the wet removal processes.   We also analyzed other non-refractory components
such as nitrate ($NO_3^-$), ammonium ($NH_4^+$), and organic matter (OM).   During the period
April 1 -7, 2015, the critical orifice of the inlet assembly of the ACSM became clogged.
ACSM-derived $SO_4^{2-}$, $NO_3^-$, $NH_4^+$, and OM (ACSM- $SO_4^{2-}$, $-NO_3^-$, $-NH_4^+$, and -OM) for
this period was not used in the analysis.
Two high volume air samplers (HV500F, Sibata Scientific Technology, Ltd., Japan)
were deployed on the rooftop of the observatory container.   The sampling flow rate for
both samplers was 500 liters per minute (lpm).   Air sampling was carried out for 21 h
(from 10:00 AM to 7:00 AM) on a 110-mm pre-combusted (900°C for 3 h) quartz filter
(QR-100, Advantec Toyo Kaisha Ltd., Japan).    Both have a $PM_{2.5}$ impactor for
classifying the particle size.   One impaction plate was coated with vacuum grease
(HIVAC-G, Shin-Etsu Chemical Co., Ltd., Japan) to minimize the impact of coarse mode
particles on the chemical analysis of fine mode particles such as radiocarbon analysis,
and a pre-combusted quartz fiber filter with slits was set on another impaction plate to
collect the coarse particles.   Water soluble ions were analyzed using ion chromatography
(IC, Dionex ICS1000, Thermo Fisher Scientific K.K., Japan).   The results from the
chemical analysis of filter samples are not discussed in this study in detail.   We only
used the mass concentration of $SO_4^{2-}$ (IC-$SO_4^{2-}$) in this study to evaluate the uncertainty
in relation to CE of the ACSM, and to analyze the temporal variations during the period
when the ACSM-$SO_4^{2-}$ data were not available (April 1-7, 2015).
The carbon monoxide (CO) mixing ratio was also continuously measured using a
nondispersive infrared (NDIR) CO monitor (model 48C, Thermo Scientific, Inc., USA).
Details of the CO measurements including the long-term variations in sensitivity and zero
level are discussed elsewhere (Kanaya et al., 2016).

**2.2.  Enhancement ratio of BC and $SO_4^{2-}$ to CO as an indicator of the transport and**
**transformation of aerosol particles**
In order to quantify the extent of the removal of BC, we calculated the hourly
enhancement ratio of BC mass concentrations to CO mixing ratios ($\Delta BC/\Delta CO$) against
the East Asian background air concentrations as follows:

$$\frac{\Delta BC}{\Delta CO} = \frac{[BC] - [BC]_{bg}}{[CO] - [CO]_{bg}}, \tag{1}$$


where [BC] and [CO] are measured hourly concentrations of the BC and CO respectively,
and $[BC]_{bg}$ and $[CO]_{bg}$ are their estimated background concentrations.   Here we assumed
that $[BC]_{bg}$ is zero (Oshima et al., 2012).   The background concentration of CO during
the analysis period (March 11 – April 14, 2015) was calculated by averaging the
concentrations lower than the 5th percentile (120 ppb).   The validity of this value is
discussed in the supporting information (S.I.).
Relative changes in $SO_4^{2-}$ to CO were also analyzed using the linear regression slopes
of their correlation in this study.   We did not calculate their hourly values, because it was
difficult to determine the background concentration of $SO_4^{2-}$.   The use of CO as a tracer
of sulfur compounds in East Asia was validated by Koike et al. (2003).   Although sulfur
dioxide ($SO_2$), which is a major precursor of anthropogenic $SO_4^{2-}$, does not always share
the emission sources with CO, the spatial distributions of $SO_2$ emissions is similar to those
of CO emissions in East Asia (Koike et al., 2003; Kurokawa et al., 2013).   Analyzing
the increase or decrease in the slopes of the $SO_4^{2-}$-CO correlation is beneficial to the
investigation of the formation and removal processes for $SO_4^{2-}$.   Especially, the aqueous-
phase reaction of $SO_4^{2-}$ in clouds is discussed using this parameter.

## 2.3. Meteorological field analysis

We used the 6-hourly meteorological data, with a resolution of $1°$ in terms of the
latitude and longitude, from the National Centers for Environmental Prediction (NCEP)
Final (FNL) operational global analysis; and daily precipitation data, with a resolution of
$1°$ in terms of the latitude and longitude, from the Global Precipitation Climatology
Project (GPCP) data set (Huffman et al., 2001). We analyzed these data sets to
investigate the general features of the meteorological field in East Asia during the
observation period.

## 2.4. Backward trajectory analysis

We calculated backward trajectories from the observation site to elucidate the impact
of the Asian outflow. Three-day backward trajectories from the observation site (the
starting altitude was 0.5 km) were calculated every hour using the National Oceanic and
Atmospheric Administration (NOAA) Hybrid Single-Particle Lagrangian Integrated
Trajectory model (Draxler and Rolph, 2012; Rolph, 2012) with the meteorological data
sets (NCEP's Global Data Assimilation system, GDAS). In this study, the residence
time over specific source regions was used as an indicator of their impacts on the observed
air masses. We defined five domains for assessing the impact over the Asian continent;
Northeast China (NE), Korea (KR), Central North China (CN), Central South China (CS),
and Japan (JP) (**Fig. 1**). The period when air masses passed over the domains NE, KR,
CN, and CS at least for one hour is defined as that of "continental outflow". The impacts
of precipitation on the observed air masses were assessed by a parameter referred to as
the "Accumulated Precipitation along Trajectory" (APT, Oshima et al., 2012). In this
study, we calculated the APT values by integrating the amount of hourly precipitation in
the Lagrangian sense along each 3-day back trajectory of the sampled air masses.    The
hourly variations of APT were merged into the observed gas and aerosol data sets.

**3.    Results and discussion**
**3.1.    The meteorological field in the spring of 2015**

The mean meteorological field during the observation period (March 11–April 14,

2015) is discussed for the purpose of characterizing the general features of the wind flow
and precipitation in this region. The migrating anticyclone and cyclone have passed
alternately over East Asia during this period, which is typically dominant in spring over
East Asia (Asai et al., 1988).    **Figure 3a** shows the mean sea level pressure (SLP) and
mean horizontal winds at the 850 hPa level in East Asia during the observation period.
The mean equivalent potential temperature ($\theta$e) and the meridional moisture transport at
the 850 hPa level during the same period are also shown in **Figure 3b**.    The mid-latitude
region (35-50°N, 120-140°E) was under the influence of a modest monsoonal
northwesterly flow, which advected cold, dry air from the continent to the observation
area.    The subtropical region (20°-30°N, 110°-130°E) was under the influence of a
persistent southwesterly flow, part of which was conversing into the observation area
(30°-35°N), and this flow was confluent with the northwesterlies from the continent.
The low-level southerly flow advected warm, moist air into the observation area to sustain
a large amount of precipitation (**Fig. 4a**).

**Figure 3c** shows the temporal variations in surface pressure and precipitable water at

the observation site.    The surface pressure is well anti-correlated with the precipitable
water.    During the observation period, migratory cyclones and anticyclones occurred
occasionally (3 times each).    The occurrence of migratory cyclones advected moist air,
which could have contributed to the wet removal of BC during transport in the PBL.    In
contrast, the occurrence of anticyclones advected dry air, which could have contributed
to the efficient transport of BC from the source regions.

**Figure 4a** depicts the mean precipitation over East Asia during the observation period.

Mean precipitation showed a latitudinal gradient over eastern China and the Yellow Sea
and East China Sea region (i.e., increasing precipitation from south to north), and these
results suggest that transport pathways can greatly affect the wet removal of aerosols.
The APT was compared with the averaged latitude of each trajectory for 48 h backwardly
from the time of -24 h ($\text{Lat}_{ORIG}$) (**Fig. 4b**), which can be interpreted as an indicator of the
latitudinal origin of the air masses arriving at Fukue Island.    The high APT values
corresponded to the air masses that originated from the southern regions (20º-40ºN).
The data points are colored according to the maximum RH values along each backward
trajectory ($\text{RH}_{max}$).    The lower relative humidity ($\text{RH}_{max}$) were observed in the air masses
with low APT values that originated from northern regions (30º-50ºN).    These air mass
characteristics were consistent with the mean precipitation field (**Fig. 4a**).    Some of the
data points showed high values of $\text{RH}_{max}$ (~100%) when their APT was almost zero.
These data probably correspond to the air masses that experienced cloud processes not
associated with precipitation.    Possible effects of cloud processes without precipitation
on the removal of aerosol particles during transport will be discussed using these data
points in the following section.

**3.2. Removal processes of fine aerosol particles**

In this study, the removal processes including dry deposition and below-cloud

scavenging were considered to be minor.    The dry deposition in this region has already
been evaluated by Kanaya et al. (2016), who found minimal decrease in ΔBC/ΔCO for
air masses not affected by wet removal but with different transport times.   The below-
cloud scavenging is dependent on the precipitation intensity and rain drop size as well as
the particle size range.   The removal rates of submicron accumulation mode particles
through the washout ($\Lambda_{accum}$) was estimated to be ~$1 \times 10^{-3}$ $h^{-1}$ ($0.5$-$2 \times 10^{-3}$ $h^{-1}$) using a
parametrization given by Wang et al. (2014) and the average precipitation intensity along
the trajectories ($0.78 \pm 0.6$ mm $h^{-1}$) as an input to the parameterization.     The values of
$\Lambda_{accum}$ can be underestimated by an order of magnitude by using the parameterization
(Wang et al., 2014), which is however overly pessimistic.   The temporal duration in rain
along trajectories for air masses with the APT greater than 0 mm was 10 ($\pm8$) hours on
average.   The average fraction of submicron aerosols removed was 1% (+2.59%/-0.9%).
Even though we took into account the uncertainties for estimating $\Lambda_{accum}$, it was found
that the below-cloud scavenging did not play a major role in the removal of BC in East
Asian outflow.

**3.3. Temporal variations in aerosols and CO**
Temporal variations in the concentrations of BC (measured using COSMOS and SP2),
$SO_4^{2-}$ (measured using ACSM and IC), $NO_3^-$, OM, and CO are shown in **Figure 5**.
ACSM-$SO_4^{2-}$ generally agreed well with IC-SO4, thus indicating that the assumed CE
(0.5) was valid for the observation period.   As $NO_3^-$ and $SO_4^{2-}$ were almost fully
neutralized by $NH_4^+$, we assumed their chemical forms were ammonium salts.   In
general, BC, $SO_4^{2-}$, and OM were positively correlated with CO at Fukue Island, and these
results illustrate the impact of continental outflow affected by incomplete combustion
sources on aerosol mass concentrations.   The mean chemical composition of fine
aerosols during the observation period was listed in **Table 1**. Ammonium sulfate and
OM were abundant components. **Figure 5** also includes the temporal variations in the
fractional residence time over the selected region defined in section 2.4 (top panel). The
CO concentrations were typically enhanced for the period with the higher contributions
of CN and CS. A previous study suggested that the majority of $SO_4^{2-}$ aerosols were
formed in less than around 1.5 days after the air masses left the Chinese continent (Sahu
et al., 2009). Kanaya et al. (2016) showed that the typical transport time of continental
outflow air masses at Fukue Island was around 1-2 days in spring. The positive
correlation of $SO_4^{2-}$ and CO suggests that the secondary formation of $SO_4^{2-}$ through
transport was significant during the observation period. The structure and composition
of fine aerosols in East Asian outflow were analyzed by using a secondary ion mass
spectrometer in a previous study (Takami et al., 2013). They suggest that $SO_4^{2-}$ and OM
are constituents in the coating of almost all BC-containing particles. Hence we
concluded that ammonium sulfate and OM contributed to the growth of BC-containing
particles. The period with the APT > 3 mm is highlighted by light blue in **Figure 5** to
show the impact of wet removal on the transport of BC and $SO_4^{2-}$ aerosols. The
maximum concentrations of aerosols and CO were observed on the morning of March 22
(Ep.1) under the influence of the anticyclone (corresponding to the trajectories colored
red in **Fig. 4a**) when the APT values were almost zero. In contrast, aerosol
concentrations did not increase with CO in the period from the evening of April 5 to the
morning of April 6 (Ep.2) under the influence of the migratory cyclone (corresponding to
the trajectories colored black in **Fig. 4a**), when the APT was greater than 10 mm.

## 3.4. Correlation of BC, $SO_4^{2-}$, and CO


**Figures 6a** and **6b** show scatter plots of CO with BC and $SO_4^{2-}$, respectively. Positive
correlation of BC and $SO_4^{2-}$ with CO was clearly found in air masses with low APT values.
The linear regression was performed to the data points with the APT higher than 15 mm
for BC-CO and $SO_4^{2-}$-CO. Note that the linear regression slope for BC-CO was
determined by forcing through the background concentrations of BC (0 µg m$^{-3}$) and CO
(120 ppb). The slopes of the fitted lines were 1.4 (±0.06) and 9.8 (±2.7) ng m$^{-3}$ ppb$^{-1}$ for
BC-CO and $SO_4^{2-}$-CO, respectively, were close to the lower envelopes of the correlations.
It is evident from these scatter plots that the relative enhancements of BC and $SO_4^{2-}$ to
CO were mainly affected by the APT. Kanaya et al. (2016) found that the estimated
emission ratios of BC to CO over the East Asian continent varied slightly depending on
the origin of the air masses (this range is overlaid on **Fig. 6a**). In their study, the
$\Delta BC/\Delta CO$ ratios for Central North and South China regions were estimated to be 5.3
(±2.1) and 6.9 (±1.2) ng m$^{-3}$ ppb$^{-1}$, respectively. $\Delta BC/\Delta CO$ observed in the PBL over
the Yellow Sea during the same season was 6.2 ng m$^{-3}$ ppb$^{-1}$ (Kondo et al., 2016). The
data points with $\Delta BC/\Delta CO$ in these ranges show low APT values (less than or ~1 mm).
Wet removal (in-cloud scavenging) was one of the most important controlling factors on
the transport efficiency of BC in this region during the observation period.
The cloud processes of aerosol particles not associated with precipitation can also
reduce the slope of their correlation. However, no decreasing tendency of BC/CO and
$SO_4^{2-}$/CO slopes against $RH_{max}$ when APT was zero was found during the observation
period (data not shown). The $SO_4^{2-}$/CO slopes with the APT values of zero were
analyzed as a function $RH_{max}$ (**Figure 6b**), and these varied from 30.7 (±1.8) to 44.1
(±13.4) ng m$^{-3}$ ppb$^{-1}$ under the conditions without ($RH_{max}$ <50%) and with ($RH_{max}$ >80%)
cloud impacts, respectively.   The difference in the slope between without and with cloud
impacts is small, however significant (based on the analysis of covariance to these data
sets).   The fact that the $SO_4^{2-}$/CO slope increased with $RH_{max}$ when the APT was zero
suggests that aqueous phase formation and subsequent droplet evaporation partly
contributed to the mass concentrations of $SO_4^{2-}$ observed at Fukue Island.   Therefore,
the changes in the $SO_4^{2-}$/CO correlation were controlled largely by the in-cloud
scavenging and weakly by aqueous-phase formation during transport.

**3.5.  Changes in fine aerosol compositions**

Chemical compositions of fine aerosols were investigated in terms of the APT and
$RH_{max}$.   Four cases are selected here, namely (1) APT of zero (no precipitation), (2) APT
of zero with $RH_{max}$ <50% (no precipitation without cloud impacts), (3) APT of zero with
$RH_{max}$ >80% (no precipitation with cloud impacts), and (4) APT >15 mm (heavily
affected by wet removal).   The comparison between cases (3) or (4) and (2) is useful to
elucidate the effect of the cloud processing.   The results are summarized in **Table 1**.
Ammonium sulfate and OM were dominant in all cases.   The relative changes in
chemical compositions of fine aerosol particles were within around 10%.   As all
components of fine aerosols were removed through the in-cloud scavenging (Fig. 10 of
Kanaya et al., 2016), it is expected that the relative abundance does not largely vary with
the in-cloud scavenging.   The relative contributions of fine aerosols in the cases (3) and
(4) increased from the case (2), indicating that cloud processes affected their relative
abundances.   Ammonium sulfate contribution slightly increased with the in-cloud
scavenging (based on the comparison between cases (2) and (3) or (4)), while the relative
contributions of ammonium nitrate, OM, and BC slightly decreased.   The contributions
of OM in the case (2) increased from the average.    The formation of secondary OM can
be significant under dry conditions during transport.    Detailed mass spectral analyses of
OM, secondary formation of OM, and cloud-phase formation of OM in East Asia are
beyond the scope of this study, and they are not discussed in this study.    The former two
issues have been investigated by previous studies (e.g., Irei et al., 2014; Yoshino et al.,

2016).


**3.6. Changes in microphysical parameters of BC-containing particles associated**
**with wet removal**

Number and mass size ($D_{core}$) distributions of BC classified by the values of $\Delta$BC/$\Delta$CO
are shown in **Figures 7a** and **7b**, respectively.    When $\Delta$BC/$\Delta$CO values in continental
outflow air masses were greater than 3 ng m$^{-3}$ ppb$^{-1}$ (within the range of the BC/CO
emission ratios given by Kanaya et al. 2016), these air masses are defined as "outflow
without BC loss".    These air masses originated mainly from CN via KR and NE.    When
$\Delta$BC/$\Delta$CO values of continental outflow air masses are less than 1 ng m$^{-3}$ ppb$^{-1}$, the air
masses were defined as "outflow with BC loss".    Considering the typical emission ratios
of BC to CO (6-7 ng m$^{-3}$ ppb$^{-1}$; Kanaya et al., 2016), transport efficiency for the "outflow
with BC loss" air masses can be estimated to be less than ~17%.    These air masses
originated mainly in CS.    The low and high APT values for "outflow without BC loss"
and "outflow with BC loss" air masses, respectively, (**Table 2**) gave us confidence in the
validity of our classification as discussed in the previous section.    As a reference for
emission sources ("source"), the average size distributions of BC in a Japanese industrial
area (see section 2.1, Miyakawa et al., 2016) are shown in **Figure 7**.    The statistics of
the size distributions are summarized in **Table 2**.    Observed differences in the size
distributions between source and outflow were generally consistent with previous studies
(Schwarz et al., 2010). Air mass aging leads to the growth of BC-containing particles.
Number-size distributions of BC largely varied in the size range less than 0.1 µm (**Fig.**
**7a**). In outflow air masses, such small BC-containing particles would be scavenged by
larger particles in the coagulation process during transport. The below-cloud
scavenging can also affect the BC-containing particles in the smaller size range ($<0.1$ µm)
when the air masses were affected by the precipitation. The peak $D_{core}$ of mass (number)
size distributions of BC became larger, from 0.16 (0.06, which is estimated by the mass
size distribution) µm to 0.18-0.2 (0.09-0.1) µm, between source and outflow. The BC-
containing particles have systematically different size distributions in outflow air masses
with and without BC loss, indicating that the BC loss process also affected the size
distributions. The peak $D_{core}$ of BC number and mass size distributions in outflow air
masses with BC loss was slightly lower than that for air masses without BC loss. The
changes in the peak diameter of the core and total (i.e., core and shell) size distributions
of BC-containing particles as a function of $\Delta BC/\Delta CO$ ratios are shown in **Figure 7c**.
The peak values of $D_{core}$ and $D_S$ (with the $D_{core}$ range of 0.15 - 0.35 µm) were determined
by fitting the log-normal function to the hourly BC mass-$D_{core}$ and BC number-$D_S$
distributions of BC-containing particles, respectively. The reason why we did not
analyze the peak values of $D_{core}$ for BC number size distributions is that they were mostly
smaller than 0.08 µm (outside the measurable range). The observed decreases in the
diameters or BC mass per particle were clear and were beyond the uncertainties of SP2
(see section 2.1). The changes in the peak $D_{core}$ and $D_S$ from the highest to lowest bins
of $\Delta BC/\Delta CO$ ratios were 0.02 µm (2-2.5 fg) and 0.05 µm, respectively, which are
statistically significant ($p < 0.01$).
**Figure 8** depicts the probability density of the $D_S/D_{core}$ ratio for the BC size of 0.2
(±0.02) µm for source and outflow air masses.   The modal values of the $D_S/D_{core}$ ratio
were systematically changed with air mass aging and BC loss (in-cloud scavenging).
The condensation of inorganic and organic vapors on BC-containing particles during
transport can account for the increase in the $D_S/D_{core}$ ratio, as discussed in previous studies
(e.g., Shiraiwa et al., 2008; Subramanian et al. 2010).   As discussed earlier, the results
of this study suggested that $SO_4^{2-}$ and OM substantially contributed to the increase in the
$D_S/D_{core}$ ratio.   In outflow air masses with BC loss, modal values of the $D_S/D_{core}$ ratio
were clearly lower than those in outflow without BC loss.   Furthermore, it is indicated
that the wet removal process also affected the coating thickness distributions for the BC
sizes in the range 0.15-0.35 µm (**Table 2**).   It should be noted that the coating of BC-
containing particles is not always thick in remote regions, and that the $D_S/D_{core}$ ratio
distributions, as well as size distributions, can be affected by the wet removal process
during transport in the PBL.

**3.7. Discussion**
Not only in-cloud scavenging of BC-containing particles but also subsequent
precipitation (i.e., the rainout process) can account for the changes in the microphysical
parameters of BC detected in this study.   Our results show a decrease of both the peak
diameters, $D_S$ and $D_{core}$, of the BC number and mass size distributions, respectively, and
the modal value of the $D_S/D_{core}$ ratios in relation to the rainout.   The observed evidence
implies that there can be the removal of large and water-soluble BC-containing particles
during transport in the PBL.   The Köhler theory suggests that a lower super saturation
is needed for the large and highly water-soluble particles, and this can qualitatively
account for the observed changes in the BC microphysics.   The $D_S/D_{core}$ ratios (with
$D_{core}$ of 0.2 µm) in **Figure 8** were converted to the critical super saturation ($SS_C$) of BC-
containing particles, which are estimated using the observed chemical composition of
non-BC materials at Fukue Island.   Hygroscopicity parameters ("CCN-derived" $\kappa$,
Petters and Kreidenweis, 2007) and material densities used for the estimation are (0.67,
1.73 g cm$^{-3}$), (0.61, 1.77 g cm$^{-3}$), (0, 1.8 g cm$^{-3}$), and (0.1, 1.2 g cm$^{-3}$) for ammonium
nitrate, ammonium sulfate, BC, and OM, respectively.   The estimation includes another
assumption that all components are internally mixed with BC.   As chemical
compositions of non-BC materials did not largely vary during the observation period
(section 3.5), the average value of $\kappa$ was calculated using the averaged chemical
compositions (Table 1) to be 0.35, and was used for the calculation of $SS_C$.   The
estimated $SS_C$ decreases with the increases in the $D_S/D_{core}$ ratios.   The observed changes
in the $D_S/D_{core}$ ratios indicated that the BC-containing particles with lower $SS_C$ were
removed through the wet removal.
Note that the magnitude of the change in the peak $D_{core}$ of the BC size distributions in
the PBL (~0.02 µm (~2-2.5 fg)) shown in **Figure 7c** is smaller than that observed in air
masses uplifted from the PBL to the FT, in association with wet removal (~0.04 µm (~3
fg), Fig 2 of Moteki et al., 2012) at a similar level of transport efficiency (<~20%).
Although the shape of mass size distributions soon after the rainout processes can be
distorted by the droplet activation of larger aerosol particles, the observed mass size
distributions were well fitted by a log-normal function (**Fig. 7b**).   **Figure 8** showed the
existence of BC-containing particles with the $D_S/D_{core}$ ratios higher than 1.2 even in
outflow air masses with BC loss that are expected to readily act as CCN.   Air masses
sampled at the ground level would be affected by turbulent mixing of those near the
clouds around the top of the PBL and those in cloud-free conditions at below-cloud levels.
On the other hand, most air masses sampled by aircraft measurements in the FT would
experience the cloud processes during upward transport from the PBL.   Mixing of air
masses in the PBL suggests that they partially experience the in-cloud scavenging
processes.   The aging (e.g., coagulation) of aerosols particles through the transport (i.e.,
around ~1 day) after the wet removal events may also lead to the further modification of
the shape of the particle size distributions and the mixing state distributions which have
been affected by cloud processes.   This factor is actually expected to be minor because
the particle concentrations are too low to have high coagulation coefficients to accelerate
this effect.   The suppression of changes in the microphysical properties of BC-
containing particles during transport in the PBL can be related to these factors.   More
quantitative assessments of the impacts of these factors on the observed features should
be performed using a model which has a function to resolve the mixing state of aerosol
particles (e.g., Matsui et al., 2013).
The transport pathways of the continental outflow air masses are horizontally and
vertically variable in spring in East Asia because of the frequent alternate
cyclone/anticyclone activities in spring (Asai et al., 1988).   Oshima et al. (2013)
examined the three-dimensional transport pathways of BC over East Asia in spring and
showed that the PBL outflow through which BC originating from China was advected by
the low-level westerlies without uplifting out of the PBL was one of the major pathways
for BC export from continental East Asia to the Pacific, thus supporting the general
features of microphysical properties of BC in continental outflow obtained by this study.
Mori et al. (2014) measured the seasonal variations in BC wet deposition fluxes at another
remote island in Japan (Okinawa, ~500 km south of Fukue Island), and revealed their
maxima in spring, which were consistent with the seasonal variations in the cyclone
frequencies.    It has been suggested that BC-containing particles were efficiently
activated to form cloud droplets in the continental outflow air masses, especially from the
CS region, and can affect the cloud physicochemical properties in spring in East Asia, as
indicated by Koike et al. (2012).    As the results from this study are based on the
observations during a limited length of time, it would be worthwhile to further investigate
the possible connections of the variabilities in BC microphysical properties with
meteorological conditions to provide useful constraints on more accurate evaluations
climatic impacts of BC-containing particles in this region (Matsui, 2016).

**4.  Conclusions**
Ground-based measurements of BC were performed near an industrial source region
and at a remote island in Japan.    We have reported the temporal variations in the
transport and the microphysics of the BC-containing particles, measured using COSMOS,
SP2, and ACSM.    The impacts of air mass aging upon the growth of BC-containing
particles were examined by comparing the ground-based observations between the near-
source and remote island sites.    $\Delta BC/\Delta CO$ was used as an indicator of the transport
efficiency of BC, because it was controlled mainly by rainout during transport in the PBL.
The BC size and coating increased during transport from the near-source to the outflow
regions on the timescale of 1-2 days when the rainout during transport was negligible.
$SO_4^{2-}$ and organic aerosols contributed to the significant increase in the coating materials
of BC (i.e., it enhanced the whole size and water-solubility of BC-containing particles).
Decreases in the peak $D_{core}$ and $D_S$ of mass and number size distributions (~0.02 and 0.05
µm), respectively, and modal $D_S/D_{core}$ ratios (~0.4 for BC of 0.2 µm) of BC-containing
particles were observed in air masses substantially affected by in-cloud scavenging.    The
observed evidences for the removal of large and water-soluble BC-containing particles
was qualitatively consistent with the Köhler theory; however the values were not as large
as those found in air masses uplifted from the PBL to the FT in East Asia associated with
precipitation.  The mixing of below-cloud and in-cloud air masses in the PBL would
result in suppression of the degree of changes in BC microphysical parameters by cloud
processes.  This study indicates (1) that the changes (sign and degree) in BC
microphysics can be affected by how the air masses are transported and (2) that the
observed removal of large and water-soluble BC-containing particles through in-cloud
scavenging in East Asia can be expected to be significant in the PBL as well as in the FT
in East Asia.


**Acknowledgments**
This study was supported by the Environment Research and Technology
Development Fund (S7, S12, and 2-1403) of the Ministry of Environment, Japan, and
the Japan Society for the Promotion of Science (JSPS), KAKENHI Grant numbers
JP26550021, JP26701004, JP26241003, JP16H01772, and JP16H01770, and was
partially carried out in the Arctic Challenge for Sustainability (ArCS) Project.   The
authors would like to thank N. Moteki at the University of Tokyo for assistance with the
SP2 calibrations.   M. Kubo, T. Takamura, and H. Irie (Chiba University) are also
acknowledged for their support at the Fukue-Island Atmospheric Environment
Monitoring Station.   The authors appreciate Gavin McMeeking and an anonymous
reviewer for constructive comments to improve this paper.

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

**Figures**

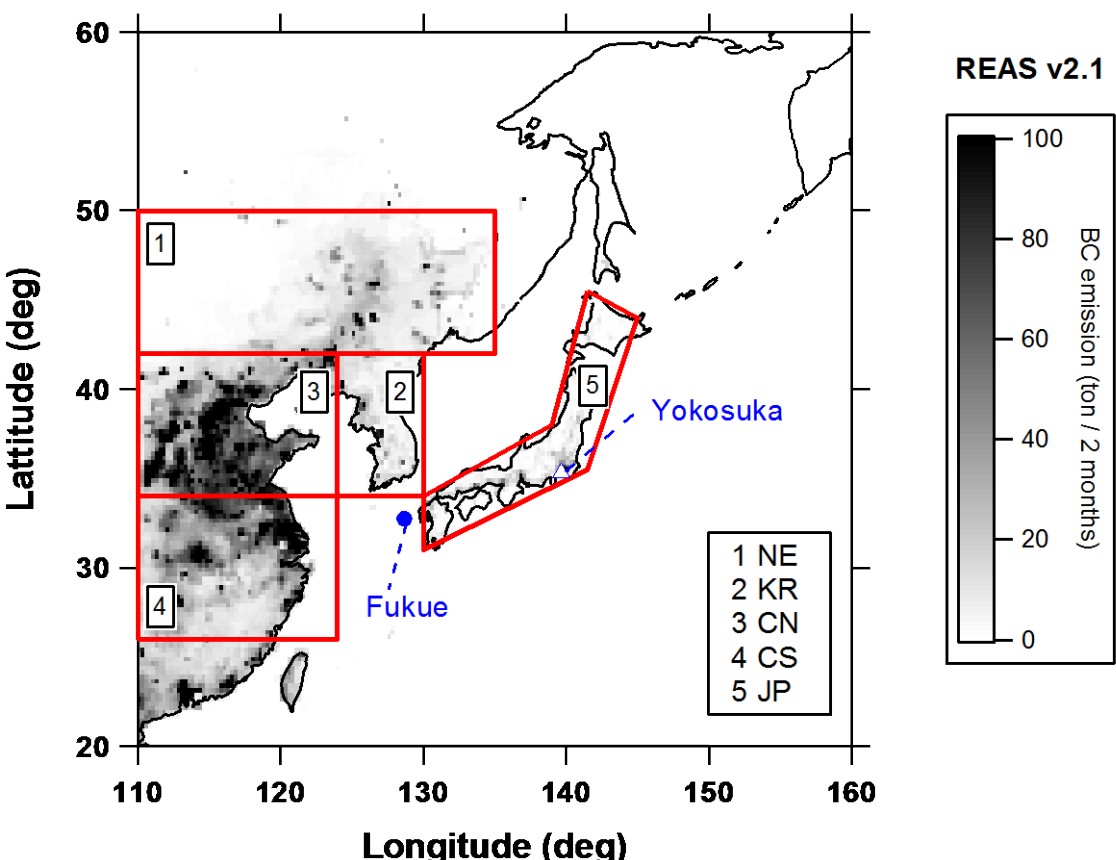



**Figure 1.** Map of the investigated region with two observation sites (Yokosuka, open
triangle; Fukue Island, closed circle) and five defined areas (1 Northeast China; 2 Korea;
3 Central North China; 4 Central South China; 5 Japan).   The bimonthly mean BC
emission rate (March-April) in 2008 is overlaid on the map (REAS ver. 2.1, Kurokawa et
al., 2013).

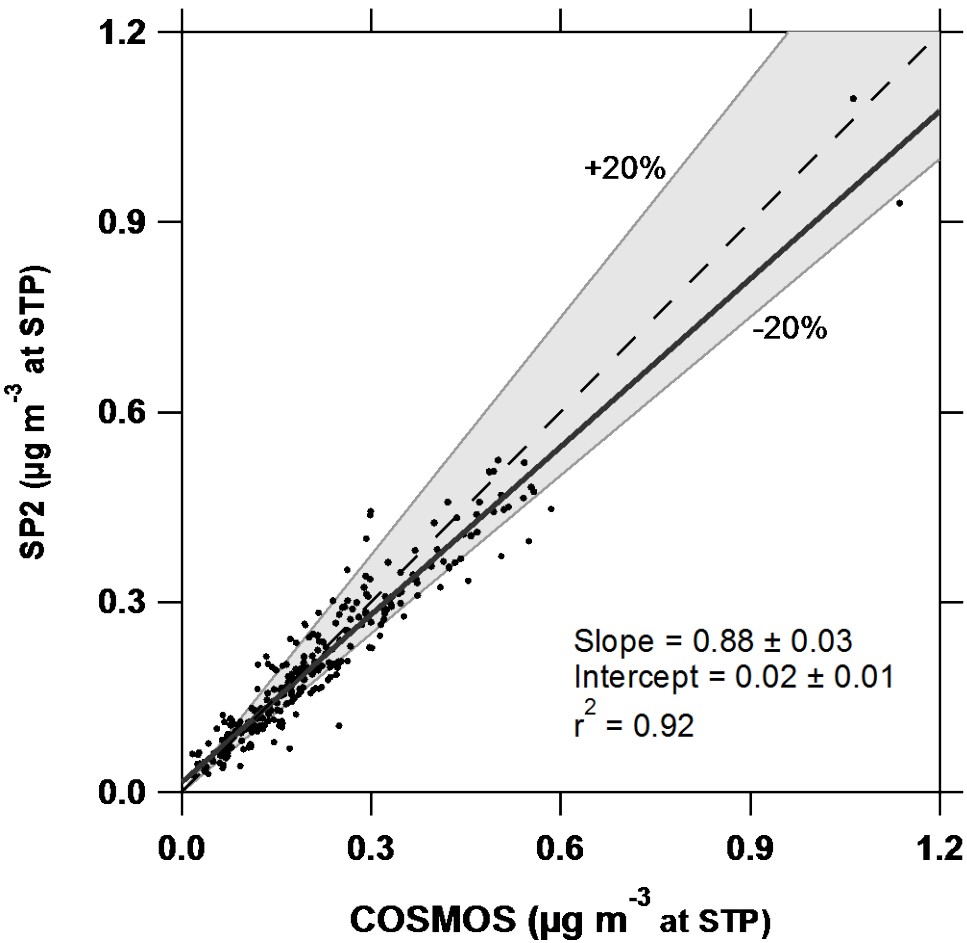



**Figure 2.** Correlation plot of SP2-rBC and COSMOS-EBC mass concentrations (at
standard temperature and pressure). The shaded region corresponds to within ±20%
from 1 : 1 line (the dashed line). The bold line depicts the linear regression line.

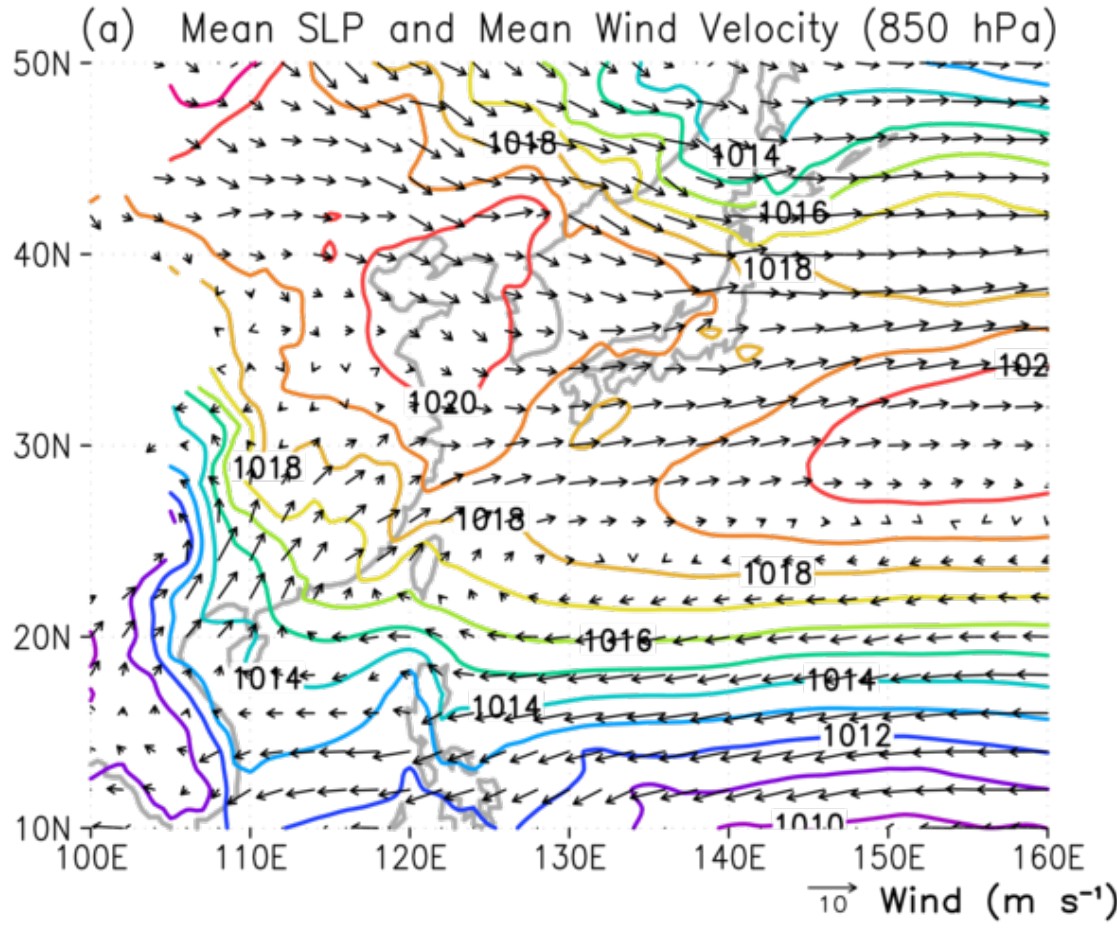


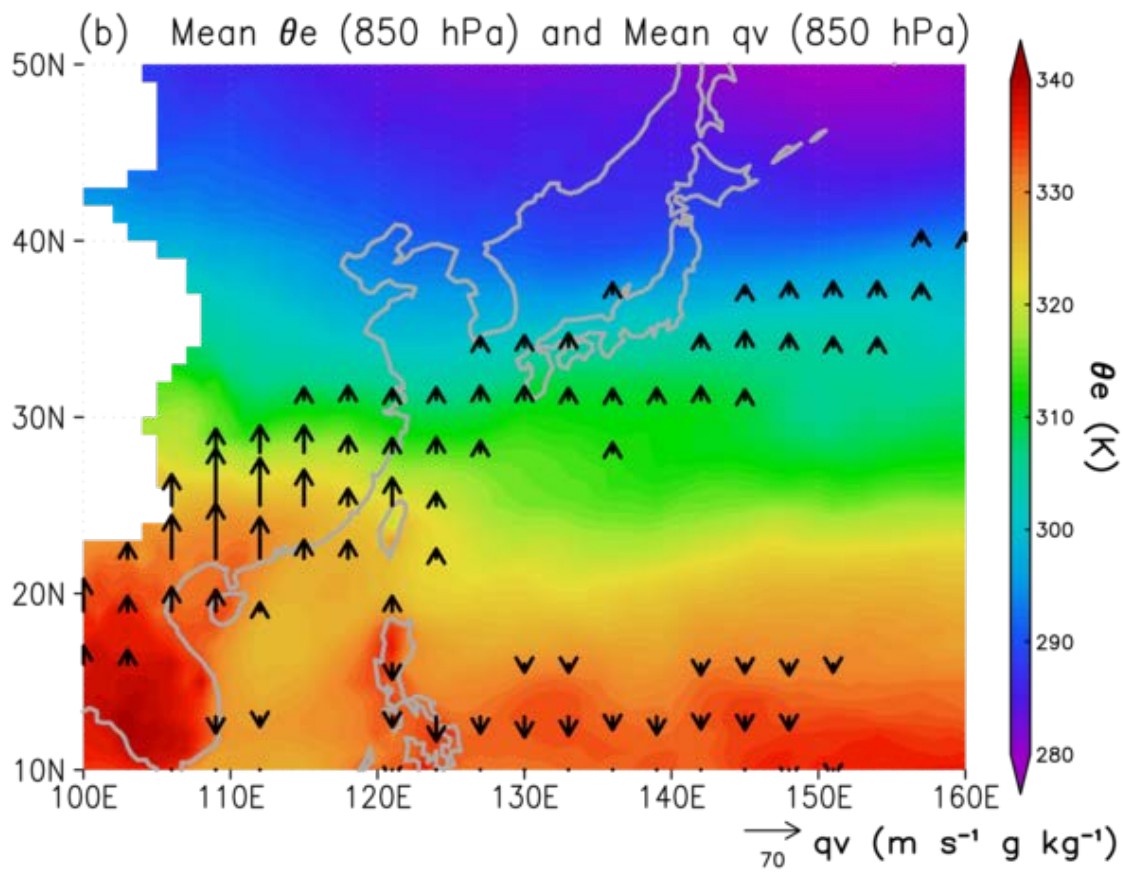


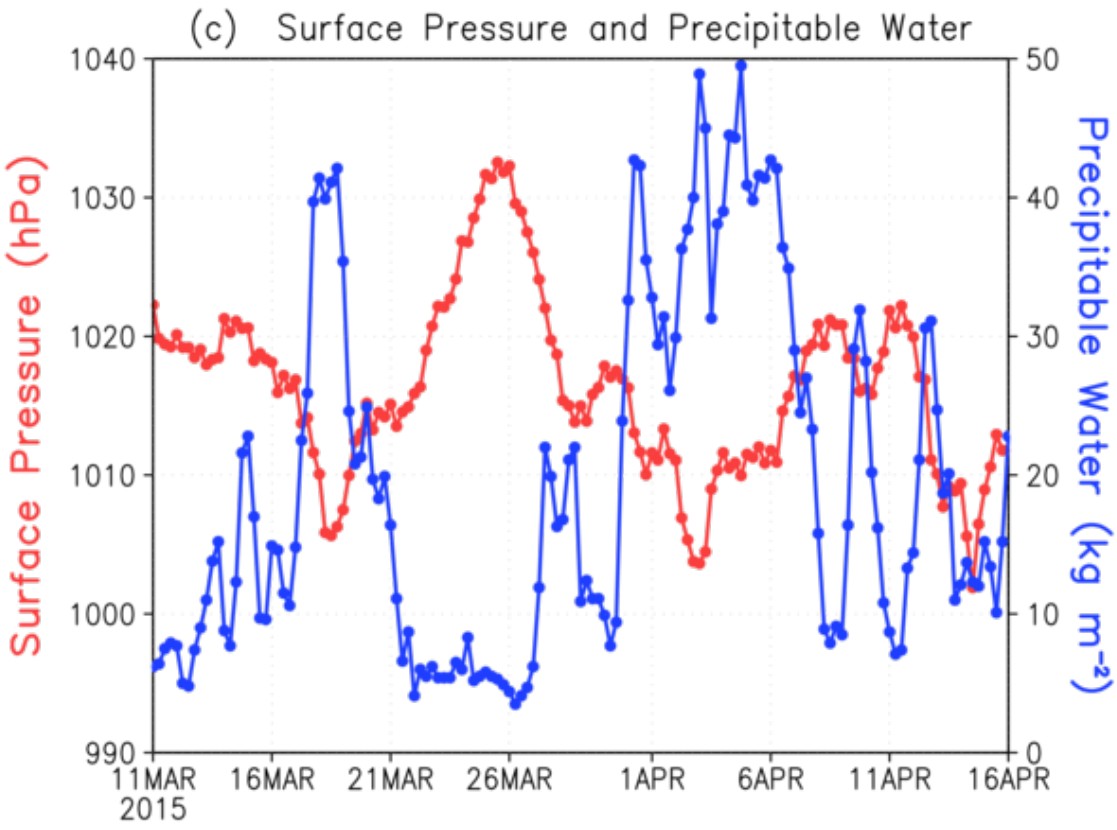



**Figure 3.** Meteorological fields in East Asia during the observation period (March 11-

April 14, 2015) based on NCEP FNL data. (a) Mean SLP (hPa, contours) and mean

horizontal wind velocity at the 850-hPa level (m s$^{-1}$). Regions without data correspond

to those of high-altitude mountains. (b) Mean θe (K) and total meridional moisture

transport (qv values) at the 850-hPa level (m s$^{-1}$ g kg$^{-1}$). Only qv vectors with

magnitudes greater than 10 m s$^{-1}$ g kg$^{-1}$ were plotted. (c) Temporal variations in the

surface pressure (hPa, red line and markers with left axis) and precipitable water (kg m$^{-2}$,

blue line and markers with right axis) at the Fukue observation site (32.75°N, 128.68°E).


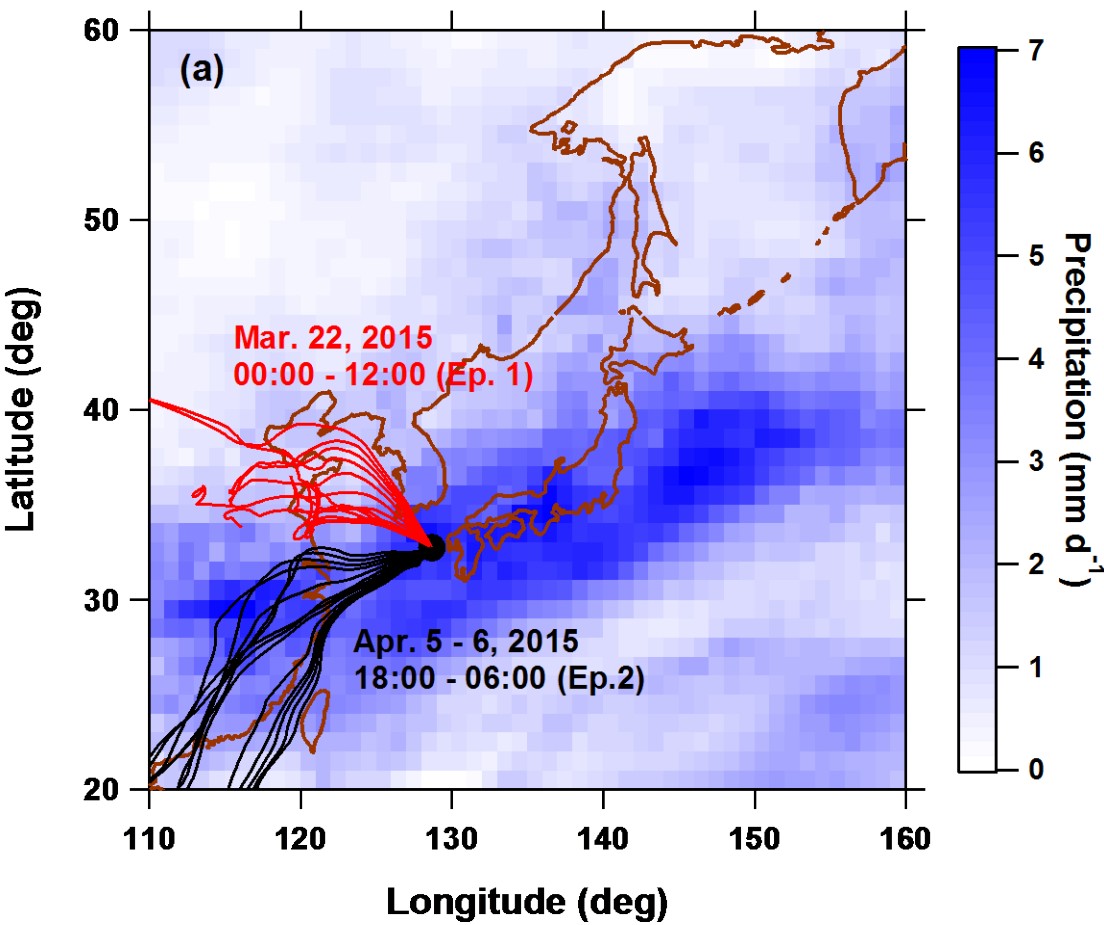


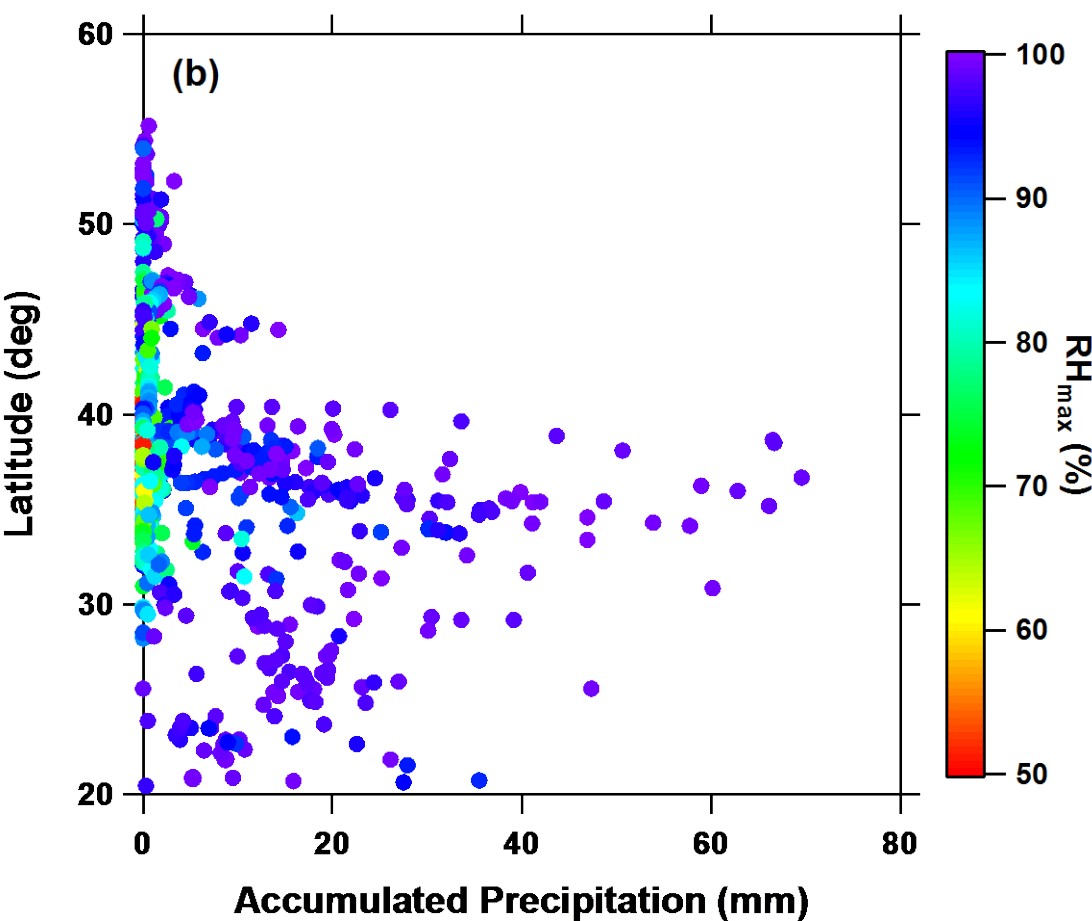



**Figure 4.** (a) Mean precipitation derived from GPCP during the observation period
(March 11-April 14, 2015). Three-day backward trajectories for selected periods are
overlaid (red lines, 00:00-12:00LT March 22, 2015 (Ep.1); black lines, 08:00LT April 5-
06:00LT April 6, 2015 (Ep.2)). (b) The relationship between APT and $Lat_{ORIG}$ (see text
for details) colored by the maximum RH along the backward trajectories.

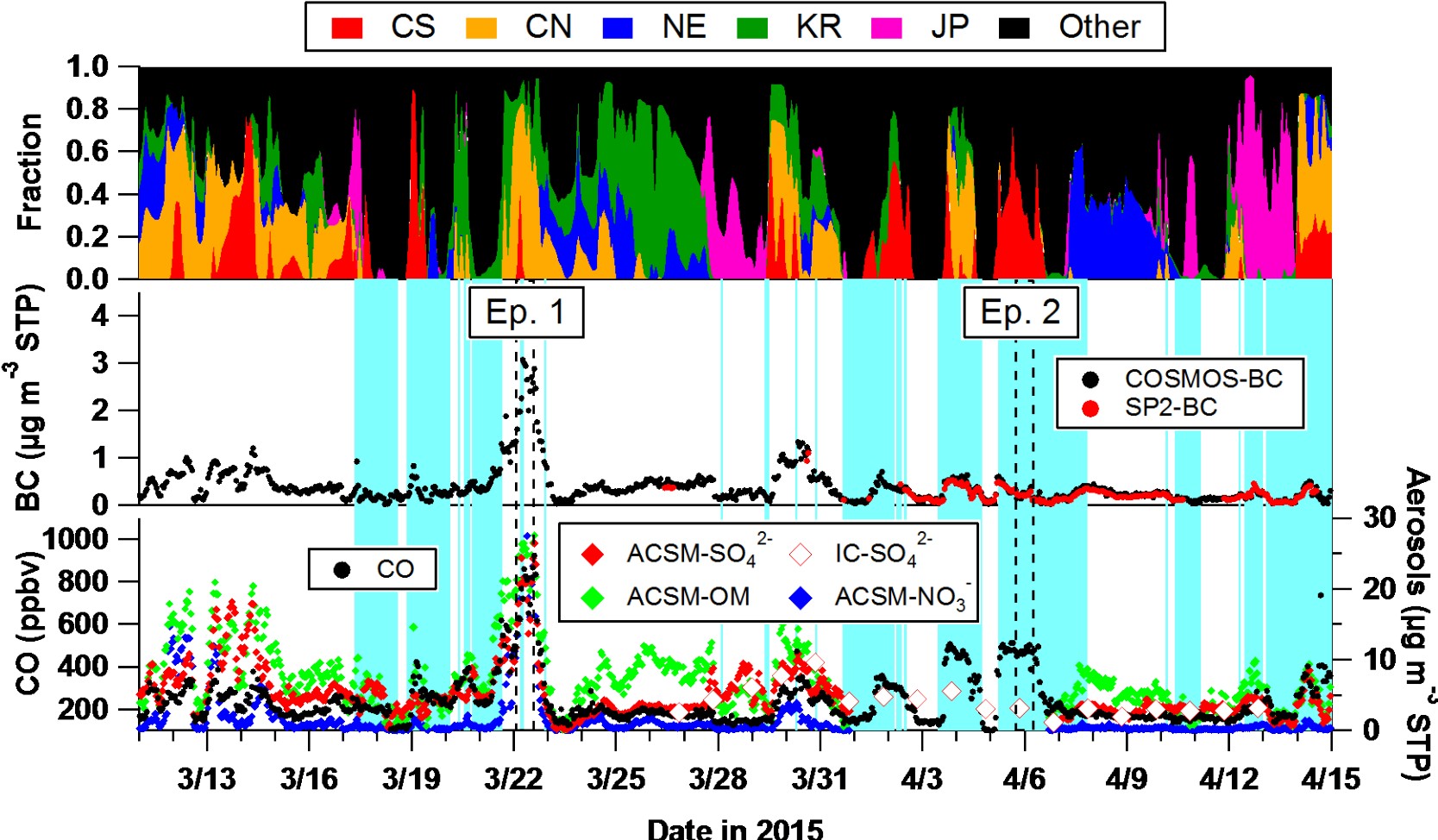

**Figure 5.** Temporal variations in air mass origin and concentration of trace species.    (Top panel) Fractional residence time of air masses
passed over selected area (Red, Central South China; Orange, Central North China; Blue, Northeast China; Green, Korea; Pink, Japan;
Black, other regions such as Ocean).    (Middle panel) mass concentrations of BC measured using the COSMOS (black markers) and SP2
(red markers).    (Bottom panel) concentrations of CO (black markers), $SO_4^{2-}$ (red closed and open makers for ACSM and IC, respectively),
ACSM-$NO_3^-$ (blue makers), and ACSM-OM (light green markers).    The periods with the APT > 3 mm are highlighted in light blue in the
middle and bottom panels.    The periods denoted as Ep.1 and Ep.2 (see the text for details) were enclosed by dashed lines.

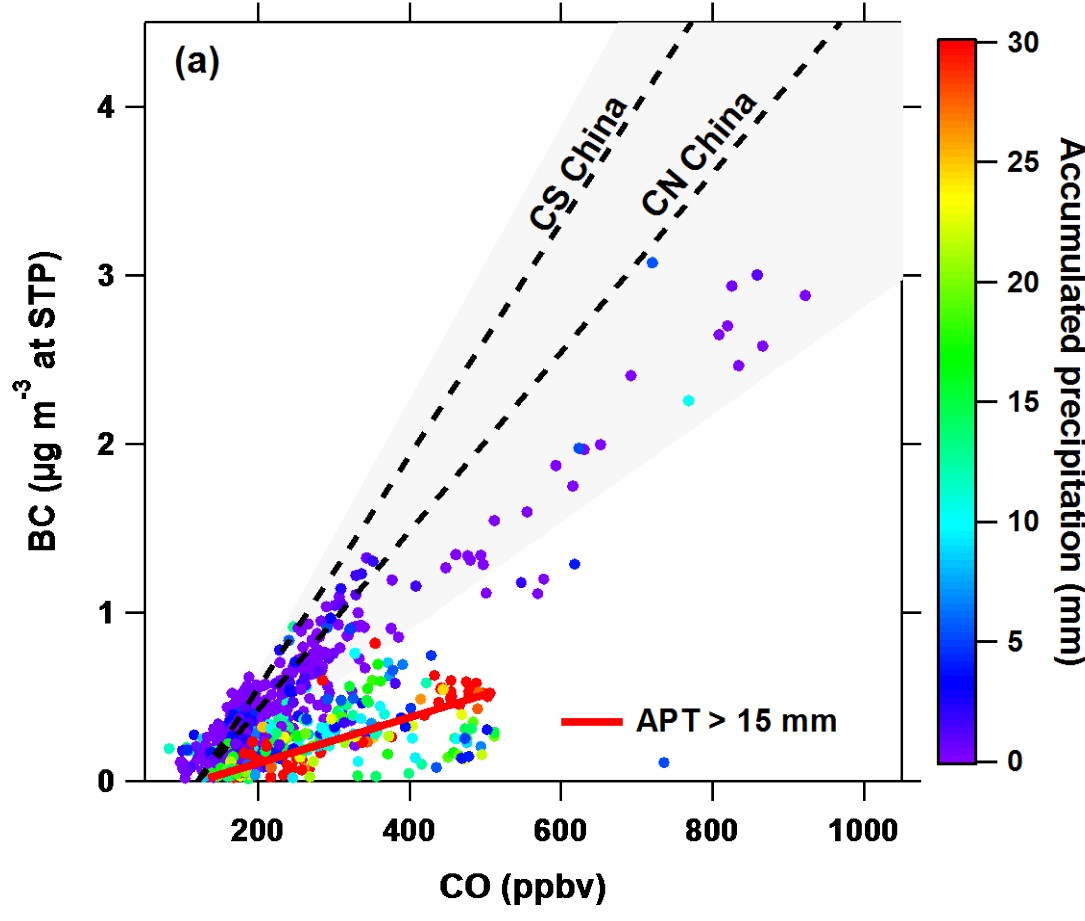

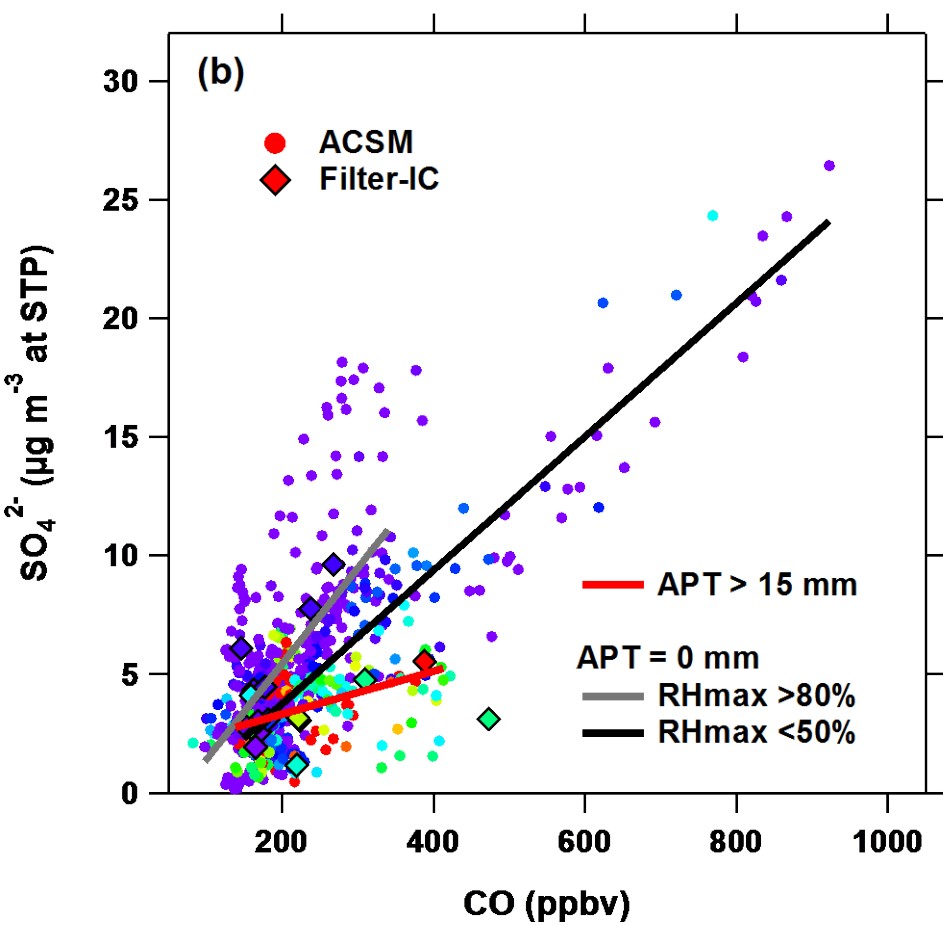

**Figure 6.** Correlation between aerosol mass concentrations and CO mixing ratio colored according to the APT. (a) BC measured by COSMOS and (b) $SO_4^{2-}$ measured by ACSM and IC (circles and diamond markers, respectively). Dashed lines and shaded area in 6 (a) represent the emission ratios of BC to CO over central North and South (CN and CS) China and their variation ranges, respectively (Kanaya et al., 2016). The bold lines shown in 6 (a) and (b) are the linear fitting to the BC/CO and ACSM-$SO_4^{2-}$/CO correlations for the selected data points, i.e., those with the APT >15 mm for BC and $SO_4^{2-}$ (red lines), those with the APT of zero and the $RH_{max}$ <50% for $SO_4^{2-}$ (black line), and those with the APT of zero and the $RH_{max}$ >80% (shaded line).

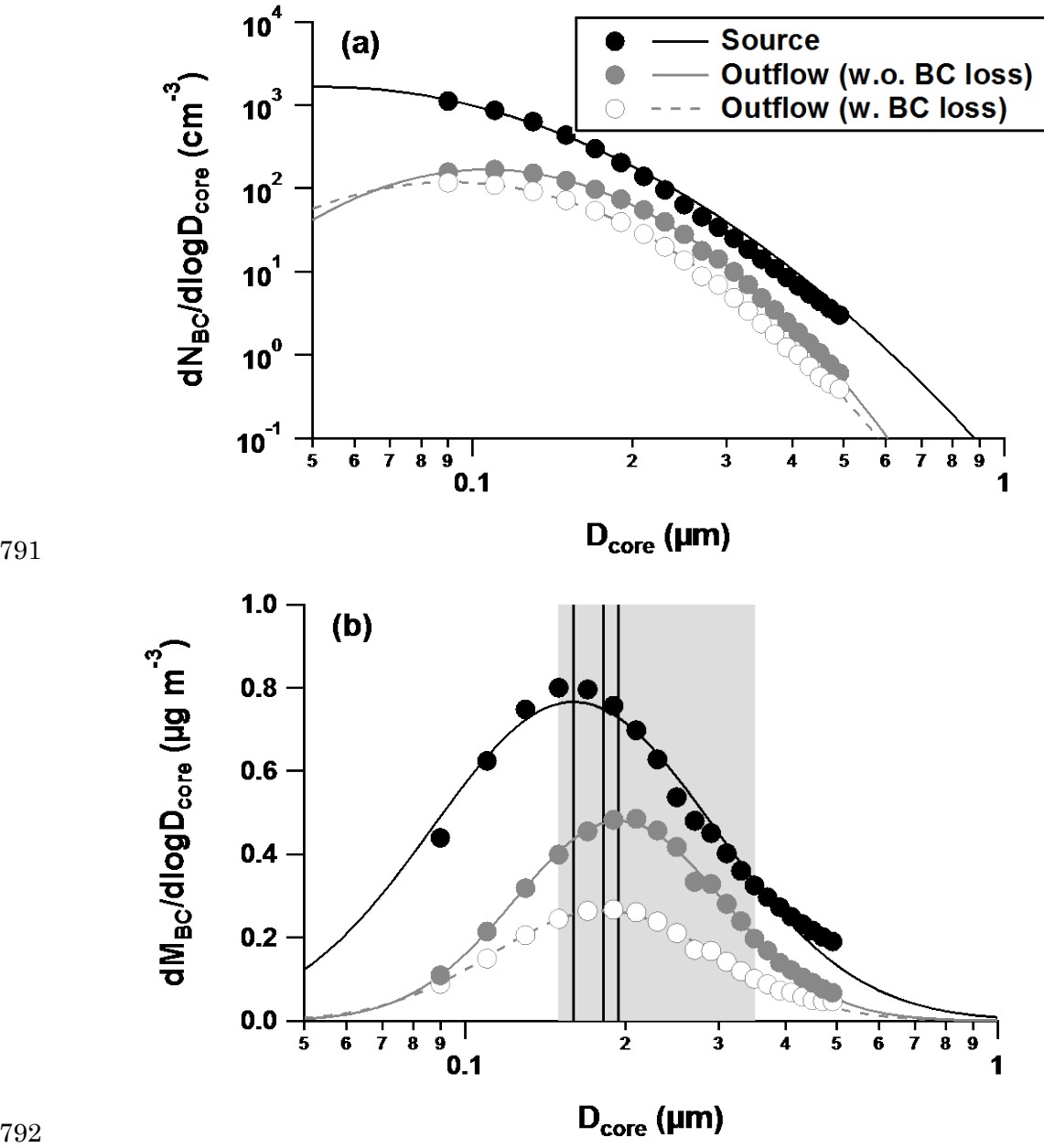



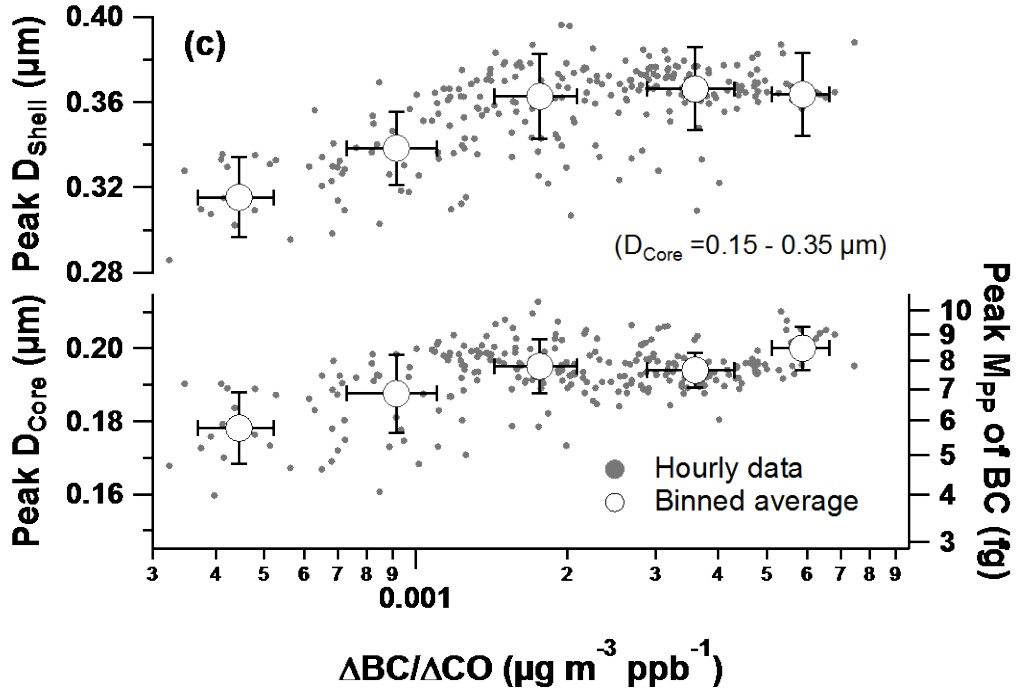



**Figure 7.** The (a) number and (b) mass size distributions of BC measured at Yokosuka
(black markers) and at Fukue Island (gray markers).    (c) The evolution of the peak $D_s$
and $D_{core}$ as a function of the degree of removal of BC.    The size distributions at Fukue
Island include the data for the outflow air masses with (open markers) and without (closed
markers) BC loss.    Lines in 7(a) and 7(b) are the lognormal fitting results.    The shaded
band in 7(b) corresponds to the size range analyzed to estimate $D_s/D_{core}$ ratios.    Vertical
lines in 7(b) represent the peak $D_{core}$ of the lognormal fit for each of three mass size
distributions.    Note that the peak $D_{core}$ of log-normal fit for the BC number size
distributions at Yokosuka was estimated from the peak $D_{core}$ of its mass size distribution
(**Table 2**).    The peak values of $D_s$ and $D_{core}$ shown in 7(c) were determined by fitting the
lognormal function to the hourly number and mass size distributions of BC-containing
particles, respectively.

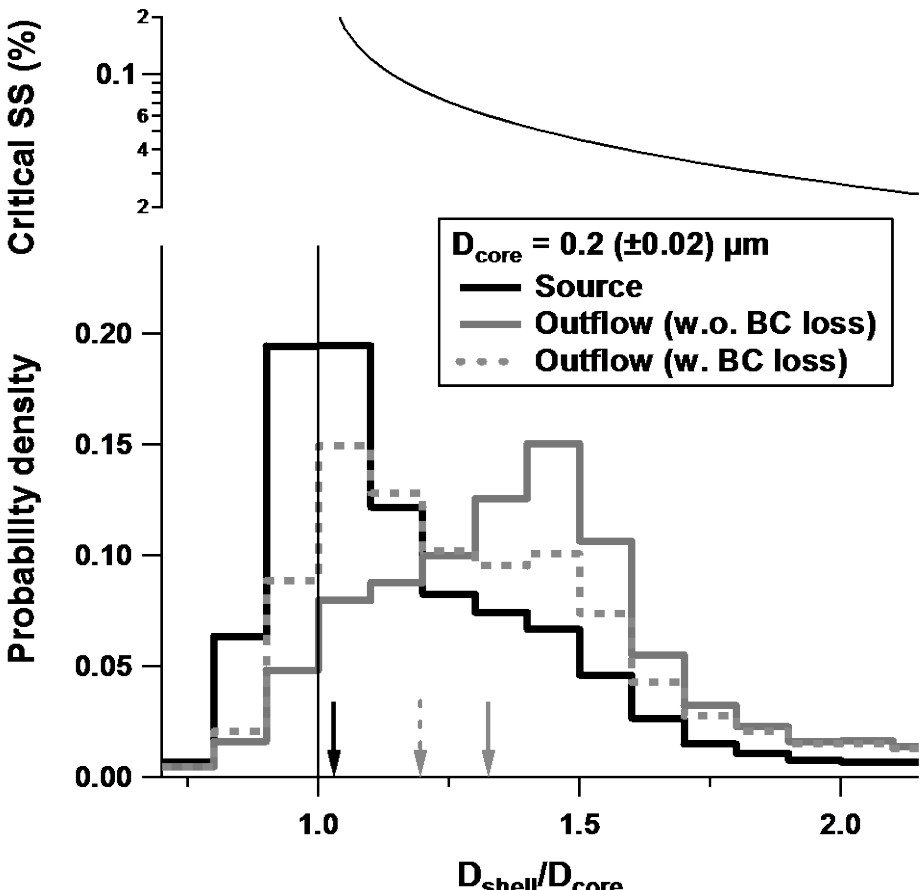



**Figure 8.** Probability density function of the estimated $D_s/D_{core}$ ratios for BC-containing
particles with the size 0.2 (±0.02) µm at Yokosuka (black line) and in the air masses of
continental outflow with (gray dashed line) and without (gray solid line) BC loss.
Vertical allows indicate the median values of $D_s/D_{core}$ ratios for three different air masses.
The estimated critical super saturation of BC-containing particles detected at Fukue
Island was also shown as a function of the $D_s/D_{core}$ ratios (see the text for details).

**Tables**
**Table 1. Mean chemical composition of fine aerosols during the observation period**

| Componnents | Period average | APT | | | |
|---|---|---|---|---|---|
| | | 0 mm | 0 mm RH$_{max}$ <50% | 0 mm RH$_{max}$ >80% | >15 mm |
| Ammonium sulfate | 44.9% | 41.8% | 34.0% | 48.9% | 50.4% |
| Ammonium nitrate | 11.7% | 15.7% | 10.7% | 8.0% | 5.0% |
| OM | 40.9% | 40.1% | 52.0% | 40.4% | 42.0% |
| BC | 2.5% | 2.4% | 3.2% | 2.6% | 2.5% |


**Table 2. Summaries of BC microphysical parameters measured at Yokosuka and Fukue Island**

| Site | Air mass type | Averaging time* (hrs) | ΔBC/ΔCO (ng m$^{-3}$ ppb$^{-1}$) | APT (mm) | Log Normal Fit Parameters Avg. (1σ) | | 1-hr Median D$_S$/D$_{core}$ for selected D$_{core}$ Avg. (1σ) | | | |
|---|---|---|---|---|---|---|---|---|---|---|
| | | | | | MMD (μm) | σ$_g$ | 0.15 - 0.2 | 0.2 - 0.25 | 0.25 - 0.3 | 0.3 - 0.35 (μm) |
| Yokosuka | Source | 184 | - | - | 0.160 (0.019) | 1.84 (0.08) | 1.18 (0.07) | 1.15 (0.06) | 1.10 (0.04) | 1.07 (0.04) |
| Fukue | Outflow | 87 | >3 | 1.2 | 0.195 (0.005) | 1.57 (0.05) | 1.37 (0.05) | 1.32 (0.03) | 1.21 (0.03) | 1.17 (0.03) |
| Fukue | Outflow | 51 | <1 | 19.9 | 0.182 (0.011) | 1.62 (0.09) | 1.25 (0.05) | 1.24 (0.04) | 1.16 (0.02) | 1.12 (0.03) |

*Time used for calculating averaged statistics of the microphysical properties of BC-containing particles.