# Peer review of "Alteration of the size distributions and mixing states of black"

_Atmospheric Chemistry and Physics, 2016_

## Referee Comment (RC1) · Anonymous Referee #1 · 20 Jul 2016

The manuscript discusses ground-based measurements, with several instruments, of black carbon (BC) near an industrial source region and at a location removed from the source to study the effects of precipitation on the size distribution and properties of the BC-containing particles. The manuscript is well written and competently explains the study, but several of the arguments do not seem supported by the data. If the comments below are addressed I would recommend that the manuscript be accepted for publication.

The title refers to "microphysical properties," which is true, but perhaps "size distribution and amount of associated non-BC material" would be more accurate, as the former term implies a host of properties that were not addressed.

[Figure]

Line 56: The sweeping statement that "washout cannot substantially affect the lifetime of atmospheric BC-containing particles," even with a reference to Seinfeld and Pandis, seems difficult to justify. Do the authors mean that because most of the BC-containing particles have diameters of several hundred nanometers, their ability to be scavenged by falling precipitation is not very large? This would seem to depend on the intensity of precipitation.

Line 148: Rather than "lower and upper boundaries" it would be preferable to state "outside the diameter range ..." so that it is clear what size is being referred to.

Lines 152-154: Some discussion of why the EC and rBC concentrations differ, and especially why the rBC concentration is less, seems to be necessary.

Line 168: Some justification for the selection of 0.5 as the collection efficiency for sulfate in the ACSM is required.

Line 206: Some discussion of how sensitive the results are to different choices for the percentile (i.e., does the background value change if concentrations lower than the 10th percentile were averaged?) would be helpful, or better yet, a distribution of the CO concentrations should be shown.

Line 277: The statement that the ACSM-SO4 and the IC-SO4 "generally agreed well" is true, but from Fig. 5c there appears to be little variability in either at concurrent times when comparison could be made.

Line 284: It is not clear why the positive correlation of SO4 and CO suggests that the SO4 was secondary and that SO4 contributed to the BC coatings; more explanation of these assumptions/conclusions is required.

Line 290: The authors note "the small variability of SO4/CO ratios," yet Figure 6b shows that these ratios vary considerably.

Lines 294, 297: The two "experiments," which consisted of two brief time periods out of a month of data, were used to justify conclusions regarding flow patterns. While the

results are indeed consistent with the arguments made, it seems difficult to justify such conclusions on the basis of one comparison.

Line 317: The authors refer to the SO4/CO ratio, but does this really refer to the delta-SO4/delta-CO ratio? It was unclear to me here and a number of places elsewhere in the test whether the CO and SO4 values referred to delta-CO and delta-SO4 values or not. For clarity, I would recommend using "delta-" values throughout.

Lines 317-319: The difference in slopes shown in the inset to Figure 6b doesn't seem sufficiently large, given the scatter of the data, to be significantly different, and certainly not to justify the conclusion that the controlling process is rainout.

Line 343: Here and elsewhere the argument is made that aging leads to growth of BC particles, which is well accepted, but such aging can also lead to loss of larger particles through rainout, het size distributions in Figure 7 doesn't show much of a difference between size distributions for air masses with BC loss and those without, and certainly not more of a difference for larger BC particles than for smaller ones. This discrepancy requires explanation.

Line 345: The statement that "small BC-containing particles were scavenged by larger particles in the coagulation process" is a hypothesis, but stated as truth. It would seem that concentrations are too low for much coagulation over the brief period (a few days), especially for particles that are many tens of nanometers in diameter. Calculations or a simple model would be required to support this hypothesis.

Line 353: It would be preferable, and less ambiguous, to rephrase "BC size of 0.2" to "BC diameter of 0.2".

Line 368: The discussion focused on transport pathways of particles in the particular region of the study, but I was expecting more discussion on the results, what they mean, and so forth. There seemed to be little relevance to the second paragraph of the discussion.

Line 372: The decrease in the peak diameter of the mass size distribution is very small, and within uncertainty.

Line 373: The statement that the evidence implies selective removal of large BC-containing particles is not supported by Figure 7, which shows a very slight difference in the size distribution between "with BC loss" and "without BC loss" but not apparent selective decrease of larger particles. If there were selective removal, I would expect the size distribution to not be lognormal, but to have a deficit on the large side below what a lognormal would be.

Figure 3a is very difficult to read; could it be made larger? Figure 3b requires units for q_v to accompany the scale. Figure 4a should be made larger also, if possible. Figure 5b: it is difficult to distinguish the COSMOS and SP2 BC values; perhaps make one red and the other black? Figure 6a: do the axes refer to delta-CO and delta-BC? If so, they should be labeled as such. Figure 6b, inset: what does "all data" refer to? If this is to label the gray dot, then it is not clear.

---

## Referee Comment (RC2) · G R McMeeking (Referee) · 10 Sep 2016

The authors present a one month case study examining measurements of black carbon properties at a remote island site, using co-located measurements of CO and sub-micron aerosol composition and reanalysis data to evaluate precipitation impacts on the observed properties. The manuscript focuses on contrasting observed properties during periods with differing accumulated precipitation along backward trajectories. The paper is well prepared and well organized and the subject is well within the topic area for ACP. There are several areas where minor revisions are needed, however, before the paper can be recommended for publication.

I agree with the points raised by Reviewer #1, so have tried to not repeat too much

of what has been already raised. The comments should be addressed in a revised manuscript. In addition:

+ Given the focus of the manuscript, the introduction would benefit from a more thorough discussion of the various BC removal mechanisms, with more mechanistic details given as to why various processes may or may not be important in the study area. Distinction should be made between in-cloud processes (nucleation scavenging versus scavenging by pre-existing droplets), below-cloud (washout) and dry deposition.

+ Two points regarding reported SP2-measured BC number/mass distributions. First, the manuscript needs to make it more clear when BC core versus shell diameters are being discussed, especially when linking the observations to theory. For example, while it is true we would expect larger particles to be removed in air masses heavily impacted by precipitation, the effects on BC core distributions will be confounded by other material mixed with the cores. Related to this, the diameter range for which the optical sizing of the BC particles should be provided in the methods section. Second, small changes in the detection efficiency of the SP2 at its lower limit due to changes in cavity laser power can look like changes in BC core number distribution. A short statement regarding any checks on cavity laser power or other approaches used to ensure consistent behavior at lower size limits for the instrument would be useful.

+ Potentially useful additional information provided by the ACSM is being ignored by examining only sulfate. Is there a reason for this?
* * *

---

## Author Comment (AC1) · 18 Oct 2016

Anonymous Referee #1 Review of "Alteration of the microphysical properties of black carbon through transport in the boundary layer in East Asia" by Takuma Miyakawa et al. submitted to Atmospheric Chemistry and Physics.

We appreciate the reviewer's helpful and constructive comments on the manuscript entitled "Alteration of the microphysical properties of black carbon through transport in the boundary layer in East Asia". As the reviewers suggested, we have modified the manuscript. Major points for the revisions are listed as follows. 1) Title has been changed. 2) Supporting information (SI) has been prepared. 3) We have modified the discussion section.

[Figure]

The manuscript discusses ground-based measurements, with several instruments, of black carbon (BC) near an industrial source region and at a location removed from the source to study the effects of precipitation on the size distribution and properties of the BC-containing particles. The manuscript is well written and competently explains the study, but several of the arguments do not seem supported by the data. If the comments below are addressed I would recommend that the manuscript be accepted for publication. The title refers to "microphysical properties," which is true, but perhaps "size distribution and amount of associated non-BC material" would be more accurate, as the former term implies a host of properties that were not addressed.

Response>As the reviewer suggested, this study has investigated a part of the microphysical parameters of BC. Shape and chemical composition of BC-containing particles, which were not directly measured in this study, are important for considering the climatic impacts of BC-containing particles. However, chemical composition of non-refractory (non-BC) materials for both BC-free and -containing particles was measured using an Aerosol Chemical Speciation Monitor (ACSM). We addressed just simply the mixing state of BC-containing particles, and therefore revised the title slightly to "Alteration of the size distributions and mixing states of black carbon through transport in the boundary layer in East Asia".

Line 56: The sweeping statement that "washout cannot substantially affect the lifetime of atmospheric BC-containing particles," even with a reference to Seinfeld and Pandis, seems difficult to justify. Do the authors mean that because most of the BC-containing particles have diameters of several hundred nanometers, their ability to be scavenged by falling precipitation is not very large? This would seem to depend on the intensity of precipitation.

Response>As the reviewer suggested, the accumulation mode aerosols including BC are not effectively removed by the falling rain droplets. Washout process is dependent on the precipitation intensity (PI) and rain drop size as well as the particle size range. In this study, the information of rain drop size is not available. The average PIs along

a backward trajectory were calculated for the rain period in 3d-backward time (PI > 0 mm h-1). They ranged from 0.1 to 2.5 mm h-1 (median = ~0.6 mm h-1). Using the PI value of 0.6 mm h-1, the scavenging rates of accumulation mode particles were estimated to be 6E-3 h-1 (6E-5 h-1) with the assumed rain drop diameter of 0.2 mm (2 mm). The corresponding time constants are around 7 and 694 days. These are longer than the typical transport time from the continent to the observation site. The details are described in SI.

Line 148: Rather than "lower and upper boundaries" it would be preferable to state "outside the diameter range . . ." so that it is clear what size is being referred to.

Response>We have revised as suggested.

Lines 152-154: Some discussion of why the EC and rBC concentrations differ, and especially why the rBC concentration is less, seems to be necessary. Line 168: Some justification for the selection of 0.5 as the collection efficiency for sulfate in the ACSM is required.

Response>In this study, we compared rBC with effective BC (EBC) measured using a light absorption technique (COSMOS). As we stated in the original manuscript, the difference between rBC and EBC is within the uncertainties related to both measurements. One of the unclear uncertainties, which have not well been studied, is the detection sensitivity of SP2 to the ambient rBC particles (incandescence signal intensity per rBC particle mass, SLII-mpp) in a remote atmosphere. It was found in previous studies (Moteki and Kondo, 2010; Miyakawa et al. 2016) that the SLII-mpp relationship of fullerene soot (FS) particles, which is used as a calibration standard for the SP2, is similar to that of ambient rBC particles in urban/industrial area. We hence assume the same sensitivity of SP2 to the ambient rBC in a remote atmosphere as that of FS particles and rBC particles in urban/industrial area. I added some explanations on the related uncertainties to the section 2.1 in the revised manuscript. The collection efficiency of ACSM-SO42- was derived from Yoshino et al. (2016). This paper is included

in the reference list of the revised manuscript.

Line 206: Some discussion of how sensitive the results are to different choices for the percentile (i.e., does the background value change if concentrations lower than the 10th percentile were averaged?) would be helpful, or better yet, a distribution of the CO concentrations should be shown.

Response>When we set 10th percentile of CO mixing ratio as the threshold value, the derived background CO mixing ratio was calculated to be 131 ppb, which is slightly higher than the original value (120 ppb). We prepared SI including the descriptions on the determination of the background CO mixing ratio. Please see SI for details.

Line 277: The statement that the ACSM-SO4 and the IC-SO4 "generally agreed well" is true, but from Fig. 5c there appears to be little variability in either at concurrent times when comparison could be made.

Response>The variability in IC-SO4$_2^-$ mass concentration was $\sim$9 $\mu$g m-3 at STP (min - max $\sim$1 - $\sim$10). Wider range of concentrations ($<\sim$20 $\mu$g m-3) were observed during an intercomparison experiment in Queens/New York (Drewnick et al., 2003). To the best of our knowledge, the observed range was larger enough to discuss the intercomparison results. For example, Takegawa et al. (2005) reported the intercomparison results of SO4$_2^-$ mass concentration between Aerodyne AMS and PILS-IC. The range given in their study ($< \sim$7 $\mu$g m-3) is smaller than ours.

Line 284: It is not clear why the positive correlation of SO4 and CO suggests that the SO4 was secondary and that SO4 contributed to the BC coatings; more explanation of these assumptions/conclusions is required.

Response>Growth of BC-containing particles should be explained separately from the formation. Besides our observation results, previous studies support the description of formation and structure of the coating of BC in the original manuscript. As the reviewer suggested, we revised the related sentences and included more explanation in the

revised manuscript.

Line 290: The authors note "the small variability of SO4/CO ratios," yet Figure 6b shows that these ratios vary considerably.

Response>As the reviewer suggested, this statement and Figure 6b seem to contradict each other. We removed this sentence for the clarity.

Lines 294, 297: The two "experiments," which consisted of two brief time periods out of a month of data, were used to justify conclusions regarding flow patterns. While the results are indeed consistent with the arguments made, it seems difficult to justify such conclusions on the basis of one comparison.

Response>As the reviewer suggested, the results shown in this study are based on the observation during not-so-long time periods. We agree that it is actually difficult to draw the general conclusions. However, we still believe that this paper shows the significance in the observational studies of the relationship between removal process and the changes in the BC microphysical properties, because the observed meteorological conditions in the spring of 2015 were not special and similar to those with an average year. We added the sentences "The migrating anticyclone and cyclone were observed during this period, which is typically dominant in spring over East Asia (Asai et al., 1988). We here only briefly describe the meteorological fields (wind flow and precipitation) in the following." behind the first sentence in section 3.1, and modified the last sentence in section 3.5 to "As the results from this study are based on observations during a limited length of time, it would be worthwhile to further investigate the possible connections of the variabilities in BC microphysical properties and meteorological conditions in this region to provide useful constraints on more accurate evaluations of climatic impacts of BC-containing particles (Matsui, 2016)". Please see the revised manuscript for details.

Line 317: The authors refer to the SO4/CO ratio, but does this really refer to the deltaSO4/delta-CO ratio? It was unclear to me here and a number of places elsewhere in the test whether the CO and SO4 values referred to delta-CO and delta-SO4 values or not. For clarity, I would recommend using "delta-" values throughout.

Response>We clearly found the lower concentrations of SO42- relative to CO for the data with the higher APT in Figure 6b of the original manuscript. Another reason not to include the $\Delta$SO42-/$\Delta$CO ratio is the uncertainty related to the variability in the background of SO42- in East Asia. Although the use of the same data treatment would be clear for the readers, we did not quantitatively analyze the hourly $\Delta$SO42- and $\Delta$CO values for considering the relative enhancements of SO42- to CO in this study. We hence added the sentences to explain why we do not analyze $\Delta$ values in the revised manuscript in section 2.2.

Lines 317-319: The difference in slopes shown in the inset to Figure 6b doesn't seem sufficiently large, given the scatter of the data, to be significantly different, and certainly not to justify the conclusion that the controlling process is rainout.

Response>The rainout lowered the transport efficiency of SO42- as well as BC (to CO). However, the cloud process not associated with the precipitation can affect the relative increases of SO42- concentration. The major purpose to include this figure is to elucidate the impact of the cloud process on the aqueous-phase formation of SO42-, and is not to discuss the loss processes. Figure 6b is modified in the revised manuscript to clarify the data points with the APT of zero (no precipitation through the transport). These data points are highlighted by marking using cross markers. Please see the revised Figure 6b for details.

Line 343: Here and elsewhere the argument is made that aging leads to growth of BC particles, which is well accepted, but such aging can also lead to loss of larger particles through rainout, het size distributions in Figure 7 doesn't show much of a difference between size distributions for air masses with BC loss and those without, and certainly not more of a difference for larger BC particles than for smaller ones. This discrepancy requires explanation.

[Figure]

Response>All the size distributions shown in Figure 7 are normalized by the number or mass integrated for the measured size range, which is described in the caption of this figure. The "absolute" size distributions show more differences between with and without BC loss. We modified the size distributions from "normalized" to "absolute" and added a new figure (fig 7c of the revised manuscript) of the relationship between BC peak diameters and $\Delta BC/\Delta CO$ (i.e., degree of the removal of BC). This figure clarifies the significance of the observed changes in the peak diameter. Please see the revised figure for more details.

Line 345: The statement that "small BC-containing particles were scavenged by larger particles in the coagulation process" is a hypothesis, but stated as truth. It would seem that concentrations are too low for much coagulation over the brief period (a few days), especially for particles that are many tens of nanometers in diameter. Calculations or a simple model would be required to support this hypothesis. Line 353: It would be preferable, and less ambiguous, to rephrase "BC size of 0.2" to "BC diameter of 0.2".

Response>In the consideration of the washout process, the removal of small BC-containing particles through the washout is expected to be significant as well as the coagulation process. We hence describe the possibility of both processes in the revised manuscript. We rephrased "BC size of 0.2" to "BC diameter of 0.2".

Line 368: The discussion focused on transport pathways of particles in the particular region of the study, but I was expecting more discussion on the results, what they mean, and so forth. There seemed to be little relevance to the second paragraph of the discussion.

Response>We reorganized the discussion part (section 3.5). We merged and reorganized the first paragraph and the half of the second paragraph into one paragraph. The latter half of the first paragraph of section 3.5 discusses the observed features and its relevance to the finding in previous studies. We consider that the relationship between transport pathways (i.e., processes during transport) and its impact on the

aerosol particles is a key and relevant to our observation results. We hence modified the sentences of the third (second in the revised manuscript) paragraph.

Line 372: The decrease in the peak diameter of the mass size distribution is very small, and within uncertainty.

Response>The change in the peak diameter is small, however, significant. Corresponded change in BC mass is $\sim 1$ fg/particle. This difference can be resolved by the SP2 and beyond the uncertainty. The variabilities of the peak diameters are summarized in Table 1 in the original and revised manuscript and are smaller than those measured. As we described in the above, we added a new figure to show the tendency of the BC particle diameter as a function of the degree of BC removal (Fig 7c of the revised manuscript).

Line 373: The statement that the evidence implies selective removal of large BC containing particles is not supported by Figure 7, which shows a very slight difference in the size distribution between "with BC loss" and "without BC loss" but not apparent selective decrease of larger particles. If there were selective removal, I would expect the size distribution to not be lognormal, but to have a deficit on the large side below what a lognormal would be. Figure 3a is very difficult to read; could it be made larger? Figure 3b requires units for q_v to accompany the scale. Figure 4a should be made larger also, if possible. Figure 5b: it is difficult to distinguish the COSMOS and SP2 BC values; perhaps make one red and the other black? Figure 6a: do the axes refer to delta-CO and delta-BC? If so, they should be labeled as such. Figure 6b, inset: what does "all data" refer to? If this is to label the gray dot, then it is not clear.

Response>The activation of aerosol particles to cloud droplets has occurred during transport. We did not observe the aerosol particles below the convective cloud, because the migratory cyclone was the dominant process for the upward transport in spring in East Asia. We thus considered that SP2 detected BC-containing particles which have been aged (about a half $\sim$ a day, typical transport time) since affected by

the wet removal. The size distributions of BC-containing particles can change during transport again after the rainout process, and therefore do not always conserve the original shape. We have corrected some figures as suggested. We enlarged all figures as large as possible as suggested. Units of all parameters in Fig 3 were clarified in the modified one. The color of SP2-BC in Fig 5 was changed to red. Axes of Fig 6a do not refer to delta (so we didn't change). Fig 6b was modified, because it was not clear. All the values in Figure 6 are absolute concentrations (not delta). Fig 7c was newly added (Please see the texts for details).

References Asai, T., Y. Kodama, and J.-C. Zhu (1988), Long-term variations of cyclone activities in East Asia, Adv. Atmos. Sci., 5, 149−158. Drewnick, F., Schwab, J. J., Hogrefe, O., Peters, S., Husain, L., Diamond, D., Weber, R., and Demerjian, K. L. (2003), Intercomparison and Evaluation of Four Semi-Continuous PM2.5 Sulfate Instruments, Atmos. Environ., 37:3335–3350. Matsui, H., Black carbon simulations using a size- and mixing-state-resolved three-dimensional model: 1. Radiative effects and their uncertainties (2016), J. Geophys. Res. Atmos., 121, 1793–1807, doi:10.1002/2015JD023998 Takami, A., et al. (2013), Structural analysis of aerosol particles by microscopic observation using a time-of-flight secondary ion mass spectrometer, J. Geophys. Res. Atmos., 118, 6726–6737, doi:10.1002/jgrd.50477. Takegawa, N., Miyazaki, Y., Kondo, Y., Komazaki, Y., Miyakawa, T., Jimenez, J. L., Jayne, J. T., Worsnop, D. R., Allan, J. D., and Weber, R. J. (2005), Characterization of an Aerodyne Aerosol Mass Spectrometer (AMS): Intercomparison with Other Aerosol Instruments, Aerosol Sci. Technol., 39:760–770. Yoshino, A., A. Takam, K. Sato, A. Shimizu, N. Kaneyasu, S. Hatakeyama, K. Hara, and M. Hayashi (2016), Influence of Trans-Boundary Air Pollution on the Urban Atmosphere in Fukuoka, Japan, Atmosphere, 7, 51, doi:10.3390/atmos7040051

Please also note the supplement to this comment:
http://www.atmos-chem-phys-discuss.net/acp-2016-570/acp-2016-570-AC1-supplement.pdf

[Figure]

**Supplement:**

**Supporting Information of**

**Alteration of the size distributions and mixing states of black carbon through transport in the boundary layer in East Asia**

Miyakawa et al.

Correspondence to Takuma Miyakawa (miyakawat@jamstec.go.jp)

**S1. Determination of the background mixing ratio of carbon monoxide (CO)**

We assume the 5th percentile value of CO mixing ratio (138 ppb) as a threshold value to extract its background level ($CO_{bg}$). $CO_{bg}$ is defined as the average of CO mixing ratios below the 5th percentile in this study, and is calculated to be 120 ppb. When we change the threshold from 5th to 10th percentiles (146 ppb), derived $CO_{bg}$ increases from 120 ppb to 131 ppb. Figure S1 depicts the probability density function of the observed CO mixing ratio with the assumed threshold. It is suggested that the assumption of the threshold value slightly affected the estimation of $CO_{bg}$.

[Figure]

Figure S1. Probability density of measured CO mixing ratio (shaded bars). Red and blue vertical lines correspond to the 5th and 10th percentile values of the observed CO mixing ratios.

**S2. Washout process**

In this study, we assume that the most important loss process of BC-containing

particles is their cloud droplet activation and subsequent precipitation (i.e., rainout). There is another wet removal process in which BC-containing particles are scavenged by falling rain droplets (i.e., washout). The scavenging rate depends on the particle size ($d_p$), raindrop size ($D_p$), and precipitation intensity (PI) (Seinfeld and Pandis, 2006). The accumulation mode aerosols (diameter of 0.1 – 1 μm) are less efficiently removed by the washout process, because the particle-raindrop collision efficiency has a minimum for the particle size range of the accumulation mode aerosols.

[Figure]

Figure S2. The probability density of the average of PI along trajectory for 3d-backward time (black line) and rain period (red line).

The average PIs along a backward trajectory were calculated for the rain period in 3d-backward time (PI > 0 mm h$^{-1}$). The probability of them is shown in Figure S2. They ranged from 0.1 to 2.5 mm h$^{-1}$ (median = ~0.6 mm h$^{-1}$). Using the PI value of 0.6 mm h$^{-1}$, the scavenging rates of accumulation mode particles were estimated to be 6E-3 h$^{-1}$ (6E-5 h$^{-1}$) with the assumed rain drop diameter of 0.2 mm (2 mm). The corresponding time constants are around 7 (694) days. These are longer than the typical transport time from the continent to the observation site. The raindrop diameter is not available. However, the derived time constants of the washout process for the accumulation mode aerosols are longer than those of the air mass transport from the continent to the observation site (<~2 days). On average, the loss particle mass concentration was controlled mainly by the rainout process.

In contrast to the accumulation mode aerosols, the Aitken mode (<0.1 μm) aerosols can be efficiently removed by the washout process (Seinfeld and Pandis, 2006). It is

important to consider the washout process for the quantitative analysis of the loss of the number concentrations of BC-containing particles, which is beyond of the scope of this study. However, the washout process can partly account for the observed changes in the number size distributions of BC-containing particles.

**References**

Seinfeld, J.H., and Pandis, S. N., Atmospheric Chemistry and Physics, 2nd ed., John

    Wiley &Sons, New York, 2006

---

## Author Comment (AC2) · 18 Oct 2016

We appreciate your helpful and constructive comments on the manuscript entitled "Alteration of the microphysical properties of black carbon through transport in the boundary layer in East Asia". As the two reviewers suggested, we have modified the manuscript. Major points for the revisions are listed as follows. 1) Title has been changed. 2) Supporting information (SI) has been prepared. 3) We have modified the discussion section. *Note the reviewers' comments in bold.

The authors present a one month case study examining measurements of black carbon properties at a remote island site, using co-located measurements of CO and sub-micron aerosol composition and reanalysis data to evaluate precipitation impacts on the observed properties. The manuscript focuses on contrasting observed properties during periods with differing accumulated precipitation along backward trajectories. The paper is well prepared and well organized and the subject is well within the topic area for ACP. There are several areas where minor revisions are needed, however, before the paper can be recommended for publication. I agree with the points raised by Reviewer #1, so have tried to not repeat too much of what has been already raised. The comments should be addressed in a revised manuscript. In addition: + Given the focus of the manuscript, the introduction would benefit from a more thorough discussion of the various BC removal mechanisms, with more mechanistic details given as to why various processes may or may not be important in the study area. Distinction should be made between in-cloud processes (nucleation scavenging versus scavenging by pre-existing droplets), below-cloud (washout) and dry deposition.

Response> We added the sentence describing the removal processes of BC to the second paragraph of section "Introduction". Relate to this, as the reviewer #1 suggested, we have modified the descriptions on the relative importance of the washout (to the rainout) (in section 3.3 of the original manuscript). A previous study (Kanaya et al., 2016) is referred for quantitatively elucidating the impacts of dry deposition. The timescale of the removal through the washout was evaluated in this study to be longer than the typical time scale of the transport (details in S.I. newly-prepared). Please see the revised manuscript and SI for more details.

+ Two points regarding reported SP2-measured BC number/mass distributions. First, the manuscript needs to make it more clear when BC core versus shell diameters are being discussed, especially when linking the observations to theory. For example, while it is true we would expect larger particles to be removed in air masses heavily impacted by precipitation, the effects on BC core distributions will be confounded by other material mixed with the cores. Related to this, the diameter range for which the

optical sizing of the BC particles should be provided in the methods section. Second, small changes in the detection efficiency of the SP2 at its lower limit due to changes in cavity laser power can look like changes in BC core number distribution. A short statement regarding any checks on cavity laser power or other approaches used to ensure consistent behavior at lower size limits for the instrument would be useful.

Response> As the reviewer suggested, we added explanations on these SP2 working conditions in section2.1. Please see the revised manuscript for more details.

+ Potentially useful additional information provided by the ACSM is being ignored by examining only sulfate. Is there a reason for this?

Response> We analyzed the concentration of SO42- measured using the ACSM for the reasons, (1) "its precursor gas (sulfur dioxide) shares the emission sources and locations with CO", and (2) "its formation process in the aqueous phase reaction is useful for analyzing the effect of a possible cloud processing through air parcel transport". We added more explanations especially on (2) to section 2.1.

References Kanaya, Y., X. Pan, T. Miyakawa, Y. Komazaki, F. Taketani, I. Uno, and Y. Kondo (2016), Long-term observations of black carbon mass concentrations at Fukue Island, western Japan, during 2009-2015: Constraining wet removal rates and emission strengths from East Asia, Atmos. Phys. Chem., 16, 10689-10705, doi:10.5194/acp-16-10689-2016, 2016

Please also note the supplement to this comment:
http://www.atmos-chem-phys-discuss.net/acp-2016-570/acp-2016-570-AC2-supplement.pdf

---

## Author Response (AR1)

**Responses to the reviewers' comment**

**Anonymous Referee #1**

**Review of "Alteration of the microphysical properties of black carbon through transport in the boundary layer in East Asia" by Takuma Miyakawa et al. submitted to Atmospheric Chemistry and Physics.**

We appreciate the reviewer's helpful and constructive comments on the manuscript entitled "Alteration of the microphysical properties of black carbon through transport in the boundary layer in East Asia". As the reviewers suggested, we have modified the manuscript. Major points for the revisions are listed as follows.

1) Title has been changed.

2) Supporting information (SI) has been prepared.

3) We have modified the discussion section.

*Note the reviewers' comments in **bold**.

**The manuscript discusses ground-based measurements, with several instruments, of black carbon (BC) near an industrial source region and at a location removed from the source to study the effects of precipitation on the size distribution and properties of the BC-containing particles. The manuscript is well written and competently explains the study, but several of the arguments do not seem supported by the data. If the comments below are addressed I would recommend that the manuscript be accepted for publication. The title refers to "microphysical properties," which is true, but perhaps "size distribution and amount of associated non-BC material" would be more accurate, as the former term implies a host of properties that were not addressed.**

>As the reviewer suggested, this study has investigated a part of the microphysical parameters of BC. Shape and chemical composition of BC-containing particles, which were not directly measured in this study, are important for considering the climatic impacts of BC-containing particles. However, chemical composition of non-refractory (non-BC) materials for both BC-free and -containing particles was measured using an Aerosol Chemical Speciation Monitor (ACSM). We addressed just simply the mixing state of BC-containing particles, and therefore revised the title slightly to "Alteration of the size distributions and mixing states of black carbon through transport in the boundary layer in East Asia".

**Line 56: The sweeping statement that "washout cannot substantially affect the**

**lifetime of atmospheric BC-containing particles," even with a reference to Seinfeld and Pandis, seems difficult to justify. Do the authors mean that because most of the BC-containing particles have diameters of several hundred nanometers, their ability to be scavenged by falling precipitation is not very large? This would seem to depend on the intensity of precipitation.**

>As the reviewer suggested, the accumulation mode aerosols including BC are not effectively removed by the falling rain droplets.   Washout process is dependent on the precipitation intensity (PI) and rain drop size as well as the particle size range.   In this study, the information of rain drop size is not available.   The average PIs along a backward trajectory were calculated for the rain period in 3d-backward time (PI > 0 mm h$^{-1}$).   They ranged from 0.1 to 2.5 mm h$^{-1}$ (median = ~0.6 mm h$^{-1}$).   Using the PI value of 0.6 mm h$^{-1}$, the scavenging rates of accumulation mode particles were estimated to be 6E-3 h$^{-1}$ (6E-5 h$^{-1}$) with the assumed rain drop diameter of 0.2 mm (2 mm).   The corresponding time constants are around 7 and 694 days.   These are longer than the typical transport time from the continent to the observation site.   The details are described in SI.

**Line 148: Rather than "lower and upper boundaries" it would be preferable to state "outside the diameter range . . ." so that it is clear what size is being referred to.**

>We have revised as suggested.

**Lines 152-154: Some discussion of why the EC and rBC concentrations differ, and especially why the rBC concentration is less, seems to be necessary. Line 168: Some justification for the selection of 0.5 as the collection efficiency for sulfate in the ACSM is required.**

>In this study, we compared rBC with effective BC (EBC) measured using a light absorption technique (COSMOS).   As we stated in the original manuscript, the difference between rBC and EBC is within the uncertainties related to both measurements.   One of the unclear uncertainties, which have not well been studied, is the detection sensitivity of SP2 to the ambient rBC particles (incandescence signal intensity per rBC particle mass, $S_{LII}$-$m_{pp}$) in a remote atmosphere.   It was found in previous studies (Moteki and Kondo, 2010; Miyakawa et al. 2016) that the $S_{LII}$-$m_{pp}$ relationship of fullerene soot (FS) particles, which is used as a calibration standard for the SP2, is similar to that of ambient rBC particles in urban/industrial area.   We hence assume the same sensitivity of SP2 to the ambient rBC in a remote atmosphere as that of FS particles and rBC particles in urban/industrial area.   I added some explanations on the related uncertainties to the section 2.1 in the revised manuscript.

The collection efficiency of ACSM-SO$_4^{2-}$ was derived from Yoshino et al. (2016). This paper is included in the reference list of the revised manuscript.

**Line 206: Some discussion of how sensitive the results are to different choices for the percentile (i.e., does the background value change if concentrations lower than the 10th percentile were averaged?) would be helpful, or better yet, a distribution of the CO concentrations should be shown.**

>When we set 10$^{th}$ percentile of CO mixing ratio as the threshold value, the derived background CO mixing ratio was calculated to be 131 ppb, which is slightly higher than the original value (120 ppb).   We prepared SI including the descriptions on the determination of the background CO mixing ratio.   Please see SI for details.

**Line 277: The statement that the ACSM-SO4 and the IC-SO4 "generally agreed well" is true, but from Fig. 5c there appears to be little variability in either at concurrent times when comparison could be made.**

>The variability in IC-SO$_4^{2-}$ mass concentration was ~9 µg m$^{-3}$ at STP (min - max ~1 - ~10).   Wider range of concentrations (<~20 µg m$^{-3}$) were observed during an intercomparison experiment in Queens/New York (Drewnick et al., 2003).   To the best of our knowledge, the observed range was larger enough to discuss the intercomparison results.   For example, Takegawa et al. (2005) reported the intercomparison results of SO$_4^{2-}$ mass concentration between Aerodyne AMS and PILS-IC.   The range given in their study (< ~7 µg m$^{-3}$) is smaller than ours.

**Line 284: It is not clear why the positive correlation of SO4 and CO suggests that the SO4 was secondary and that SO4 contributed to the BC coatings; more explanation of these assumptions/conclusions is required.**

>Growth of BC-containing particles should be explained separately from the formation. Besides our observation results, previous studies support the description of formation and structure of the coating of BC in the original manuscript.   As the reviewer suggested, we revised the related sentences and included more explanation in the revised manuscript.

**Line 290: The authors note "the small variability of SO4/CO ratios," yet Figure 6b shows that these ratios vary considerably.**

>As the reviewer suggested, this statement and Figure 6b seem to contradict each other. We removed this sentence for the clarity.

**Lines 294, 297: The two "experiments," which consisted of two brief time periods out of a month of data, were used to justify conclusions regarding flow patterns. While the results are indeed consistent with the arguments made, it seems difficult to justify such conclusions on the basis of one comparison.**

>As the reviewer suggested, the results shown in this study are based on the observation during not-so-long time periods. We agree that it is actually difficult to draw the general conclusions. However, we still believe that this paper shows the significance in the observational studies of the relationship between removal process and the changes in the BC microphysical properties, because the observed meteorological conditions in the spring of 2015 were not special and similar to those with an average year. We added the sentences "The migrating anticyclone and cyclone were observed during this period, which is typically dominant in spring over East Asia (Asai et al., 1988). We here only briefly describe the meteorological fields (wind flow and precipitation) in the following." behind the first sentence in section 3.1, and modified the last sentence in section 3.5 to "As the results from this study are based on observations during a limited length of time, it would be worthwhile to further investigate the possible connections of the variabilities in BC microphysical properties and meteorological conditions in this region to provide useful constraints on more accurate evaluations of climatic impacts of BC-containing particles (Matsui, 2016)". Please see the revised manuscript for details.

**Line 317: The authors refer to the SO4/CO ratio, but does this really refer to the deltaSO4/delta-CO ratio? It was unclear to me here and a number of places elsewhere in the test whether the CO and SO4 values referred to delta-CO and delta-SO4 values or not. For clarity, I would recommend using "delta-" values throughout.**

>We clearly found the lower concentrations of $SO_4^{2-}$ relative to CO for the data with the higher APT in Figure 6b of the original manuscript. Another reason not to include the $\Delta SO_4^{2-}/\Delta CO$ ratio is the uncertainty related to the variability in the background of $SO_4^{2-}$ in East Asia. Although the use of the same data treatment would be clear for the readers, we did not quantitatively analyze the hourly $\Delta SO_4^{2-}$ and $\Delta CO$ values for considering the relative enhancements of $SO_4^{2-}$ to CO in this study. We hence added the sentences to explain why we do not analyze $\Delta$ values in the revised manuscript in section 2.2.

**Lines 317-319: The difference in slopes shown in the inset to Figure 6b doesn't seem sufficiently large, given the scatter of the data, to be significantly different, and certainly not to justify the conclusion that the controlling process is rainout.**
>The rainout lowered the transport efficiency of $SO_4^{2-}$ as well as BC (to CO). However, the cloud process not associated with the precipitation can affect the relative increases of $SO_4^{2-}$ concentration. The major purpose to include this figure is to elucidate the impact of the cloud process on the aqueous-phase formation of $SO_4^{2-}$, and is not to discuss the loss processes. Figure 6b is modified in the revised manuscript to clarify the data points with the APT of zero (no precipitation through the transport). These data points are highlighted by marking using cross markers. Please see the revised Figure 6b for details.

**Line 343: Here and elsewhere the argument is made that aging leads to growth of BC particles, which is well accepted, but such aging can also lead to loss of larger particles through rainout, het size distributions in Figure 7 doesn't show much of a difference between size distributions for air masses with BC loss and those without, and certainly not more of a difference for larger BC particles than for smaller ones. This discrepancy requires explanation.**
>All the size distributions shown in Figure 7 are normalized by the number or mass integrated for the measured size range, which is described in the caption of this figure. The "absolute" size distributions show more differences between with and without BC loss. We modified the size distributions from "normalized" to "absolute" and added a new figure (fig 7c of the revised manuscript) of the relationship between BC peak diameters and $\Delta BC/\Delta CO$ (i.e., degree of the removal of BC). This figure clarifies the significance of the observed changes in the peak diameter. Please see the revised figure for more details.

**Line 345: The statement that "small BC-containing particles were scavenged by larger particles in the coagulation process" is a hypothesis, but stated as truth. It would seem that concentrations are too low for much coagulation over the brief period (a few days), especially for particles that are many tens of nanometers in diameter. Calculations or a simple model would be required to support this hypothesis. Line 353: It would be preferable, and less ambiguous, to rephrase "BC size of 0.2" to "BC diameter of 0.2".**

>In the consideration of the washout process, the removal of small BC-containing particles through the washout is expected to be significant as well as the coagulation process. We hence describe the possibility of both processes in the revised manuscript. We rephrased "BC size of 0.2" to "BC diameter of 0.2".

**Line 368: The discussion focused on transport pathways of particles in the particular region of the study, but I was expecting more discussion on the results, what they mean, and so forth. There seemed to be little relevance to the second paragraph of the discussion.**

>We reorganized the discussion part (section 3.5). We merged and reorganized the first paragraph and the half of the second paragraph into one paragraph. The latter half of the first paragraph of section 3.5 discusses the observed features and its relevance to the finding in previous studies. We consider that the relationship between transport pathways (i.e., processes during transport) and its impact on the aerosol particles is a key and relevant to our observation results. We hence modified the sentences of the third (second in the revised manuscript) paragraph.

**Line 372: The decrease in the peak diameter of the mass size distribution is very small, and within uncertainty.**

>The change in the peak diameter is small, however, significant. Corresponded change in BC mass is ~1 fg/particle. This difference can be resolved by the SP2 and beyond the uncertainty. The variabilities of the peak diameters are summarized in Table 1 in the original and revised manuscript and are smaller than those measured. As we described in the above, we added a new figure to show the tendency of the BC particle diameter as a function of the degree of BC removal (Fig 7c of the revised manuscript).

**Line 373: The statement that the evidence implies selective removal of large BC containing particles is not supported by Figure 7, which shows a very slight difference in the size distribution between "with BC loss" and "without BC loss" but not apparent selective decrease of larger particles. If there were selective removal, I would expect the size distribution to not be lognormal, but to have a deficit on the large side below what a lognormal would be. Figure 3a is very difficult to read; could it be made larger? Figure 3b requires units for q_v to accompany the scale. Figure 4a should be made larger also, if possible. Figure 5b: it is difficult to distinguish the COSMOS and SP2 BC values; perhaps make one**

**red and the other black? Figure 6a: do the axes refer to delta-CO and delta-BC? If so, they should be labeled as such. Figure 6b, inset: what does "all data" refer to? If this is to label the gray dot, then it is not clear.**

>The activation of aerosol particles to cloud droplets has occurred during transport. We did not observe the aerosol particles below the convective cloud, because the migratory cyclone was the dominant process for the upward transport in spring in East Asia. We thus considered that SP2 detected BC-containing particles which have been aged (about a half ~ a day, typical transport time) since affected by the wet removal. The size distributions of BC-containing particles can change during transport again after the rainout process, and therefore do not always conserve the original shape.

We have corrected some figures as suggested. We enlarged all figures as large as possible as suggested. Units of all parameters in Fig 3 were clarified in the modified one. The color of SP2-BC in Fig 5 was changed to red. Axes of Fig 6a do not refer to delta (so we didn't change). Fig 6b was modified, because it was not clear. All the values in Figure 6 are absolute concentrations (not delta). Fig 7c was newly added (Please see the texts for details).

*Note the reviewers' comments in **bold**.

**The authors present a one month case study examining measurements of black carbon properties at a remote island site, using co-located measurements of CO and sub-micron aerosol composition and reanalysis data to evaluate precipitation impacts on the observed properties. The manuscript focuses on contrasting observed properties during periods with differing accumulated precipitation along backward trajectories. The paper is well prepared and well organized and the subject is well within the topic area for ACP. There are several areas where minor revisions are needed, however, before the paper can be recommended for publication. I agree with the points raised by Reviewer #1, so have tried to not repeat too much of what has been already raised. The comments should be addressed in a revised manuscript. In addition:**
**+ Given the focus of the manuscript, the introduction would benefit from a more thorough discussion of the various BC removal mechanisms, with more mechanistic details given as to why various processes may or may not be important in the study area. Distinction should be made between in-cloud processes (nucleation scavenging versus scavenging by pre-existing droplets), below-cloud (washout) and dry deposition.**
We added the sentence describing the removal processes of BC to the second paragraph of section "Introduction".   Relate to this, as the reviewer #1 suggested, we have modified the descriptions on the relative importance of the washout (to the rainout) (in section 3.3 of the original manuscript).   A previous study (Kanaya et al., 2016) is referred for quantitatively elucidating the impacts of dry deposition.   The timescale of the removal through the washout was evaluated in this study to be longer than the typical time scale of the transport (details in S.I. newly-prepared).   Please see the revised manuscript and SI for more details.

**+ Two points regarding reported SP2-measured BC number/mass distributions. First, the manuscript needs to make it more clear when BC core versus shell diameters are being discussed, especially when linking the observations to theory. For example, while it is true we would expect larger particles to be removed in air masses heavily impacted by precipitation, the effects on BC core distributions will be confounded by other material mixed with the cores. Related to this, the diameter range for which the optical sizing of the BC particles should be provided in the methods section. Second, small changes in the detection efficiency of the SP2 at its lower limit due to changes in cavity laser power can look like changes in BC core number distribution. A short statement regarding any checks on cavity laser power or other approaches used to ensure consistent behavior at lower size limits for the instrument would be useful.**

As the reviewer suggested, we added explanations on these SP2 working conditions in section2.1.   Please see the revised manuscript for more details.

**+ Potentially useful additional information provided by the ACSM is being ignored by examining only sulfate. Is there a reason for this?**

>We analyzed the concentration of $SO_4^{2-}$ measured using the ACSM for the reasons, (1) "its precursor gas (sulfur dioxide) shares the emission sources and locations with CO", and (2) "its formation process in the aqueous phase reaction is useful for analyzing the effect of a possible cloud processing through air parcel transport".   We added more explanations especially on (2) to section 2.1.

References

[revised manuscript text omitted]

The removal timescale of submicron accumulation particles through the washout was estimated to be around a week or longer, which is much longer than the typical transcript time of air masses exported from the East Asian continent.   More details about the estimation are given in the S.I.   Note that the Aitken and coarse mode particles, which were not measured by the SP2, can be significantly affected.

The $SO_4^{2-}$/CO slopes with the APT values of zero (i.e., non-precipitation) were analyzed as a function of $RH_{max}$ (**Figure 6b**).   The $SO_4^{2-}$/CO slopes without precipitation varied from 30.7 to 44.1 ng m$^{-3}$ ppb$^{-1}$ under the conditions without ($RH_{max}$ <50%) and with ($RH_{max}$ >80%) cloud impacts, respectively.

[revised manuscript text omitted]

*Time used for calculating the averaged statistics of the microphysical properties of BC-containing particles.

---

## Author Response (ED1)

**Responses to the reviewers' comment**

**Anonymous Referee #1**

**Review of "Alteration of the microphysical properties of black carbon through transport in the boundary layer in East Asia" by Takuma Miyakawa et al. submitted to Atmospheric Chemistry and Physics.**

We appreciate the reviewer's helpful and constructive comments on the manuscript entitled "Alteration of the microphysical properties of black carbon through transport in the boundary layer in East Asia".   As the reviewers suggested, we have modified the manuscript.   Major points for the revisions are listed as follows.

1) Title has been changed.

2) Relative importance of washout and rainout has been quantitatively discussed in a section that we newly produced (section 3.2 in the revised manuscript).

3) We have added a new section (section 3.5 in the revised manuscript) focusing the changes in chemical compositions of fine aerosols measured using an Aerosol Chemical Speciation Monitor.

4) We have modified the discussion section especially to clarify our speculations based on the observed results.

5) We have modified the size of figures for visible clarity.

*Note the reviewers' comments in **bold**.

**The manuscript discusses ground-based measurements, with several instruments, of black carbon (BC) near an industrial source region and at a location removed from the source to study the effects of precipitation on the size distribution and properties of the BC-containing particles. The manuscript is well written and competently explains the study, but several of the arguments do not seem supported by the data. If the comments below are addressed I would recommend that the manuscript be accepted for publication. The title refers to "microphysical properties," which is true, but perhaps "size distribution and amount of associated non-BC material" would be more accurate, as the former term implies a host of properties that were not addressed.**

>As the reviewer suggested, this study has investigated a part of the microphysical parameters of BC.   Shape and chemical composition of BC-containing particles, which were not directly measured in this study, are important for considering the climatic impacts of BC-containing particles.   However, chemical composition of non-refractory (non-BC) materials for both BC-free and -containing particles was measured using an

Aerosol Chemical Speciation Monitor (ACSM).   We addressed just simply the mixing state of BC-containing particles, and therefore revised the title slightly to "Alteration of the size distributions and mixing states of black carbon through transport in the boundary layer in East Asia".

**Line 56: The sweeping statement that "washout cannot substantially affect the lifetime of atmospheric BC-containing particles," even with a reference to Seinfeld and Pandis, seems difficult to justify. Do the authors mean that because most of the BC-containing particles have diameters of several hundred nanometers, their ability to be scavenged by falling precipitation is not very large? This would seem to depend on the intensity of precipitation.**

>As the reviewer suggested, the accumulation mode aerosols including BC are not effectively removed by the falling rain droplets.   Washout process is dependent on the precipitation intensity (PI) and raindrop size as well as the particle size range.   Using a parameterization (Wang et al., GMD, 2014) including the raindrop size information, we estimated the removal rate of aerosol particles via below-cloud-scavenging.   The precipitation intensity along trajectories and the parameterization suggests that the removal rate is estimated to be $1 \times 10^{-3}$ h$^{-1}$ on average, and be ranging from $0.5 \times 10^{-3}$ to $2 \times 10^{-3}$ h$^{-1}$ in the submicron size range.   The temporal duration in rain along the trajectory was also calculated.   The combination of their estimations enables us to estimate the fraction of the accumulation mode particles removed through the rainout. The fraction removed was estimated to be only 1.0% on average (+2.59%/-0.9%).   The rainout process is a major process to reduce the loss of aerosols in wet removal.

We added a new section to describe the above explanations (section 3.2 in the revised manuscript) as follows.

"3.2 Removal processes of fine aerosol particles

   In this study, the removal processes including dry deposition and washout were considered to be minor.   The dry deposition in this region has already been evaluated by Kanaya et al. (2016).   The washout is dependent on the precipitation intensity and rain drop size as well as the particle size range.   We quantitatively investigated the relative importance of rainout to washout in this study.   The removal rates of submicron accumulation mode particles through the washout ($\Lambda_{accum}$) was estimated to be ~$1 \times 10^{-3}$ h$^{-1}$ ($0.5$-$2 \times 10^{-3}$ h$^{-1}$) using a parametrization given by Wang et al. (2014) and the average precipitation intensity along the trajectories ($0.78 \pm 0.6$ mm h$^{-1}$) as an input to the parameterization.   The possible uncertainties in this estimation are derived from the discrepancies in $\Lambda_{accum}$ the removal rates between the parameterization and some experimental results (Wang et al., 2014). The values of $\Lambda_{accum}$ can be underestimated by an order of magnitude by using the parameterization, which is however overly pessimistic. The temporal duration in rain along trajectories for air masses with the APT greater than 0 mm was 10 ($\pm 8$) hours on average. These values can be used for the estimation of the removed fraction of submicron aerosols through the washout process. The average fraction of submicron aerosols removed was 1% (+2.59%/-0.9%). Even though we took into account the uncertainties for estimating $\Lambda_{accum}$, it was found that the washout process did not play a major role in the removal of BC in East Asian outflow."

**Line 148: Rather than "lower and upper boundaries" it would be preferable to state "outside the diameter range . . ." so that it is clear what size is being referred to.**
>We have revised as suggested.

**Lines 152-154: Some discussion of why the EC and rBC concentrations differ, and especially why the rBC concentration is less, seems to be necessary. Line 168: Some justification for the selection of 0.5 as the collection efficiency for sulfate in the ACSM is required.**
>In this study, we compared rBC with effective BC (EBC) measured using a light absorption technique (COSMOS). As we stated in the original manuscript, the difference between rBC and EBC is within the uncertainties related to both measurements. One of the unclear uncertainties, which have not well been studied, is the detection sensitivity of SP2 to the ambient rBC particles (incandescence signal intensity per rBC particle mass, $S_{LII}$-$m_{pp}$) in a remote atmosphere. It was found in previous studies (Moteki and Kondo, 2011; Miyakawa et al. 2016) that the $S_{LII}$-$m_{pp}$ relationship of fullerene soot (FS) particles, which is used as a calibration standard for the SP2, is similar to that of ambient rBC particles in urban/industrial area. We hence assume the same sensitivity of SP2 to the ambient rBC in a remote atmosphere as that of FS particles and rBC particles in urban/industrial area.

We inserted the sentences in the second paragraph of section 2.1 as follows.
"Fullerene soot (FS, stock 40971, lot L20W054, Alfa Aesar, USA) particles were used as a calibration standard for the SP2. A differential mobility analyzer (Model 3081, TSI Inc., USA) was used for preparing the monodisperse FS particles."

We also added the sentences in the second paragraph of section 2.1 as follows.

"While the validity of the calibration standard, FS particles, has been evaluated only near source regions (Moteki and Kondo, 2011; Miyakawa et al., 2016), the discrepancy can be partly attributed to the differences in physicochemical properties between ambient BC in remote air and FS particles."

The collection efficiency of ACSM-$SO_4^{2-}$ was derived from Yoshino et al. (2016). This study is referred in the revised manuscript.

**Line 206: Some discussion of how sensitive the results are to different choices for the percentile (i.e., does the background value change if concentrations lower than the 10th percentile were averaged?) would be helpful, or better yet, a distribution of the CO concentrations should be shown.**

>When we set $10^{th}$ percentile of CO mixing ratio as the threshold value, the derived background CO mixing ratio was calculated to be 131 ppb, which is very slightly higher than the original value (120 ppb).   We have prepared SI including the descriptions on the determination of the background CO mixing ratio as follows.

"S1. Determination of the background mixing ratio of carbon monoxide (CO)

   We assume the 5th percentile value of CO mixing ratio (138 ppb) as a threshold value to extract its background level ($CO_{bg}$).   $CO_{bg}$ is defined as the average of CO mixing ratios below the 5th percentile in this study, and is calculated to be 120 ppb.   When we change the threshold from 5th to 10th percentiles (146 ppb), derived $CO_{bg}$ increases from 120 ppb to 131 ppb.   Figure S1 depicts the probability density function of the observed CO mixing ratio with the assumed threshold.   It is suggested that the assumption of the threshold value very slightly affected the estimation of $CO_{bg}$.

[Figure]

Figure S1. Probability density of measured CO mixing ratio (shaded bars). Red and blue vertical lines correspond to the 5th and 10th percentile values of the observed CO mixing ratios."

**Line 277: The statement that the ACSM-SO4 and the IC-SO4 "generally agreed well" is true, but from Fig. 5c there appears to be little variability in either at concurrent times when comparison could be made.**

>The variability in IC-$SO_4^{2-}$ mass concentration was ~9 μg m$^{-3}$ at STP (min - max ~1 - ~10). Wider range of concentrations (<~20 μg m$^{-3}$) were observed during an intercomparison experiment in Queens/New York (Drewnick et al., 2003). To the best of our knowledge, the observed range was larger enough to discuss the intercomparison results. For example, Takegawa et al. (2005) reported the intercomparison results of $SO_4^{2-}$ mass concentration between Aerodyne AMS and PILS-IC. The range given in their study (< ~7 μg m$^{-3}$) is smaller than ours.

**Line 284: It is not clear why the positive correlation of SO4 and CO suggests that the SO4 was secondary and that SO4 contributed to the BC coatings; more explanation of these assumptions/conclusions is required.**

>Air masses are well mixed and diluted through transport before sampling in outflow regions. The effects of differences in the source types can be cancelled by the transport process when the spatial distributions are similar. Anthropogenic SO4, which is abundant in this region, is produced from SO2 oxidation in atmosphere. SO2 does not always share the emission sources with CO, because power generation sector has a great contribution to SO2 emission but not to CO.   Actually, the spatial distribution of SO2 emissions in East Asia is similar to that of CO emissions (Koike et al., JGR, 2003; Kurokawa et al., ACP, 2013).   For clarifying this fact, we referred in the revised manuscript the previous studies where CO is used as a tracer to investigate the transport and transformation of sulfur compounds in East Asian region (Koike et al., JGR, 2003; Sahu et al., JGR, 2009).   We added more explanations on this point in section 2.2 in the revised manuscript as follows

"Relative changes in $SO_4^{2-}$ to CO were also analyzed using the linear regression slopes of their correlation in this study.   We did not calculate of their hourly values, because it was difficult to determine the background concentration of $SO_4^{2-}$.   The ues of CO as a tracer of sulfur compounds in East Asia was validated by Koike et al. (2003). Although sulfur dioxide ($SO_2$), which is a major precursor of anthropogenic $SO_4^{2-}$, does not always share the emission sources with CO, the special distributions of $SO_2$ emissions is similar to those of CO emissions in East Asia (Koike et al., 2003; Kurokawa et al., 2013).   Analyzing the increase or decrease in the slopes of the $SO_4^{2-}$-CO correlation is beneficial to the investigation of the formation and removal processes for $SO_4^{2-}$.   Especially, the aqueous-phase reaction of $SO_4^{2-}$ in clouds is discussed using this parameter."

Growth of BC-containing particles should be explained separately because the coating material was not directly measured in this study.   The ACSM measurements supported the interpretation of chemical composition of non-BC components.   It is found that the major components of non-BC materials were ammonium sulfate and organic matter (OM) as summarized in Table 1 newly added in the revised manuscript.   We suggested that the major coating materials of BC were ammonium sulfate and OM.   Besides our observation results, a previous study (Takami et al., JGR., 2013) supports our suggestions.

"Table 1. Mean chemical composition of fine aerosols during the observation period

| Componnents | Period average | APT | | | |
| --- | --- | --- | --- | --- | --- |
| | | 0 mm | 0 mm RH$_{max}$ <50% | 0 mm RH$_{max}$ >80% | >15 mm |
| Ammonium sulfate | 44.9% | 41.8% | 34.0% | 48.9% | 50.4% |
| Ammonium nitrate | 11.7% | 15.7% | 10.7% | 8.0% | 5.0% |
| OM | 40.9% | 40.1% | 52.0% | 40.4% | 42.0% |
| BC | 2.5% | 2.4% | 3.2% | 2.6% | 2.5% |

"

**Line 290: The authors note "the small variability of SO4/CO ratios," yet Figure 6b shows that these ratios vary considerably.**
>As the reviewer suggested, this statement and Figure 6b seem to contradict each other. We removed this sentence for the clarity.

**Lines 294, 297: The two "experiments," which consisted of two brief time periods out of a month of data, were used to justify conclusions regarding flow patterns. While the results are indeed consistent with the arguments made, it seems difficult to justify such conclusions on the basis of one comparison.**
>As the reviewer suggested, the results shown in this study are based on the observation during not-so-long time periods. We agree that it is actually difficult to draw the general conclusions. However, we still believe that this paper shows the significance in the observational studies of the relationship between removal process and the changes in the BC microphysical properties, because the observed meteorological conditions in the spring of 2015 were not special and similar to those with an average year.
We added the sentences as follows.
"The migrating anticyclone and cyclone were observed during this period, which is typically dominant in spring over East Asia (Asai et al., 1988). We here only briefly describe the meteorological fields (wind flow and precipitation) in the following."
(behind the first sentence in section 3.1)

We modified the last sentence in section 3.5 to
"As the results from this study are based on observations during a limited length of time, it would be worthwhile to further investigate the possible connections of the variabilities in BC microphysical properties and meteorological conditions in this region to provide useful constraints on more accurate evaluations of climatic impacts of BC-containing particles (Matsui, 2016)".

**Line 317: The authors refer to the SO4/CO ratio, but does this really refer to the deltaSO4/delta-CO ratio? It was unclear to me here and a number of places elsewhere in the test whether the CO and SO4 values referred to delta-CO and delta-SO4 values or not. For clarity, I would recommend using "delta-" values throughout.**

>We clearly found the lower concentrations of $SO_4^{2-}$ relative to CO for the data with the higher APT in Figure 6b of the original manuscript.    Another reason not to include the $\Delta SO_4^{2-}/\Delta CO$ ratio is the uncertainty related to the variability in the background of $SO_4^{2-}$ in East Asia.    Although the use of the same data treatment would be clear for the readers, we did not quantitatively analyze the hourly $\Delta SO_4^{2-}$ and $\Delta CO$ values for considering the relative enhancements of $SO_4^{2-}$ to CO in this study.    We hence added the sentences to explain why we do not analyze $\Delta$ values in section 2.2 in the revised manuscript as follows.

"Relative changes in $SO_4^{2-}$ to CO were also analyzed by using the linear regression slopes of their correlation in this study.    We did not calculate the hourly $\Delta SO_4^{2-}/\Delta CO$ values, because it was difficult to determine the background concentration of $SO_4^{2-}$. Analyzing the slope of the $SO_4^{2-}$-CO correlation is beneficial to the investigation of the formation processes as well as the removal processes for $SO_4^{2-}$.    Especially, the aqueous-phase formation of $SO_4^{2-}$ in clouds is discussed by using this parameter."

We modified the section 3.4 in the revised manuscript.    The slopes of $SO_4^{2-}$-CO correlation were more systematically investigated.    We selected three cases.    In the original manuscript, we have already analyzed the data points with the APT of zero and higher and lower $RH_{max}$ (i.e., no precipitation with and without cloud impacts, respectively).    In addition to these cases, we added a case for the data points with the APT >15 mm which represent the data points heavily affected by the wet removal. The linear regression slopes for three cases were added to Figure 6b in the revised manuscript.    It is very clear to investigate the enhancement ratios.

**Lines 317-319: The difference in slopes shown in the inset to Figure 6b doesn't seem sufficiently large, given the scatter of the data, to be significantly different, and certainly not to justify the conclusion that the controlling process is rainout.**
>The rainout lowered the transport efficiency of $SO_4^{2-}$ as well as BC (to CO). However, the cloud process not associated with the precipitation can affect the relative increases of $SO_4^{2-}$ concentration.    The major purpose to include this figure is to elucidate the impact of the cloud process on the aqueous-phase formation of $SO_4^{2-}$, and is not to discuss the loss processes.    Figure 6b is modified in the revised manuscript to clarify the data points with the higher values of APT and with the APT value of zero (no precipitation through the transport).    These data points are analyzed by the linear regression.    Please see the revised Figure 6b for details.    In section 3.3, we added the descriptions on the changes in regression slopes associated depending on the air mass histories.

**Line 343: Here and elsewhere the argument is made that aging leads to growth of BC particles, which is well accepted, but such aging can also lead to loss of larger particles through rainout, het size distributions in Figure 7 doesn't show much of a difference between size distributions for air masses with BC loss and those without, and certainly not more of a difference for larger BC particles than for smaller ones. This discrepancy requires explanation.**

>All the size distributions shown in Figure 7 are normalized by the number or mass integrated for the measured size range, which is described in the caption of this figure. The "absolute" size distributions show more differences between with and without BC loss. We modified the size distributions from "normalized" to "absolute" and added a new figure (fig 7c of the revised manuscript) of the relationship between BC peak diameters and $\Delta BC/\Delta CO$ (i.e., degree of the removal of BC). Please see the revised figure for more details. The air mass mixing in the PBL as well as partial experience of can also change the shape of particle size distributions. Furthermore, we could not perform quantitative evaluations for these effects. We believe that these complicated processes can be evaluated by a model study. We added the sentences to the last part of the first paragraph in "Discussion" section (section 3.7 in the revised manuscript) as follows.

"The coagulation of aerosols particles through the transport after the wet removal events can lead to the modification of the particle size and mixing state distributions affected by cloud processes. The suppression of changes in the microphysical properties of BC-containing particles during transport in the PBL can be related to these factors. More quantitative assessments of the impacts of these factors should be performed using a model which has a function to resolve the mixing state of aerosol particles (e.g., Matsui et al., 2013)."

**Line 345: The statement that "small BC-containing particles were scavenged by larger particles in the coagulation process" is a hypothesis, but stated as truth. It would seem that concentrations are too low for much coagulation over the brief period (a few days), especially for particles that are many tens of nanometers in diameter. Calculations or a simple model would be required to support this hypothesis. Line 353: It would be preferable, and less ambiguous, to rephrase "BC**

**size of 0.2" to "BC diameter of 0.2".**

>In the consideration of the washout process, the removal of small BC-containing particles through the washout is expected to be significant as well as the coagulation process. We hence describe the possibility of both processes in the revised manuscript. We rephrased "BC size of 0.2" to "BC diameter of 0.2".

**Line 368: The discussion focused on transport pathways of particles in the particular region of the study, but I was expecting more discussion on the results, what they mean, and so forth. There seemed to be little relevance to the second paragraph of the discussion.**

>We reorganized the discussion part (section 3.7 in the revised manuscript). We merged and reorganized the first paragraph and the half of the second paragraph into one paragraph. We added the explanations to interpret the observed results and to show the limitation at this moment as follows.

"The coagulation of aerosols particles through the transport after the wet removal events can lead to the modification of the particle size and mixing state distributions affected by cloud processes. The suppression of changes in the microphysical properties of BC-containing particles during transport in the PBL can be related to these factors. More quantitative assessments of the impacts of these factors should be performed using a model which has a function to resolve the mixing state of aerosol particles (e.g., Matsui et al., 2013)."

We consider that the relationship between transport pathways (i.e., processes during transport) and its impact on the aerosol particles is a key and relevant to our observation results. We hence did not removed this part and modified the sentences of the third (second in the revised manuscript) paragraph.

**Line 372: The decrease in the peak diameter of the mass size distribution is very small, and within uncertainty.**

>The change in the peak diameter in Fig 7b is small (corresponded change in BC mass is 1 fg/particle). As we described in the above, we added a new figure to show the tendency of the BC particle diameter as a function of the degree of BC removal (Fig 7c of the revised manuscript). Fig 7c indicates 2-2.5 fg/particle decrease from the higher ($\sim$6 ng m$^{-3}$ ppb$^{-1}$) to lower values (0.4-0.5 ng m$^{-3}$ ppb$^{-1}$) of $\Delta$BC/$\Delta$CO. This difference shown in Figure 7c can be resolved by the SP2 (beyond the uncertainty as described in section 2.1 of the revised manuscript.). The variabilities of the peak diameters are summarized in Table 2 in the revised manuscript and are smaller than those measured.

**Line 373: The statement that the evidence implies selective removal of large BC containing particles is not supported by Figure 7, which shows a very slight difference in the size distribution between "with BC loss" and "without BC loss" but not apparent selective decrease of larger particles. If there were selective removal, I would expect the size distribution to not be lognormal, but to have a deficit on the large side below what a lognormal would be. Figure 3a is very difficult to read; could it be made larger? Figure 3b requires units for q_v to accompany the scale. Figure 4a should be made larger also, if possible. Figure 5b: it is difficult to distinguish the COSMOS and SP2 BC values; perhaps make one red and the other black? Figure 6a: do the axes refer to delta-CO and delta-BC? If so, they should be labeled as such. Figure 6b, inset: what does "all data" refer to? If this is to label the gray dot, then it is not clear.**

>The activation of aerosol particles to cloud droplets has occurred during transport. We did not observe the aerosol particles below the convective cloud, because the migratory cyclone was the dominant process for the upward transport in spring in East Asia. We thus considered that SP2 detected BC-containing particles which have been aged (about a half ~ a day, typical transport time) since affected by the wet removal. The size distributions of BC-containing particles can change during transport again after the rainout process, and therefore do not always conserve the original shape.

We have corrected some figures as suggested. We enlarged all figures as large as possible as suggested. Units of all parameters in Fig 3 were clarified in the modified one. The color of SP2-BC in Fig 5 was changed to red. Axes of Fig 6a do not refer to delta (so we didn't change). Fig 6b was modified, because it was not clear. All the values in Figure 6 are absolute concentrations (not delta). Fig 7c was newly added (Please see the texts for details).

*Note the reviewers' comments in **bold**.

**The authors present a one month case study examining measurements of black carbon properties at a remote island site, using co-located measurements of CO and sub-micron aerosol composition and reanalysis data to evaluate precipitation impacts on the observed properties. The manuscript focuses on contrasting observed properties during periods with differing accumulated precipitation along backward trajectories. The paper is well prepared and well organized and the subject is well within the topic area for ACP. There are several areas where minor revisions are needed, however, before the paper can be recommended for publication. I agree with the points raised by Reviewer #1, so have tried to not repeat too much of what has been already raised. The comments should be addressed in a revised manuscript. In addition:**

**+ Given the focus of the manuscript, the introduction would benefit from a more thorough discussion of the various BC removal mechanisms, with more mechanistic details given as to why various processes may or may not be important in the study area. Distinction should be made between in-cloud processes (nucleation scavenging versus scavenging by pre-existing droplets), below-cloud (washout) and dry deposition.**

We added the sentence describing the removal processes of BC to the second paragraph of section "Introduction".   Relate to this, as the reviewer #1 suggested, we have modified the descriptions on the relative importance of the washout (to the rainout) (in section 3.3 of the original manuscript).   We made a new section for the explanation as follows.

"3.2. Removal processes of fine aerosol particles
In this study, the removal processes including dry deposition and washout were considered to be minor.   The dry deposition in this region has already been evaluated by Kanaya et al. (2016).   The washout is dependent on the precipitation intensity and rain drop size as well as the particle size range.   We quantitatively investigated the relative importance of rainout to washout in this study.   The removal rates of submicron accumulation mode particles through the washout was estimated to be $\sim 1 \times 10^{-3}$ h$^{-1}$ ($0.5$-$2 \times 10^{-3}$ h$^{-1}$) using a parametrization given by Wang et al. (2014) and the average precipitation intensity along the trajectories ($0.78 \pm 0.6$ mm h$^{-1}$) as an input to the parameterization.   The temporal duration in rain along trajectories for air masses with the APT greater than 0 mm was 10 ($\pm 8$) hours on average.   These values can be used for the estimation of the removed fraction of submicron aerosols through the washout process.   The average fraction of submicron aerosols removed was 1% (+2.59%/-0.9%), indicating that the washout process played a minor role in the removal of BC in East Asian outflow."

**+ Two points regarding reported SP2-measured BC number/mass distributions. First, the manuscript needs to make it more clear when BC core versus shell diameters are being discussed, especially when linking the observations to theory. For example, while it is true we would expect larger particles to be removed in air masses heavily impacted by precipitation, the effects on BC core distributions will be confounded by other material mixed with the cores. Related to this, the diameter range for which the optical sizing of the BC particles should be provided in the methods section. Second, small changes in the detection efficiency of the SP2 at its lower limit due to changes in cavity laser power can look like changes in BC core number distribution. A short statement regarding any checks on cavity laser power or other approaches used to ensure consistent behavior at lower size limits for the instrument would be useful.**

As the reviewer suggested, we added explanations on these SP2 data analyses and working conditions in section 2.1 as follows.

● Estimation of shell to core diameter ratios of BC-containing particles

We added the following sentences to describe the diameter range of BC-containing particles.

"In this study, we analyzed the $D_S$ of BC-containing particles with a $D_{core}$ range between 0.15 and 0.35 μm. The maximum value of $D_S/D_{core}$ ratios analyzed is 4 in this study. Retrieved results suggest that almost all BC-containing particles were not so thickly coated (for example, $D_S/D_{core}$ ratios of 2.5 at highest at $D_{core}$ of 0.2 μm)."

● Laser power and lower BC diameter limit

We added the following sentence to describe that the variations in the housekeeping parameters of SP2 cannot change the main conclusions in this study.

"The variations in the laser power were within ±3% during the observation period, thus indicating that the fluctuations of laser power did not largely affect the lower limit of the detectable BC size of the SP2."

**+ Potentially useful additional information provided by the ACSM is being ignored by examining only sulfate. Is there a reason for this?**

>We analyzed the concentration of $SO_4^{2-}$ measured using the ACSM for the reasons, (1) "its precursor gas (sulfur dioxide) shares the emission sources and locations with CO", and (2) "its formation process in the aqueous phase reaction is useful for analyzing the effect of a possible cloud processing through air parcel transport". We analyzed the chemical composition of fine aerosols measured using the ACSM and made a new section (section 3.5 in the revised manuscript). To show the results for the analyses, we made a new Table (Table 1 in the revised manuscript.). Figure 5 was modified by adding the temporal variations in the mass concentrations of nitrate and organic matter (OM). Sulfate and OM were the major components for fine aerosol particles in this study.

"Table 1. Mean chemical composition of fine aerosols during the observation period

| Componnents | Period average | APT | | | |
| --- | --- | --- | --- | --- | --- |
| | | 0 mm | 0 mm RH$_{max}$ <50% | 0 mm RH$_{max}$ >80% | >15 mm |
| Ammonium sulfate | 44.9% | 41.8% | 34.0% | 48.9% | 50.4% |
| Ammonium nitrate | 11.7% | 15.7% | 10.7% | 8.0% | 5.0% |
| OM | 40.9% | 40.1% | 52.0% | 40.4% | 42.0% |
| BC | 2.5% | 2.4% | 3.2% | 2.6% | 2.5% |

"

"3.5. Changes in fine aerosol compositions

Chemical compositions of fine aerosols were investigated in terms of the APT and $RH_{max}$. Four cases are selected here, namely (1) APT of zero (no precipitation), (2) APT of zero with $RH_{max}$ <50% (no precipitation without cloud impacts), (3) APT of zero with $RH_{max}$ >80% (no precipitation with cloud impacts), and (4) APT >15 mm (heavily affected by wet removal). The results are summarized in **Table 1**. Ammonium sulfate and OM were dominant in all cases. The relative contributions of ammonium sulfate in the cases (3) and (4) increased from the average, indicating that cloud processes affected the relative abundance of ammonium sulfate. The contributions of OM in the case (2) increased from the average. The formation of secondary OM can be significant under dry conditions during transport. Detailed mass spectral analyses of OM and formation of OM in cloud are beyond the scope of this study, and they are not discussed in this study. The relative changes in chemical compositions were within around 10%."

**Referee #1 Comment on Line 372 regarding the BC peak diameter from the SP2: You state that a difference in ~1 fg BC can be resolved within the SP2's uncertainties, but do not state these uncertainties. Your response is too vague and requires more specific quantitative details to adequately respond to the Referee.**

>We included the technical descriptions of SP2 performance on the resolving power of incandescence intensity (proportional to mass per particle, mpp) in section 2.1.   The values of the changes in mass per particle were quantitatively evaluated.   1 fg BC around 1 fg of BC can be resolved, however the resolution is dependent on the signal levels.   Figure 7c which is newly added shows the systematic change in peak diameter or mpp as a function of the degree of BC loss.   The observed change in mpp is as large as 2-2.5 fg, which is larger than the uncertainties.

**Referee #1 Comment on Line 373: You have not addressed the main concern raised regarding the supposed evidence for the selective removal of large BC particles, and how this evidence is not clear in the presented size distributions.**

>As discussed in the second paragraph of section 3.6 (3.5 in the original manuscript), turbulent mixing in the PBL leads partial experience of the in-cloud scavenging for aerosol particles suspended within the PBL.   This indicates that a certain fraction of aerosol particles in the PBL does not experience.   The degree of changes in the shape as well as peak diameter of size distributions can be reduced by this effect.   The air mass aging leads to the redistribution of the particle size distribution through the coagulation.   The aging process after the wet removal process as well as the mixing process can qualitatively account for why the evidence is not clear in the observed data sets.   Quantitative assessments of these effects should be performed using a chemical transport model which can resolve the mixing state of aerosol particles (Matsui et al., JGR, 2013).   We included these points in "Discussion" section in the revised manuscript.

**Referee #2 comment on SP2 BC mass/number distributions: The Referee's question**

**regarding the need for more details on when BC core vs. shell diameters are being discussed has not been addressed.**

>In the revised manuscript, (1) the shell and core diameter range, and (2) the effect of laser power fluctuation to the lower limit of measurable diameter of BC-containing particles are clarified in section 2.1.

(1) The range of diameters of BC-core was from 150 to 350 nm.  The maximum value of shell to core (S/C) ratio of these BC-containing particles analyzed was 4.  Retrieved results in this study suggest that almost all particles had no such high S/C ratios (~2.5 at the highest for 200 nm core BC-containing particles).

(2) We diagnosed the housekeeping data to check the stability of the detection efficiency of BC-containing particles.  The variations of the laser power were within 3% throughout the observation period.  It is found that this factor does not largely affect the lower size limit of detection.

**Referee #2 comment on ACSM data: The focus only on the sulfate measurements is not appropriate. You need to expand the results presented to discuss the full set of measurements from the ACSM. Your response regarding this question did not satisfactorily answer the Referee's question. Discussing how SO2 and CO share the same emission sources (which as discussed above is not a great assumption) does not actually explain why you don't present ACSM measurements of other aerosol components such as organics, ammonium, or nitrate. If you are going to assume that sulfate is the only major secondary aerosol component mixed with the BC (line 195), you must justify this assumption based on the measurements available. I would be surprised if there were not significant contributions from other secondary species such as organic carbon. On Page 343 you discuss another study that did observe organic coatings on BC particles in East Asian outflow.**

>As described the above, we included in the revisions the data analyses of chemical composition variations in non-BC materials measured by using the ACSM.  The temporal variations of nitrate, ammonium, and OM are shown in Fig 5 of section 3.2, and the average relative abundance of them is discussed.  Ion balance was investigated to consider the chemical form of inorganic ions.  Sulfate and nitrate were almost fully neutralized by ammonium.  We found that the major components of fine aerosols were ammonium sulfate and organic material. We made a new section (section 3.4 in the revised manuscript) of the discussion on the changes in the composition of aerosols in air masses depending upon their histories.  The contribution of ammonium sulfate increased with the cloud processing through the transport. The contribution of OM is significant in all cases, especially for air masses with no precipitation and no cloud impacts. Relative changes in compositions against the air mass histories were not so large and were around ~10%. The two components, ammonium sulfate and OM, are the most important contributors to the non-BC materials in fine aerosol masses. Detailed speciation of OM based on the mass spectral analyses in East Asian outflow is beyond the scope of this study and has been discussed in previous studies (e.g., Irei et al., EST, 2011). We therefore did not include this topic in the manuscript. We refer a previous paper to present the chemical and structural feature of BC-containing particles in East Asian outflow air masses (Takami et al., JGR, 2013) as a great example of the mixing state of BC at the observation site. The major components, ammonium sulfate and OM, revealed by the ACSM measurements in this study were same as their study.

**Ryan Sullivan**
**ACP Co-Editor**

We again appreciate your efforts. The revisions will improve our manuscripts. We hope our paper contribute the significance of the Atmospheric Chemistry and Physics.

[revised manuscript text omitted]

---

## Author Response (AR3)

**Responses to the reviewers' comment**

Anonymous Referee #1

Review of "Alteration of the microphysical properties of black carbon through transport in the boundary layer in East Asia" by Takuma Miyakawa et al. submitted to Atmospheric Chemistry and Physics.

We appreciate the reviewer's helpful and constructive comments on the manuscript entitled "Alteration of the microphysical properties of black carbon through transport in the boundary layer in East Asia". As the reviewers suggested, we have modified the manuscript. Major points for the revisions are listed as follows.

- 1) Title has been changed.
- 2) Relative importance of washout and rainout has been quantitatively discussed in a section that we newly produced (section 3.2 in the revised manuscript).
- 3) We have added a new section (section 3.5 in the revised manuscript) focusing the changes in chemical compositions of fine aerosols measured using an Aerosol Chemical Speciation Monitor.
- 4) We have modified the discussion section especially to clarify our speculations based on the observed results.
- 5) We have modified the size of figures for visible clarity.

\*Note the reviewers' comments in **bold**.

The manuscript discusses ground-based measurements, with several instruments, of black carbon (BC) near an industrial source region and at a location removed from the source to study the effects of precipitation on the size distribution and properties of the BC-containing particles. The manuscript is well written and competently explains the study, but several of the arguments do not seem supported by the data. If the comments below are addressed I would recommend that the manuscript be accepted for publication. The title refers to "microphysical properties," which is true, but perhaps "size distribution and amount of associated non-BC material" would be more accurate, as the former term implies a host of properties that were not addressed.

>As the reviewer suggested, this study has investigated a part of the microphysical parameters of BC. Shape and chemical composition of BC-containing particles, which were not directly measured in this study, are important for considering the climatic impacts of BC-containing particles. However, chemical composition of non-refractory (non-BC) materials for both BC-free and -containing particles was measured using an

Aerosol Chemical Speciation Monitor (ACSM). We addressed just simply the mixing state of BC-containing particles, and therefore revised the title slightly to "Alteration of the size distributions and mixing states of black carbon through transport in the boundary layer in East Asia".

Line 56: The sweeping statement that "washout cannot substantially affect the lifetime of atmospheric BC-containing particles," even with a reference to Seinfeld and Pandis, seems difficult to justify. Do the authors mean that because most of the BC-containing particles have diameters of several hundred nanometers, their ability to be scavenged by falling precipitation is not very large? This would seem to depend on the intensity of precipitation.

>As the reviewer suggested, the accumulation mode aerosols including BC are not effectively removed by the falling rain droplets. Washout process is dependent on the precipitation intensity (PI) and raindrop size as well as the particle size range. Using a parameterization (Wang et al., GMD, 2014) including the raindrop size information, we estimated the removal rate of aerosol particles via below-cloud-scavenging. The precipitation intensity along trajectories and the parameterization suggests that the removal rate is estimated to be  $1 \times 10^{-3}$  h-1 on average, and be ranging from  $0.5 \times 10^{-3}$  to  $2 \times 10^{-3}$  h-1 in the submicron size range. The temporal duration in rain along the trajectory was also calculated. The combination of their estimations enables us to estimate the fraction of the accumulation mode particles removed through the rainout. The fraction removed was estimated to be only 1.0% on average (+2.59%/-0.9%). The rainout process is a major process to reduce the loss of aerosols in wet removal.

We added a new section to describe the above explanations (section 3.2 in the revised manuscript) as follows.

**"3.2 Removal processes of fine aerosol particles**

In this study, the removal processes including dry deposition and washout were considered to be minor. The dry deposition in this region has already been evaluated by Kanaya et al. (2016). The washout is dependent on the precipitation intensity and rain drop size as well as the particle size range. We quantitatively investigated the relative importance of rainout to washout in this study. The removal rates of submicron accumulation mode particles through the washout ( $\Lambda_{accum}$ ) was estimated to be ~1 × 10-3 h-1 (0.5-2 × 10-3 h-1) using a parametrization given by Wang et al. (2014) and the average precipitation intensity along the trajectories (0.78 ± 0.6 mm h-1) as an input to the parameterization. The possible uncertainties in this estimation are derived

from the discrepancies in  $\Lambda_{accum}$  the removal rates between the parameterization and some experimental results (Wang et al., 2014). The values of  $\Lambda_{accum}$  can be underestimated by an order of magnitude by using the parameterization, which is however overly pessimistic. The temporal duration in rain along trajectories for air masses with the APT greater than 0 mm was 10 (±8) hours on average. These values can be used for the estimation of the removed fraction of submicron aerosols through the washout process. The average fraction of submicron aerosols removed was 1% (+2.59%/-0.9%). Even though we took into account the uncertainties for estimating  $\Lambda_{accum}$ , it was found that the washout process did not play a major role in the removal of BC in East Asian outflow."

Line 148: Rather than "lower and upper boundaries" it would be preferable to state "outside the diameter range . . ." so that it is clear what size is being referred to.

>We have revised as suggested.

**Lines 152-154: Some discussion of why the EC and rBC concentrations differ, and especially why the rBC concentration is less, seems to be necessary. Line 168: Some justification for the selection of 0.5 as the collection efficiency for sulfate in the ACSM is required.**

>In this study, we compared rBC with effective BC (EBC) measured using a light absorption technique (COSMOS). As we stated in the original manuscript, the difference between rBC and EBC is within the uncertainties related to both measurements. One of the unclear uncertainties, which have not well been studied, is the detection sensitivity of SP2 to the ambient rBC particles (incandescence signal intensity per rBC particle mass,  $S_{LII}$ -mpp) in a remote atmosphere. It was found in previous studies (Moteki and Kondo, 2011; Miyakawa et al. 2016) that the  $S_{LII}$ -mpp relationship of fullerene soot (FS) particles, which is used as a calibration standard for the SP2, is similar to that of ambient rBC particles in urban/industrial area. We hence assume the same sensitivity of SP2 to the ambient rBC in a remote atmosphere as that of FS particles and rBC particles in urban/industrial area.

We inserted the sentences in the second paragraph of section 2.1 as follows.

"Fullerene soot (FS, stock 40971, lot L20W054, Alfa Aesar, USA) particles were used as a calibration standard for the SP2. A differential mobility analyzer (Model 3081, TSI Inc., USA) was used for preparing the monodisperse FS particles." We also added the sentences in the second paragraph of section 2.1 as follows.

"While the validity of the calibration standard, FS particles, has been evaluated only near source regions (Moteki and Kondo, 2011; Miyakawa et al., 2016), the discrepancy can be partly attributed to the differences in physicochemical properties between ambient BC in remote air and FS particles."

The collection efficiency of  $ACSM-SO_4^{2-}$  was derived from Yoshino et al. (2016). This study is referred in the revised manuscript.

Line 206: Some discussion of how sensitive the results are to different choices for the percentile (i.e., does the background value change if concentrations lower than the 10th percentile were averaged?) would be helpful, or better yet, a distribution of the CO concentrations should be shown.

>When we set 10th percentile of CO mixing ratio as the threshold value, the derived background CO mixing ratio was calculated to be 131 ppb, which is very slightly higher than the original value (120 ppb). We have prepared SI including the descriptions on the determination of the background CO mixing ratio as follows.

"S1. Determination of the background mixing ratio of carbon monoxide (CO)

We assume the 5th percentile value of CO mixing ratio (138 ppb) as a threshold value to extract its background level ( $CO_{bg}$ ).  $CO_{bg}$  is defined as the average of CO mixing ratios below the 5th percentile in this study, and is calculated to be 120 ppb. When we change the threshold from 5th to 10th percentiles (146 ppb), derived  $CO_{bg}$  increases from 120 ppb to 131 ppb. Figure S1 depicts the probability density function of the observed CO mixing ratio with the assumed threshold. It is suggested that the assumption of the threshold value very slightly affected the estimation of  $CO_{bg}$ .

Figure S1. Probability density of measured CO mixing ratio (shaded bars). Red and blue vertical lines correspond to the 5th and 10th percentile values of the observed CO mixing ratios."

**Line 277: The statement that the ACSM-SO4 and the IC-SO4 "generally agreed well" is true, but from Fig. 5c there appears to be little variability in either at concurrent times when comparison could be made.**

>The variability in IC-SO42- mass concentration was ~9  $\mu$ g m-3 at STP (min - max ~1 - ~10). Wider range of concentrations (<~20  $\mu$ g m-3) were observed during an intercomparison experiment in Queens/New York (Drewnick et al., 2003). To the best of our knowledge, the observed range was larger enough to discuss the intercomparison results. For example, Takegawa et al. (2005) reported the intercomparison results of SO42- mass concentration between Aerodyne AMS and PILS-IC. The range given in their study (< ~7  $\mu$ g m-3) is smaller than ours.

**Line 284: It is not clear why the positive correlation of SO4 and CO suggests that the SO4 was secondary and that SO4 contributed to the BC coatings; more explanation of these assumptions/conclusions is required.**

>Air masses are well mixed and diluted through transport before sampling in outflow regions. The effects of differences in the source types can be cancelled by the transport process when the spatial distributions are similar. Anthropogenic SO4, which is abundant in this region, is produced from SO2 oxidation in atmosphere. SO2 does not always share the emission sources with CO, because power generation sector

has a great contribution to SO2 emission but not to CO. Actually, the spatial distribution of SO2 emissions in East Asia is similar to that of CO emissions (Koike et al., JGR, 2003; Kurokawa et al., ACP, 2013). For clarifying this fact, we referred in the revised manuscript the previous studies where CO is used as a tracer to investigate the transport and transformation of sulfur compounds in East Asian region (Koike et al., JGR, 2003; Sahu et al., JGR, 2009). We added more explanations on this point in section 2.2 in the revised manuscript as follows

"Relative changes in  $SO_4^{2^-}$  to CO were also analyzed using the linear regression slopes of their correlation in this study. We did not calculate of their hourly values, because it was difficult to determine the background concentration of  $SO_4^{2^-}$ . The ues of CO as a tracer of sulfur compounds in East Asia was validated by Koike et al. (2003). Although sulfur dioxide (SO2), which is a major precursor of anthropogenic  $SO_4^{2^-}$ , does not always share the emission sources with CO, the special distributions of SO2 emissions is similar to those of CO emissions in East Asia (Koike et al., 2003; Kurokawa et al., 2013). Analyzing the increase or decrease in the slopes of the  $SO_4^{2^-}$ -CO correlation is beneficial to the investigation of the formation and removal processes for  $SO_4^{2^-}$ . Especially, the aqueous-phase reaction of  $SO_4^{2^-}$  in clouds is discussed using this parameter."

Growth of BC-containing particles should be explained separately because the coating material was not directly measured in this study. The ACSM measurements supported the interpretation of chemical composition of non-BC components. It is found that the major components of non-BC materials were ammonium sulfate and organic matter (OM) as summarized in Table 1 newly added in the revised manuscript. We suggested that the major coating materials of BC were ammonium sulfate and OM. Besides our observation results, a previous study (Takami et al., JGR., 2013) supports our suggestions.

|                  | Period average | APT   |                   |                   |        |
|------------------|----------------|-------|-------------------|-------------------|--------|
| Componnents      |                | 0 mm  | 0 mm              | 0 mm              | >15 mm |
|                  |                |       | $RH_{max} < 50\%$ | $RH_{max} > 80\%$ |        |
| Ammonium sulfate | 44.9%          | 41.8% | 34.0%             | 48.9%             | 50.4%  |
| Ammonium nitrate | 11.7%          | 15.7% | 10.7%             | 8.0%              | 5.0%   |
| OM               | 40.9%          | 40.1% | 52.0%             | 40.4%             | 42.0%  |
| BC               | 2.5%           | 2.4%  | 3.2%              | 2.6%              | 2.5%   |

"Table 1. Mean chemical composition of fine aerosols during the observation period

**Line 290: The authors note "the small variability of SO4/CO ratios," yet Figure 6b shows that these ratios vary considerably.**

>As the reviewer suggested, this statement and Figure 6b seem to contradict each other. We removed this sentence for the clarity.

**Lines 294, 297: The two "experiments," which consisted of two brief time periods out of a month of data, were used to justify conclusions regarding flow patterns. While the results are indeed consistent with the arguments made, it seems difficult to justify such conclusions on the basis of one comparison.**

>As the reviewer suggested, the results shown in this study are based on the observation during not-so-long time periods. We agree that it is actually difficult to draw the general conclusions. However, we still believe that this paper shows the significance in the observational studies of the relationship between removal process and the changes in the BC microphysical properties, because the observed meteorological conditions in the spring of 2015 were not special and similar to those with an average year.

We added the sentences as follows.

"The migrating anticyclone and cyclone were observed during this period, which is typically dominant in spring over East Asia (Asai et al., 1988). We here only briefly describe the meteorological fields (wind flow and precipitation) in the following." (behind the first sentence in section 3.1)

We modified the last sentence in section 3.5 to

"As the results from this study are based on observations during a limited length of time, it would be worthwhile to further investigate the possible connections of the variabilities in BC microphysical properties and meteorological conditions in this region to provide useful constraints on more accurate evaluations of climatic impacts of BC-containing particles (Matsui, 2016)".

Line 317: The authors refer to the SO4/CO ratio, but does this really refer to the deltaSO4/delta-CO ratio? It was unclear to me here and a number of places elsewhere in the test whether the CO and SO4 values referred to delta-CO and delta-SO4 values or not. For clarity, I would recommend using "delta-" values throughout.

>We clearly found the lower concentrations of  $SO_4^{2-}$  relative to CO for the data with the higher APT in Figure 6b of the original manuscript. Another reason not to include the  $\Delta SO_4^{2-}/\Delta CO$  ratio is the uncertainty related to the variability in the background of  $SO_4^{2-}$  in East Asia. Although the use of the same data treatment would be clear for the readers, we did not quantitatively analyze the hourly  $\Delta SO_4^{2-}$  and  $\Delta CO$  values for considering the relative enhancements of  $SO_4^{2-}$  to CO in this study. We hence added the sentences to explain why we do not analyze  $\Delta$  values in section 2.2 in the revised manuscript as follows.

"Relative changes in  $SO_4^{2^-}$  to CO were also analyzed by using the linear regression slopes of their correlation in this study. We did not calculate the hourly  $\Delta SO_4^{2^-}/\Delta CO$ values, because it was difficult to determine the background concentration of  $SO_4^{2^-}$ . Analyzing the slope of the  $SO_4^{2^-}$ -CO correlation is beneficial to the investigation of the formation processes as well as the removal processes for  $SO_4^{2^-}$ . Especially, the aqueous-phase formation of  $SO_4^{2^-}$  in clouds is discussed by using this parameter."

We modified the section 3.4 in the revised manuscript. The slopes of  $SO_4^{2-}$ -CO correlation were more systematically investigated. We selected three cases. In the original manuscript, we have already analyzed the data points with the APT of zero and higher and lower RHmax (i.e., no precipitation with and without cloud impacts, respectively). In addition to these cases, we added a case for the data points with the APT >15 mm which represent the data points heavily affected by the wet removal. The linear regression slopes for three cases were added to Figure 6b in the revised manuscript. It is very clear to investigate the enhancement ratios.

**Lines 317-319: The difference in slopes shown in the inset to Figure 6b doesn't seem sufficiently large, given the scatter of the data, to be significantly different, and certainly not to justify the conclusion that the controlling process is rainout.**

>The rainout lowered the transport efficiency of  $SO_4^{2-}$  as well as BC (to CO). However, the cloud process not associated with the precipitation can affect the relative increases of  $SO_4^{2-}$  concentration. The major purpose to include this figure is to elucidate the impact of the cloud process on the aqueous-phase formation of  $SO_4^{2-}$ , and is not to discuss the loss processes. Figure 6b is modified in the revised manuscript to clarify the data points with the higher values of APT and with the APT value of zero (no precipitation through the transport). These data points are analyzed by the linear regression. Please see the revised Figure 6b for details. In section 3.3, we added the descriptions on the changes in regression slopes associated depending on the air mass histories.

Line 343: Here and elsewhere the argument is made that aging leads to growth of BC particles, which is well accepted, but such aging can also lead to loss of larger particles through rainout, het size distributions in Figure 7 doesn't show much of a difference between size distributions for air masses with BC loss and those without, and certainly not more of a difference for larger BC particles than for smaller ones. This discrepancy requires explanation.

>All the size distributions shown in Figure 7 are normalized by the number or mass integrated for the measured size range, which is described in the caption of this figure. The "absolute" size distributions show more differences between with and without BC loss. We modified the size distributions from "normalized" to "absolute" and added a new figure (fig 7c of the revised manuscript) of the relationship between BC peak diameters and  $\Delta$ BC/ $\Delta$ CO (i.e., degree of the removal of BC). Please see the revised figure for more details. The air mass mixing in the PBL as well as partial experience of can also change the shape of particle size distributions. Furthermore, we could not perform quantitative evaluations for these effects. We believe that these complicated processes can be evaluated by a model study. We added the sentences to the last part of the first paragraph in "Discussion" section (section 3.7 in the revised manuscript) as follows.

"The coagulation of aerosols particles through the transport after the wet removal events can lead to the modification of the particle size and mixing state distributions affected by cloud processes. The suppression of changes in the microphysical properties of BC-containing particles during transport in the PBL can be related to these factors. More quantitative assessments of the impacts of these factors should be performed using a model which has a function to resolve the mixing state of aerosol particles (e.g., Matsui et al., 2013)."

Line 345: The statement that "small BC-containing particles were scavenged by larger particles in the coagulation process" is a hypothesis, but stated as truth. It would seem that concentrations are too low for much coagulation over the brief period (a few days), especially for particles that are many tens of nanometers in diameter. Calculations or a simple model would be required to support this hypothesis. Line 353: It would be preferable, and less ambiguous, to rephrase "BC

**size of 0.2" to "BC diameter of 0.2".**

>In the consideration of the washout process, the removal of small BC-containing particles through the washout is expected to be significant as well as the coagulation process. We hence describe the possibility of both processes in the revised manuscript. We rephrased "BC size of 0.2" to "BC diameter of 0.2".

Line 368: The discussion focused on transport pathways of particles in the particular region of the study, but I was expecting more discussion on the results, what they mean, and so forth. There seemed to be little relevance to the second paragraph of the discussion.

>We reorganized the discussion part (section 3.7 in the revised manuscript). We merged and reorganized the first paragraph and the half of the second paragraph into one paragraph. We added the explanations to interpret the observed results and to show the limitation at this moment as follows.

"The coagulation of aerosols particles through the transport after the wet removal events can lead to the modification of the particle size and mixing state distributions affected by cloud processes. The suppression of changes in the microphysical properties of BC-containing particles during transport in the PBL can be related to these factors. More quantitative assessments of the impacts of these factors should be performed using a model which has a function to resolve the mixing state of aerosol particles (e.g., Matsui et al., 2013)."

We consider that the relationship between transport pathways (i.e., processes during transport) and its impact on the aerosol particles is a key and relevant to our observation results. We hence did not removed this part and modified the sentences of the third (second in the revised manuscript) paragraph.

**Line 372: The decrease in the peak diameter of the mass size distribution is very small, and within uncertainty.**

>The change in the peak diameter in Fig 7b is small (corresponded change in BC mass is 1 fg/particle). As we described in the above, we added a new figure to show the tendency of the BC particle diameter as a function of the degree of BC removal (Fig 7c of the revised manuscript). Fig 7c indicates 2-2.5 fg/particle decrease from the higher (~6 ng m-3 ppb-1) to lower values (0.4-0.5 ng m-3 ppb-1) of  $\Delta$ BC/ $\Delta$ CO. This difference shown in Figure 7c can be resolved by the SP2 (beyond the uncertainty as described in

section 2.1 of the revised manuscript.). The variabilities of the peak diameters are summarized in Table 2 in the revised manuscript and are smaller than those measured.

Line 373: The statement that the evidence implies selective removal of large BC containing particles is not supported by Figure 7, which shows a very slight difference in the size distribution between "with BC loss" and "without BC loss" but not apparent selective decrease of larger particles. If there were selective removal, I would expect the size distribution to not be lognormal, but to have a deficit on the large side below what a lognormal would be. Figure 3a is very difficult to read; could it be made larger? Figure 3b requires units for q\_v to accompany the scale. Figure 4a should be made larger also, if possible. Figure 5b: it is difficult to distinguish the COSMOS and SP2 BC values; perhaps make one red and the other black? Figure 6a: do the axes refer to delta-CO and delta-BC? If so, they should be labeled as such. Figure 6b, inset: what does "all data" refer to? If this is to label the gray dot, then it is not clear.

>The activation of aerosol particles to cloud droplets has occurred during transport. We did not observe the aerosol particles below the convective cloud, because the migratory cyclone was the dominant process for the upward transport in spring in East Asia. We thus considered that SP2 detected BC-containing particles which have been aged (about a half ~ a day, typical transport time) since affected by the wet removal. The size distributions of BC-containing particles can change during transport again after the rainout process, and therefore do not always conserve the original shape.

We have corrected some figures as suggested. We enlarged all figures as large as possible as suggested. Units of all parameters in Fig 3 were clarified in the modified one. The color of SP2-BC in Fig 5 was changed to red. Axes of Fig 6a do not refer to delta (so we didn't change). Fig 6b was modified, because it was not clear. All the values in Figure 6 are absolute concentrations (not delta). Fig 7c was newly added (Please see the texts for details).

\*Note the reviewers' comments in **bold**.

The authors present a one month case study examining measurements of black carbon properties at a remote island site, using co-located measurements of CO and sub-micron aerosol composition and reanalysis data to evaluate precipitation impacts on the observed properties. The manuscript focuses on contrasting observed properties during periods with differing accumulated precipitation along backward trajectories. The paper is well prepared and well organized and the subject is well within the topic area for ACP. There are several areas where minor revisions are needed, however, before the paper can be recommended for publication. I agree with the points raised by Reviewer #1, so have tried to not repeat too much of what has been already raised. The comments should be addressed in a revised manuscript. In addition:

+ Given the focus of the manuscript, the introduction would benefit from a more thorough discussion of the various BC removal mechanisms, with more mechanistic details given as to why various processes may or may not be important in the study area. Distinction should be made between in-cloud processes (nucleation scavenging versus scavenging by pre-existing droplets), below-cloud (washout) and dry deposition.

We added the sentence describing the removal processes of BC to the second

paragraph of section "Introduction". Relate to this, as the reviewer #1 suggested, we have modified the descriptions on the relative importance of the washout (to the rainout) (in section 3.3 of the original manuscript). We made a new section for the explanation as follows.

**"3.2. Removal processes of fine aerosol particles**

In this study, the removal processes including dry deposition and washout were considered to be minor. The dry deposition in this region has already been evaluated by Kanaya et al. (2016). The washout is dependent on the precipitation intensity and rain drop size as well as the particle size range. We quantitatively investigated the relative importance of rainout to washout in this study. The removal rates of submicron accumulation mode particles through the washout was estimated to be ~1 ×  $10^{-3}$  h-1 (0.5-2 ×  $10^{-3}$  h-1) using a parametrization given by Wang et al. (2014) and the average precipitation intensity along the trajectories (0.78 ± 0.6 mm h-1) as an input to the parameterization. The temporal duration in rain along trajectories for air masses with the APT greater than 0 mm was 10 (±8) hours on average. These values can be used for the estimation of the removed fraction of submicron aerosols through the washout process. The average fraction of submicron aerosols removed was 1% (+2.59%/-0.9%), indicating that the washout process played a minor role in the removal of BC in East Asian outflow."

+ Two points regarding reported SP2-measured BC number/mass distributions. First, the manuscript needs to make it more clear when BC core versus shell diameters are being discussed, especially when linking the observations to theory. For example, while it is true we would expect larger particles to be removed in air masses heavily impacted by precipitation, the effects on BC core distributions will be confounded by other material mixed with the cores. Related to this, the diameter range for which the optical sizing of the BC particles should be provided in the methods section. Second, small changes in the detection efficiency of the SP2 at its lower limit due to changes in cavity laser power can look like changes in BC core number distribution. A short statement regarding any checks on cavity laser power or other approaches used to ensure consistent behavior at lower size limits for the instrument would be useful.

As the reviewer suggested, we added explanations on these SP2 data analyses and working conditions in section 2.1 as follows.

• Estimation of shell to core diameter ratios of BC-containing particles

We added the following sentences to describe the diameter range of BC-containing particles.

"In this study, we analyzed the  $D_{\rm S}$  of BC-containing particles with a  $D_{\rm core}$  range between 0.15 and 0.35 µm. The maximum value of  $D_{\rm S}/D_{\rm core}$  ratios analyzed is 4 in this study. Retrieved results suggest that almost all BC-containing particles were not so thickly coated (for example,  $D_{\rm S}/D_{\rm core}$  ratios of 2.5 at highest at  $D_{\rm core}$  of 0.2 µm)."

**• Laser power and lower BC diameter limit**

We added the following sentence to describe that the variations in the housekeeping parameters of SP2 cannot change the main conclusions in this study.

"The variations in the laser power were within  $\pm 3\%$  during the observation period, thus indicating that the fluctuations of laser power did not largely affect the lower limit of the detectable BC size of the SP2."

**+ Potentially useful additional information provided by the ACSM is being ignored by examining only sulfate. Is there a reason for this?**

>We analyzed the concentration of  $SO_4^{2^-}$  measured using the ACSM for the reasons, (1) "its precursor gas (sulfur dioxide) shares the emission sources and locations with CO", and (2) "its formation process in the aqueous phase reaction is useful for analyzing the effect of a possible cloud processing through air parcel transport". We analyzed the chemical composition of fine aerosols measured using the ACSM and made a new section (section 3.5 in the revised manuscript). To show the results for the analyses, we made a new Table (Table 1 in the revised manuscript.). Figure 5 was modified by adding the temporal variations in the mass concentrations of nitrate and organic matter (OM). Sulfate and OM were the major components for fine aerosol particles in this study.

|                  |                | APT   |                   |                        |        |
|------------------|----------------|-------|-------------------|------------------------|--------|
| Componnents      | Period average | 0 mm  | 0 mm              | 0 mm                   | >15 mm |
|                  |                |       | $RH_{max} < 50\%$ | RH max >80% |        |
| Ammonium sulfate | 44.9%          | 41.8% | 34.0%             | 48.9%                  | 50.4%  |
| Ammonium nitrate | 11.7%          | 15.7% | 10.7%             | 8.0%                   | 5.0%   |
| OM               | 40.9%          | 40.1% | 52.0%             | 40.4%                  | 42.0%  |
| BC               | 2.5%           | 2.4%  | 3.2%              | 2.6%                   | 2.5%   |

"Table 1. Mean chemical composition of fine aerosols during the observation period

"

**"3.5. Changes in fine aerosol compositions**

Chemical compositions of fine aerosols were investigated in terms of the APT and  $RH_{max}$ . Four cases are selected here, namely (1) APT of zero (no precipitation), (2) APT of zero with  $RH_{max} <50\%$  (no precipitation without cloud impacts), (3) APT of zero with  $RH_{max} >80\%$  (no precipitation with cloud impacts), and (4) APT >15 mm (heavily affected by wet removal). The results are summarized in **Table 1**. Ammonium sulfate and OM were dominant in all cases. The relative contributions of ammonium sulfate in the cases (3) and (4) increased from the average, indicating that cloud processes affected the relative abundance of ammonium sulfate. The contributions of OM in the case (2) increased from the average. The formation of secondary OM can be significant under dry conditions during transport. Detailed mass spectral analyses of OM and formation of OM in cloud are beyond the scope of this study, and they are not discussed in this study. The relative changes in chemical compositions were within around 10%."

Responses to the co-editor's comments

Takuma Miyakawa, Japan Agency for Marine Earth Science and Technology, Japan (miyakawat@jamstec.go.jp)

Comments to the Author: Dear Authors,

Thank you for submitting your revised manuscript. Unfortunately, you have not adequately addressed the many comments and concerns raised by the Referees in your Response to the Referees, nor have you significantly revised your manuscript to consider the Referee comments. In many cases your response to the Referee does not actually directly address the question being raised. Instead you focus on another aspect loosely related to the question raised by the Referee, instead of directly answering the question. In some other cases, your response to the Referee involved just deleting the section/sentence in your manuscript in question. This does not properly address the question or concern under discussion, and further weakens the quality of the science and strength of your arguments presented in your paper. Some more specific comments follow below.

(Note that I looked at the Track Changes version of the manuscript submitted with the Author's Response when making my decision, and therefore the incorrect version of the manuscript originally submitted and then updated on Nov. 8th did not affect my decision.)

>We appreciate your efforts to suggest the further revisions of our manuscript for the improvement. We have modified the manuscript as follows.

In order to proceed with peer review, you will need to further revise your manuscript to properly address the many important and valuable points and concerns raised by the Referees. Please note that the 2nd Referee echoed the comments raised by Referee #1, highlighting the importance of these comments. As part of your revisions, you must also produce a new more comprehensive Response to the Referees that properly responds to the specific comments being raised. In your point-by-point response please paste the section of the manuscript text that has been changed to address that point, so it is immediately clear how the manuscript has been revised to address that question. Your revised manuscript submitted actually contains only a few substantial revisions. The manuscript must be revised to properly address the major comments raised during peer-review. Be sure to address the specific question that was raised by the Referee, instead of side-stepping the issue by discussing a different but related issue.

In your revisions please refrain from making small numerous changes to the manuscript text, correcting typos and grammar, etc. These distract from focusing on what revisions were made to the actual science, results, and arguments of the paper. It is not appropriate to make so many extensive changes to the manuscript's language and wording while it is undergoing peer review.

>We have ordered a proofreading service upon the resubmission of the revised manuscript, because we are not non-native speakers and need some helps to improve the quality of English-writing. Small changes to the main text were raised in this process. We would like you to understand that we hope to improve the readability of the manuscript by reducing mistakes in typos and grammer. We minimized the corrections to such mistakes. We tried to further revise the manuscript again by focusing the scientific issues raised from the peer review process.

In your Response summary a major revision you listed in your bulleted list was revision of the manuscript's Discussion section, yet this was in fact barely changed. Only the final paragraph was significantly modified, and this did not change the central points raised in the Discussion.

>We considered that the major points previously raised in "Discussion" are important and should not be removed and corrected without any clear reasons. In this revision, we modified the former part of this section which discusses the interpretation of the observed evidences. In the revised manuscript, the degree of changes in the size and mixing state distributions was interpreted with the complex processes in the PBL (compared to the uplifting from the PBL to the FT). Further quantitative assessments of the proposed explanation to the observed changes in the SP2-derived BC microsphysics should be performed using a model, for example, chemical transport model with a module to resolve particle mixing states (e.g., Matsui et al., JGR, 2013).

I agree that the focus only on the sulfate concentrations measured by the ACSM is odd and seems inappropriate. At other points in the paper the importance of organic components mixed with BC are discussed. The ACSM measurements should be presented, so the contributions of all the major species measured at these sites is known (e.g. sulfate, nitrate, organics, others). >Why we focused the data analyses of and discussion on sulfate in the original manuscript is that sulfate is one of the major components of fine aerosols in this region and that it is useful to connect its formation process with cloud process, as we have already insisted. Because we still believe that the discussion on sulfate is valuable and insightful in the main context of this paper, we did not delete this point.

In the revised manuscript, we included the data analyses of chemical composition variations in non-BC materials measured by using the ACSM. In section 3.2, the temporal variations of nitrate, ammonium, and organic material (OM) are shown in Fig 5, and the average relative abundance of them is discussed. Ion balance was investigated to consider the chemical form of inorganic ions. Sulfate and nitrate were almost fully neutralized by ammonium. We found that the major components of fine aerosols were ammonium sulfate and organic material. We made a new section (section 3.4 in the revised manuscript) of the discussion on the changes in the composition of aerosols in air masses depending upon their histories (e.g., w. or w.o. precipitation). The contribution of ammonium sulfate increased with the cloud processing through the transport. The contribution of OM is significant in all cases, especially for air masses with no precipitation and no cloud impacts. Relative changes in compositions against the air mass histories were not so large and were around  $\sim 10\%$ . These two components, ammonium sulfate and OM, are the most important contributors to the non-BC materials in fine aerosol masses. Detailed speciation of OM based on the mass spectral analyses in East Asian outflow is beyond the scope of this study and has been discussed in previous studies (e.g., Irei et al., EST, 2011). Aqueous phase formation of OM is also beyond the scope of this study. We therefore did not include these topics in the manuscript.

The response regarding the proposed small contribution from washout is not clear. This should be discussed more clearly and quantitatively in the main text, with citations to key references, instead of just putting all the details in the SI. Please provide a much more detailed response regarding this calculation and the inherent assumptions and uncertainties in your response to Referee #1. The calculated washout lifetimes seem to be unrealistically long.

>We included the descriptions on the washout process in East Asia in this period in the main text. The reason why the lifetime estimated in the previous revision is long is that we oversimplified the input parameters, especially sizes of rain drops, in calculating the scavenging rates. Using a parameterization (Wang et al., GMD, 2014) including the raindrop size information, we estimated the removal rate of aerosol particles via below-cloud-scavenging. The precipitation intensity (PI) along trajectories and the parameterization suggests that the

removal rate is estimated to be  $1 \times 10^{-3}$  h-1 on average, and be ranging from  $0.5 \times 10^{-3}$  to  $2 \times 10^{-3}$  h-1 (depending upon the PI value) in the submicron size range. The temporal duration in raining along the trajectories was also calculated. The combination of their estimations enables us to estimate the fraction of the accumulation mode particles removed through the rainout. The fraction removed was estimated to be only 1.0% on average (+2.59%/-0.9%). The possible uncertainties raised in this section cannot change the major conclusion that the washout process did not played a major role in the removal of fine aerosol particles.

Referee #1 Comment on Line 284: Your response and revisions do not adequately address the questions raised regarding the appropriateness of using the correlation between CO and SO4 to conclude that the SO4 was secondary and coated the BC particles. While SO4 and its precursors are co-emitted with CO, there are many other possible combustion sources that emit CO but emit much less SO2/SO4. Assuming that most or all of the SO4 and CO measured were co-emitted is not a justifiable assumption.

>Air masses are well mixed and diluted through transport before sampling in outflow regions. The effects of differences in the source types can be cancelled by these processes when the emission spatial distributions are similar among species. Anthropogenic sulfate  $(SO_4^{2^-})$  is secondarily produced from sulfur dioxide  $(SO_2)$  in atmosphere. As the co-editor suggested,  $SO_2$  does not always share the emission sources with CO, because power generation sector has a great contribution to  $SO_2$  emission but not to CO. However, the spatial distribution of  $SO_2$  emissions in East Asia is similar to that of CO emissions (Koike et al., JGR, 2003; Kurokawa et al., ACP, 2013). For clarifying this fact, we should refer these previous studies where CO is used as a tracer to investigate the transport and transformation of sulfur compounds in East Asian region (Koike et al., JGR, 2003). We hence suggested by referring Koike et al. (2003) that  $SO_2$  and CO emissions have similar spatial distributions over China and that CO can be used as a tracer of sulfur compounds as used in the previous studies.

**Referee #1 Comment on Line 317-319: Your response does not actually address the point raised regarding how the slopes in Fig. 6b do not appear to be different enough to justify your conclusion that the controlling removal process is rainout.**

>The scatter plots of BC,  $SO_4^{2^-}$ , and CO were altered in association with the accumulated precipitation along the trajectory (APT) as seen in Fig 6a and 6b. To clarify this, we selected the data points for air masses significantly affected by precipitation (APT >15 mm) and applied the linear regression analyses to the selected data. Both the slopes for the selected data points

of BC-CO and  $SO_4^{2-}$ -CO correlations were significantly lower than the upper envelopes of the scatter plots, and were close to the lower envelope. The wet removal is a key to reduce the abundance of BC and  $SO_4^{2-}$  relative to CO. The washout process was reassessed in the main text (described in the above), and found to be not so important during the observation period. Indirectly, we found that only the rainout can account for the removal of the accumulation mode particles in this study.

Referee #1 Comment on Line 372 regarding the BC peak diameter from the SP2: You state that a difference in ~1 fg BC can be resolved within the SP2's uncertainties, but do not state these uncertainties. Your response is too vague and requires more specific quantitative details to adequately respond to the Referee.

>We included the technical descriptions of SP2 performance on the resolving power of incandescence intensity (proportional to mass per particle, mpp) in section 2.1. The values of the changes in mass per particle were quantitatively evaluated. 1 fg BC around 1 fg of BC can be resolved, however the resolution is dependent on the signal levels. Figure 7c which is newly added shows the systematic change in peak diameter or mpp as a function of the degree of BC loss. The observed change in mpp is as large as 2-2.5 fg, which is larger than the uncertainties.

Referee #1 Comment on Line 373: You have not addressed the main concern raised regarding the supposed evidence for the selective removal of large BC particles, and how this evidence is not clear in the presented size distributions.

>As discussed in the second paragraph of section 3.6 (3.5 in the original manuscript), turbulent mixing in the PBL leads partial experience of the in-cloud scavenging for aerosol particles suspended within the PBL. This indicates that a certain fraction of aerosol particles in the PBL does not experience. The degree of changes in the shape as well as peak diameter of size distributions can be reduced by this effect. The air mass aging leads to the redistribution of the particle size distribution through the coagulation. The aging process after the wet removal process as well as the mixing process can qualitatively account for why the evidence is not clear in the observed data sets. Quantitative assessments of these effects should be performed using a chemical transport model which can resolve the mixing state of aerosol particles (Matsui et al., JGR, 2013). We included these points in "Discussion" section in the revised manuscript.

Referee #2 comment on SP2 BC mass/number distributions: The Referee's question

regarding the need for more details on when BC core vs. shell diameters are being discussed has not been addressed.

>In the revised manuscript, (1) the shell and core diameter range, and (2) the effect of laser power fluctuation to the lower limit of measurable diameter of BC-containing particles are clarified in section 2.1.

- (1) The range of diameters of BC-core was from 150 to 350 nm. The maximum value of shell to core (S/C) ratio of these BC-containing particles analyzed was 4. Retrieved results in this study suggest that almost all particles had no such high S/C ratios (~2.5 at the highest for 200 nm core BC-containing particles).
- (2) We diagnosed the housekeeping data to check the stability of the detection efficiency of BC-containing particles. The variations of the laser power were within 3% throughout the observation period. It is found that this factor does not largely affect the lower size limit of detection.

Referee #2 comment on ACSM data: The focus only on the sulfate measurements is not appropriate. You need to expand the results presented to discuss the full set of measurements from the ACSM. Your response regarding this question did not satisfactorily answer the Referee's question. Discussing how SO2 and CO share the same emission sources (which as discussed above is not a great assumption) does not actually explain why you don't present ACSM measurements of other aerosol components such as organics, ammonium, or nitrate. If you are going to assume that sulfate is the only major secondary aerosol component mixed with the BC (line 195), you must justify this assumption based on the measurements available. I would be surprised if there were not significant contributions from other secondary species such as organic carbon. On Page 343 you discuss another study that did observe organic coatings on BC particles in East Asian outflow.

>As described the above, we included in the revisions the data analyses of chemical composition variations in non-BC materials measured by using the ACSM. The temporal variations of nitrate, ammonium, and OM are shown in Fig 5 of section 3.2, and the average relative abundance of them is discussed. Ion balance was investigated to consider the chemical form of inorganic ions. Sulfate and nitrate were almost fully neutralized by ammonium. We found that the major components of fine aerosols were ammonium sulfate and organic material. We made a new section (section 3.4 in the revised manuscript) of the discussion on the changes in the composition of aerosols in air masses depending upon their histories. The contribution

of ammonium sulfate increased with the cloud processing through the transport. The contribution of OM is significant in all cases, especially for air masses with no precipitation and no cloud impacts. Relative changes in compositions against the air mass histories were not so large and were around ~10%. The two components, ammonium sulfate and OM, are the most important contributors to the non-BC materials in fine aerosol masses. Detailed speciation of OM based on the mass spectral analyses in East Asian outflow is beyond the scope of this study and has been discussed in previous studies (e.g., Irei et al., EST, 2011). We therefore did not include this topic in the manuscript. We refer a previous paper to present the chemical and structural feature of BC-containing particles in East Asian outflow air masses (Takami et al., JGR, 2013) as a great example of the mixing state of BC at the observation site. The major components, ammonium sulfate and OM, revealed by the ACSM measurements in this study were same as their study.

**Ryan Sullivan ACP Co-Editor**

We again appreciate your efforts. The revisions will improve our manuscripts. We hope our paper contribute the significance of the Atmospheric Chemistry and Physics.

[revised manuscript text omitted]
 158159thickly coated (for example,  $D_{\rm S}/D_{\rm core}$  ratios of ~2.5 at highest at  $D_{\rm core}$  of 0.2 µm). We 160 also analyzed the microphysical parameters of rBC particles measured using the SP2 in the early summer of 2014 at Yokosuka (35.32°N, 139.65°E, Fig. 1), located near 161 162industrial sources beside along Tokyo Bay (Miyakawa et al., 2016). These data sets 163 were used as a reference for the BC-containing particles in air masses strongly affected 164by combustion sources.

Equivalent BC (EBC, Petzold et al., 2013) mass concentrations are continuously measured at Fukue Island using two instruments; a continuous soot-monitoring system (COSMOS; model 3130, Kanomax, Japan), and a multi-angle absorption photometer (MAAP; MAAP5012, Thermo Scientific, Inc., USA). The details of the air sampling and intercomparisons for EBC measurements at Fukue Island have been described elsewhere (Kanaya et al., 2013; 2016). In this study, mass concentrations of EBC
measured using the COSMOS were evaluated by comparison with those of
SP2-derived rBC. The intercomparison between SP2 and COSMOS will be briefly
discussed in the followingbelow.

174Figure 2 depicts the correlation between COSMOS-EBC and SP2-rBC hourly mass 175concentrations. The unmeasured fraction of the rBC mass was corrected by 176extrapolation of the lognormal fit for the measured mass size distributions, to the 177outsides of the lower and upper boundaries measurable  $D_{core}$  range (0.08-and-0.5  $\mu m_{\tau}$ 178respectively). 
[revised manuscript text omitted]

362chemical composition of fine aerosols during the observation period was listed in 363Table 1. Ammonium sulfate and OM were abundant components. Figure 5 also 364includes the temporal variations in the fractional residence time over the selected 365 region defined in section 2.4 (top panel). The CO concentrations were typically 366enhanced for the period with the higher contributions of CN and CS.-The positive correlation of SO42 and CO suggests that the secondary formation of SO42 through 367transport was significant during the observation period, and that SO42 contributed to 368369the coating of BC-containing particles. A previous study suggested that the majority of SO42- aerosols were formed in less than around 1.5 days after the air masses left the 370 Chinese continent (Sahu et al., 2009). Kanaya et al. (2016) showed that the typical 371372transport time of continental outflow air masses at Fukue Island was around 1-2 days 373in spring. The positive correlation of SO42- and CO suggests that the secondary formation of SO42- through transport was significant during the observation period, and 374that SO42- contributed to the coating of BC-containing particles. The structure and 375376composition of fine aerosols in East Asian outflow were analyzed by using a secondary ion mass spectrometer in a previous study (Takami et al., 2013). They suggest that 377SO42- and OM are constituents in the coating of almost all BC-containing particles. 378Hence we concluded that ammonium sulfate and OM contributed to the growth of 379BC-containing particles. The small variability of SO42-/CO ratios is consistent with 380381these facts. The period with the APT > 3 mm is highlighted by light blue in Figure 5 to show the impact of wet removal on the transport of BC and  $SO_4^{2-}$  aerosols. The 382maximum concentrations of  $\frac{BC}{SO_4^2}$ , aerosols and CO were observed on the morning 383384 of March 22 (Ep.1) under the influence of the anticyclone (corresponding to the trajectories colored red in Fig. 4a) when the APT values were almost zero. In 385

contrast,  $\frac{BC \text{ and } SO_4^2}{A \text{ aerosol}}$  concentrations did not increase with CO in the period from the evening of April 5 to the morning of April 6 (Ep.2) under the influence of the migratory cyclone (corresponding to the trajectories colored black in **Fig. 4a**), when the APT was greater than 10 mm.

390

3.4. Correlation of BC, SO42-, and CO-as an indicator of the removal of acrosols 391Figures 6a and 6b show scatter plots of CO with BC and SO42-, respectively. 392 Positive correlation of BC and SO42- with CO was clearly found in air masses with low 393 394APT values. The linear regression was performed to the data points with the APT higher than 15 mm for BC-CO and  $SO_4^{2-}$ -CO. Note that the linear regression slope 395for BC-CO was determined by forcing through the background concentrations of BC 396(0 µg m-3) and CO (120 ppb). The slopes of the fitted lines were 1.4 and 9.8 ng m-3 397ppb-1 for BC-CO and SO42--CO, respectively, were close to the lower envelopes of the 398399correlations. It is evident from these scatter plots that the correlations relative enhancements of BC/CO and SO42-/ to CO weare mainly affected by the APT. The 400cloud processes of acrosol 
[revised manuscript text omitted]

---

## Author Response (AR4)

**Comments by Reviewer #1**

**Review of acp-2016-570, Alteration of the size distributions and mixing states of black carbon through transport in the boundary layer in East Asia, by T. Miyakawa et al.**

**The revised manuscript takes into account most of the reviewers' comments on the first version and I recommend that it be published when the items below (most of which are minor) are addressed. There are several instances where the wording is awkward; for these I have made recommendations on alternative wording.**

We appreciate the reviewer's comments on the manuscript entitled "Alteration of the size distributions and mixing states of black carbon through transport in the boundary layer in East Asia".   As the reviewers suggested, we have modified the manuscript.

(Reviewer's comments in bold)

Major revision points

1.   Terminology

The washout and rainout were rephrased by below-cloud and in-cloud scavenging, respectively, as suggested.    (We used simply "wet removal" only in Abstract.)

2.   More analyses of total particle diameters of BC-containing particles.

We included not only $D_S/D_{core}$ ratio but also $D_S$ in the results and discussion.    Figure 7c modified includes the evolution of the peak diameter of number-$D_S$ distribution as a function of the degree of the removal of BC.    Removal of large BC-containing particles was clearer in the modified figure than in the previous one.    The discussions on the CCN activity of BC-containing particles were included in the revised manuscript.    The ACSM-derived chemical composition and physicochemical properties of fine mode aerosols were analyzed to estimate the critical supersaturation ($SS_C$) as a function of $D_S/D_{core}$ ratio in Figure 8.    The estimated $SS_C$ decreases as increases in $D_S/D_{core}$ ratio, which we can easily expect.    It is indicated that the relative abundance of BC-containing particles with higher $D_S/D_{core}$ ratio and lower $SS_C$ decreased through the in-cloud scavenging during the observation period.    The words "selective removal" in this manuscript were rephrased by simply "removal", as the BC-containing particles with $D_{core} < 0.1$ µm can be significantly

[Figure]

Figure 7c modified (The evolution of $D_S$ as a function of ΔBC/ΔCO ratios was added to the previous figure. The range of $D_{core}$ for the calculation of $D_S$ ranged from 0.15 to 0.35 µm)

[Figure]

Figure 8 modified (Median values for all distributions were plotted as vertical allows to more clearly illustrate the changes in the distributions.    The estimated $SS_C$ as a function of $D_S/D_{core}$ ratio was plotted.)

3. Chemical composition

We modified section 3.5 as the reviewer suggested.   The comparison between cases (2) and (3)/(4) is the most useful to illustrate the changes in chemical composition with the cloud processing.   This comparison suggests the slight increases in ammonium sulfate and slight decreases in ammonium nitrate, OM, and BC.   The cloud processing only slightly changed the chemical composition of fine aerosols.   We hence modified this section based on the interpretation above.   Needless to say, one the most important point is that the variations of chemical compositions were small.   Therefore, we did not modified this point.

4. The process to change the size distributions and mixing state

We described the coagulation as one of candidates for the process controlling the changes in the size and mixing states without any quantitative evidences.   As the reviewers suggested, the particle concentrations are a key to consider how this process is effective to change the microphysical properties of fine mode particles.   In this study, we observed the BC-containing particles mainly at a remote island in Japan.   We hence considered that coagulation can be expected to be minor, especially in air masses affected by wet removal.   In the revised manuscript, we weakened the expression of the statements on the role of coagulation in the aging.   For example, the sentence (Line 478-481) was modified into "The aging (e.g., coagulation) of aerosols particles through the transport (i.e., around ~1 day) after the wet removal events may also lead to the further modification of the shape of the particle size distributions and the mixing state distributions which have been affected by cloud processes.   This factor is actually expected to be minor because the particle concentrations are too low to have high coagulation coefficients to accelerate this effect."

**Comments.**

**Line 86: The word "control" seems a bit strong, as it could easily be argued that global- and regional-scale distributions are controlled by sources; perhaps "strongly affect" or "have a large influence upon"**

We have corrected as suggested.

**Line 113: define quantitatively what diameter range "fine mode" refers to**

We replaced "fine mode" by "$PM_{2.5}$".

**Line 140: how is Dcore different from MED? If they are the same, then this should be explicitly stated, or better yet, only use one term for this quantity.**

Same. We modified the expression of the BC diameter for BC-containing particles. We only used $D_{core}$ to represent the BC diameter.

**Line 146: The statement that the particles were not so thickly coated seems at odds with the statement that the ratio was as high as 4 (line 145) or even 2.5 (line 146) – these seem like rather large coatings. Figure 8 indeed shows that Ds/Dcore is typically ~1.4 or so, but the statement as given on line 146 is unconvincing.**

The $D_S/D_{core}$ of 4 is the upper limit of the calculation. We actually never found such high $D_S/D_{core}$ BC-containing particles. The $D_S/D_{core}$ of ~2.5 means around the maximum levels of the retrieved values. We modified the sentences into "The upper limit of the estimation of $D_S/D_{core}$ ratios is 4 in this study. Maximum levels of $D_S/D_{core}$ ratios retrieved were ~2.5 at $D_{core}$ of 0.2 µm.".

**Lines 163-164: It is not the uncertainty that is minor, but rather the contribution to the mass that was outside the measured range – a very different quantity.**

As the reviewer suggested, this is not uncertainty. We hence simply modified the sentence into "Note that the unmeasured fraction of rBC mass was minor (<5%) in this study.".

**Line 178: This sentence is redundant to one on line 167 that stated that the average discrepancy was "comparable to the uncertainty of the COSMOS", which is not the same as "within the uncertainty." Perhaps remove one of the statements.**

We modified the sentences as suggested.

**Line 370: It is not clear what is meant by "lower envelopes of correlations", especially as something that can be compared to a slope.**

**Line 377: It appears the authors meant to state ~10 mm rather than ~1 mm.**

This statement is true. We hence have not modified the sentence.

**Lines 379-380: last sentence can be removed with no loss of information**

We have corrected as suggested.

**Line 398: The fraction of BC seems to be the same with or without rain (2.4 vs 2.5%); this seems to require some discussion.**

**Line 402: The statement that cloud processes affected the relative abundance of ammonium sulfate is true, but misleading, as it seemed that the only effect was a reduction in the ammonium nitrate, why would result in an increase in the relative abundance of ammonium sulfate. To focus on clouds affecting ammonium sulfate seems to misrepresent what actually occurred. Thus this statement requires a bit more discussion.**

**Line 403: Rather than state that the concentrations of OM increased from average in case 2, it would be better to state that OM seemed to be removed during precipitation, which attributes a physical explanation to the observation. That is, unless the authors are arguing that OM is formed during transport under dry conditions (which seems to be the statement made on lines 403-404 without supporting evidence).**

We respond to the above three comments as follows.

Differences between cases (2) (i.e., w.o. precipitation and cloud impacts) and (3)/(4) (i.e., w. could impacts) are appropriate to clarify the differences in chemical compositions between with and without cloud processing. We hence added the sentences "The comparison between cases (3) or (4) and (2) is useful to elucidate the effect of the cloud processing." and "Ammonium sulfate contribution slightly increased with the in-cloud scavenging (based on the comparison between cases (2) and (3) or (4)), while the relative contributions of ammonium nitrate, OM, and BC slightly decreased.".

As the reviewer suggested, all components of fine aerosols were removed by the wet removal process (this feature has been discussed in Kanaya et al., 2016). We hence added the sentence "As all components of fine aerosols were removed through the in-cloud scavenging (Fig. 10 of Kanaya et al., 2016), it is expected that the relative abundance does not largely vary with the in-cloud scavenging." after the sentence "The relative ~ 10%". The secondary formation of OM at this site has been discussed in previous studies listed in the manuscript. We modified the last two sentences in this section into "Detailed mass spectral analyses of OM, secondary formation of OM, and cloud-phase formation of OM in East Asia are beyond the scope of this study, and they are not discussed in this study. The former two issues have been investigated by previous studies (e.g., Irei et al., 2014; Yoshino et al., 2016).".

**Line 421: I don't see where the APT values for the "outflow without BC loss" and "outflow with BC loss" are given. These criteria were selected based on delta_BC/delta-CO ratios rather than APT values.**

These data sets were classified by the values of ΔBC/ΔCO.    The average APT values for these two air masses are listed in Table 2.

**Line 427: The size distributions of BC in Figure 7 differed among all three graphs; what the authors mean is the shape of the size distributions different primarily at BC diameters less than 0.1 micrometer.**

Yes for the number size distributions.    Other aspects are to show the typical size distributions of BC-containing particles at the observation sites, and to show the changes in the size distributions as a function of degree of removal.

**Lines 428-430: This statement is presented without evidence; it may be true, but merely stating it as true because it is one explanation is not sufficient.**

The sentences were modified into "In outflow air masses, such small BC-containing particles would be scavenged by larger particles in the coagulation process during transport.    The below-cloud scavenging can also affect the BC-containing particles in the smaller size range (<0.1 µm) when the air masses were affected by the precipitation.".

**Line 479: A simple calculation would give a good estimate for the amount of coagulation experienced over 1 day, which I would think would be quite low.**

At this moment, we did not perform the calculation of this fraction.    However, the particle concentrations after the wet removal are too low to show the large changes in the size distributions only through the coagulation.    We modified the sentence into "The aging (e.g., coagulation) of aerosols particles through the transport (i.e., around ~1 day) after the wet removal events may also lead to the further modification of the shape of the particle size distributions and the mixing state distributions which have been affected by cloud processes, which is actually expected to be minor because the particle concentrations are too low to have high coagulation coefficients to accelerate this effect.".

**Line 711: It should be stated that the +/- 20% is from the 1-1 line.**

We have corrected as suggested.

**There were a few places where the meaning was not clear, or where sentences were awkward to read, probably because of language difficulties. Suggestions are presented for how these could be reworded.**

We appreciate your kindness for such proper suggestions to improve the readability of this paper.

**Line 49: "exhibits on hygroscopicity" – perhaps "exhibits increased hygroscopicity"**

We have corrected as suggested.

**Lines 77: "… of BC during the cloud droplets formation, in air masses…: - perhaps "… of BC in air masses …"**

We have corrected as suggested.

**Lines 84: "the cloud processes" – perhaps "through cloud processing"**

We have corrected as suggested.

**Line 92: "synthetically" – meaning not clear; perhaps omit this word**

We have corrected as suggested.

**Lines 93-94: sentence reads awkwardly; perhaps "This study determined that the transport efficiency of BC aerosol particles through the PBL was substantially reduced by wet removal."**

We have corrected as suggested.

**Line 162: "distributions, to the outsides of the measurable…" – perhaps "distributions outside the measurable"**

We have corrected as suggested.

**Line 272: "migrating anticyclone and cyclone" – not clear what is meant; do the authors mean that both occur together, or that one of each was observed, or that either can be typically observed?**

We have corrected to "The migrating anticyclone and cyclone have passed alternately over East Asia during this period, which is typically dominant in spring over East Asia (Asai et al., 1988).".

**Line 274: sentence reads awkwardly; perhaps omit.**

We have removed it as suggested.

**Line 320-321: perhaps "Possible uncertainties in this estimate result from inaccuracies in the parameterization of the washout rate."**

We have corrected as suggested.

**Lines 311-329: entire paragraph seemed repetitive and could have been stated in 3-4 sentences**

We actually tried to reduce the sentences in the section 3.3. We consider that this part is a key to represent which processes (below-cloud or in-cloud scavenging) are more important as the wet removal of BC mass during the observation period. We hence could not summarize this part in 3-4 sentences as we have to include the details on the estimation of the removal rate of below-cloud scavenging (this should be included as pointed out by the co-editor). We finally reduced the number of sentences in this section from 10 to 7.

**Comments by Reviewer #2 Dr. Gavin McMeeking**

**Rather than review the revised manuscript in full, I have focused on an evaluation of the strength of the responses to the comments raised by myself and the other reviewer, and as to whether they adequately address the comments. While many of the changes have improved the manuscript, there still remains areas where the reviewer comments have not been addressed, as detailed below.**

We appreciate the reviewer's comments on the manuscript entitled "Alteration of the size distributions and mixing states of black carbon through transport in the boundary layer in East Asia". As the reviewers suggested, we have modified the manuscript.
(Reviewer's comments in bold)

Major revision points
1. Terminology
The washout and rainout were rephrased by below-cloud and in-cloud scavenging, respectively, as suggested. (We used simply "wet removal" only in Abstract.)

2. More analyses of total particle diameters of BC-containing particles.
We included not only $D_S/D_{core}$ ratio but also $D_S$ in the results and discussion. Figure 7c modified includes the evolution of the peak diameter of number-$D_S$ distribution as a function of the degree of the removal of BC. Removal of large BC-containing particles was clearer in the modified figure than in the previous one. The discussions on the CCN activity of BC-containing particles were included in the revised manuscript. The ACSM-derived chemical composition and physicochemical properties of fine mode aerosols were analyzed to estimate the critical supersaturation ($SS_C$) as a function of $D_S/D_{core}$ ratio in Figure 8. The estimated $SS_C$ decreases as increases in $D_S/D_{core}$ ratio, which we can easily expect. It is indicated that the relative abundance of BC-containing particles with higher $D_S/D_{core}$ ratio and lower $SS_C$ decreased through the in-cloud scavenging during the observation period.

[Figure]

Figure 7c modified (The evolution of $D_S$ as a function of $\Delta BC/\Delta CO$ ratios was added to the previous figure. The range of $D_{core}$ for the calculation of $D_S$ ranged from 0.15 to 0.35 µm)

[Figure]

Figure 8 modified (Median values for all distributions were plotted as vertical allows to more clearly illustrate the changes in the distributions. The estimated $SS_C$ as a function of $D_S/D_{core}$ ratio was plotted.)

3. Chemical composition

We modified section 3.5 as the reviewer suggested.   The comparison between cases (2) and (3)/(4) is the most useful to illustrate the changes in chemical composition with the cloud processing.   This comparison suggests the slight increases in ammonium sulfate and slight decreases in ammonium nitrate, OM, and BC.   The cloud processing only slightly changed the chemical composition of fine aerosols.   We hence modified this section based on the interpretation above.   Needless to say, one the most important point is that the variations of chemical compositions were small.   Therefore, we did not modified this point.

4. The process to change the size distributions and mixing state

We described the coagulation as one of candidates for the process controlling the changes in the size and mixing states without any quantitative evidences.   As the reviewers suggested, the particle concentrations are a key to consider how this process is effective to change the microphysical properties of fine mode particles.   In this study, we observed the BC-containing particles mainly at a remote island in Japan.   We hence considered that coagulation can be expected to be minor, especially in air masses affected by wet removal.   In the revised manuscript, we weakened the expression of the statements on the role of coagulation in the aging.   For example, the sentence (Line 478-481) was modified into "The aging (e.g., coagulation) of aerosols particles through the transport (i.e., around ~1 day) after the wet removal events may also lead to the further modification of the shape of the particle size distributions and the mixing state distributions which have been affected by cloud processes.   This factor is actually expected to be minor because the particle concentrations are too low to have high coagulation coefficients to accelerate this effect."

**Responses to Reviewer #1**

**Response to Line 56 comment: The changes/additions to Section 3.2 are good, but I am confused by the second-to-last statement in the response: "The rainout process is a major process to reduce the loss of aerosols in wet removal". Is this just a typo, since the calculations show only a minor estimated contribution? I think the terms washout and rain out should be avoided, and instead use "in-cloud" and "below-cloud" scavenging to describe the different physical processes.**

We modified the representations as suggested.

**Response to Line 152-154: Also a useful addition, however change "the discrepancy can be partly attributed to …" to "the discrepancy may be partly attributed to", since it has not been established whether there is a difference in the SP2 response to BC in remote air and FS.**

We have corrected as suggested.

**Response to reviewer comment on line 317-319:**

**The final line of text in Section 3.4 states "the changes in SO4/CO correlation were largely controlled by the rainout process and weakly influenced by aqueous-phase formation during transport." The argument in the response to the reviewer is that the aim is to determine the impact of the cloud process on aqueous-phase formation of SO4. The difference in slopes in this case is also small, and neither the response nor the revised manuscript addresses the main point of the reviewer comment that questions the significance of the different relationships in the data. A stronger response, and argument in the revised text, would provide uncertainties in the regression coefficients and discussion of the significance of the differences in the relationships. The underlying reasons given by the authors (wet removal and in-cloud formation) are certainly plausible, but not proven based on the data shown here. Additional minor point, but the range shown in the figure 6a is different from that stated in the revised text. The range shown in the plot includes the upper/lower limits associated with each of the Kanaya ranges...it may be better to give the same range in both (mean values?), whichever is most appropriate for the comparison.**

We evaluated the significance of differences in the slope of $SO_4^{2-}$/CO correlation between with and without cloud impacts using the analysis of covariance to investigate whether linear regression slopes for two data sets are statistically different.   We found that the difference was significant. Figure 6a was modified.   The ranges of the emission ratios of BC to CO (ER) shown in Kanaya et al. (2016) are actually large (This is because the uncertainty in their estimation is large.).   We should consider this point when we compare those with the observed $\Delta BC/\Delta CO$ ratios.   We hence did not remove the 1 σ range of the emission ratios (shaded region in the figure), however we added the representative values of ER for central north- and central south-china as the lines in the modified figure (see what the modified figure actually looks like in the following).

[Figure]

Modified figure 6 (a)

(Representative values of the emission ratios of BC to CO for central north- and central south-China shown in Kanaya et al. (2016) were plotted on the modified figure.)

**Response to reviewer comment on line 343:**
**The inclusion of "absolute" size distributions does not directly address the main point of the reviewer comment, that there is little evidence of preferential loss of larger BC particles relative to the loss of smaller particles. The new Figure 7c is more helpful, showing a trend, though quite noisy, in the mode BC core diameter. A stronger response would also note that the more important parameter to examine here would be total particle diameter, not just that of the BC core. Just because the BC core is small does not mean that the total particle diameter, including a coating, is also small, and therefore may be as easily scavenged as a bare or weakly coated larger BC core.**

We further analyzed the SP2 data sets to respond the reviewer's comment, as already described earlier (see "Major revision points"). The hourly peak values of $D_S$ were analyzed by fitting a lognormal function to BC number size distributions.    The evolution of peak $D$s as a function of $\Delta BC/\Delta CO$ was also plotted in Fig 7 (c).    A decreasing trend of $D_S$ with the removal of BC, which is similar to $D_{core}$, gave us an additional insight into the removal of BC-containing particles as suggested by the reviewers.    As the larger particles have a higher CCN activity, the observed decreasing trend in $D_S$ is consistent with our proposal, "selective removal of large BC-containing particles", which we have made in the previous manuscript.    However, below-cloud scavenging can also affect such smaller BC-containing particle concentrations.    We hence rephrased "selective removal" by simply "removal".

**Response to reviewer comment on line 345:**
**The response to the reviewer is weak. It does not address the main point of the comment, that concentrations are too low for coagulation to be an important process in removal of small BC particles. Given the manuscript focuses on changes in BC size distributions it seems such a fundamental topic should be discussed, even briefly, in the manuscript.**

As the reviewers suggested, the coagulation process cannot solely affect the observed changes. However, the size distributions of BC in air masses near sources can be significantly affected by the coagulation process.    We modified the original sentence, to weaken the expression, into "In outflow air masses, such small BC-containing particles would be scavenged by larger particles in the coagulation process during transport." to weaken the expression.    The modified manuscript include two factors, coagulation and below-cloud scavenging, as factors to affect the concentrations of sub-0.1 µm BC-containing particles.

**Response to reviewer comment to line 372:**
**I'm not clear which uncertainty in section 2.1 is being referred to in the response to this comment, but while I would agree that the SP2 can resolve quite small differences in rBC mass (assuming constant material properties), I think the uncertainty the reviewer is talking about here is the statistical uncertainty associated with the spread in the data, and whether there is a significant difference in fg/particle for the two conditions. Note that the comparisons of log-normal fit MMDs in Table 2 is less reliable because it includes an assumed density, which might not be constant for the two cases.**

We interpreted the uncertainty suggested by the reviewer #1 is related to the SP2 performance.    In order to further illustrate the differences, we have tested the statistical significance.    We found that the observed differences are statistically significant ($p < 0.01$).    The descriptions on this point were added to support the significance of the changes.    "The changes in the peak $D_{core}$ and $D_S$ from the highest to lowest bins of $\Delta BC/\Delta CO$ ratios were 0.02 µm (2-2.5 fg) and 0.05 µm, respectively, which are statistically significant (p < 0.01)."

**Response to reviewer comment to line 373:**
**The response to this comment somewhat undercuts the response to previous comments and the usefulness of Figure 7c. If size distributions change again during subsequent aging following the wet removal process, then is the apparent decrease in BC core size shown in Figure 7c meaningful or simply a random example where postwet removal aging processes happen to give a somewhat smaller average BC core size? Could a slightly different aging process following wet removal lead to a larger average size? On this basis I think any conclusions drawn regarding size dependent loss of BC should be removed or at minimum highly qualified noting the confounding factors the authors have pointed out in several of their responses.**

The point we included in the previous responses was that the shape of the size distributions can be modified through aging.    We interpreted the relatively small changes in BC microphysical parameters as the result of the mixing process in the PBL (i.e., mixing of BC-containing particles between in the cloud and w.o. cloud processing in the PBL).    We modified this point in the manuscript to prevent misunderstanding our interpretation and to show that is might be possible but minor, as follows.    "The aging (e.g., coagulation) of aerosols particles through the transport (i.e., around ~1 day) after the wet removal events may also lead to the further modification of the shape of the particle size distributions and the mixing state distributions which have been affected by cloud processes, which is actually expected to be minor because the particle concentrations are too low to have high coagulation coefficients to accelerate this effect."

**Responses to my comments:**
**First general comment (BC removal processes):**
**The additional discussion of BC removal processes is good, but suggest changing "Their" to "previous" in line 56 of the revised manuscript, and giving a very brief summary of the Kanaya et al. (2016) dry deposition results in section 3.2. For example, "The dry deposition in this region has already been evaluated by Kanaya et al. (2016), who found minimal decrease in BC/CO ratios for air masses unaffected by wet removal but with different transport times." The addition of a quantitative examination of below cloud scavenging is good.**

We have corrected as suggested.

**Second general comment (BC core versus shell; SP2 operating parameters):**

**The inclusion of SP2 operating conditions during the study is a good addition, however I do not think the response really addresses my point about the physical meaning and impacts of the BC core size versus the diameter of the mixed particle (core + shell). The BC core diameter is not the relevant diameter for CCN activation or other in-cloud scavenging processes, unless all particles are uncoated. I feel the manuscript should more clearly address this point, as well as the implications for some of the observations. For example, if most of the particles detectable by the SP2 are coated to the point where they roughly interact and/or activate in/as cloud droplets in a similar fashion then we would not expect a strong size dependence of removal. A more thorough and quantitative treatment of the interactions of BC particles mixed to varying degrees with other material with clouds would greatly strengthen the manuscript. While a full-blown microphysical modeling study would probably be beyond the scope of the investigation, some theoretical work treating particles as a simple core-shell morphology mixed with sulfate and organic aerosol and applying this to Kohler theory could be a great addition and strengthen the science presented.**

As already described earlier (see "Major revision points"), we added the evolution of the total diameter of BC-containing particles as a function of the degree of the removal of BC (modified Fig 7(c)).   This figure illustrates the removal of large BC-containing particles through the in-cloud process.   Furthermore, we included the estimation of critical supersaturation ($SS_C$) of BC-containing particles as its CCN activity in "Discussion".   As an example, we added the $SS_C$ as a function of $D_S/D_{core}$ ratios of BC-containing particles with $D_{core}$ of 0.2 µm on Figure 8.   The changes in the distributions of $D_S/D_{core}$ ratio can be easily connected with the CCN activity, even though the estimated $SS_C$ was not experimentally evaluated.   The modified figures 7 (c) and 8 support one of the major outcomes in this study, namely, size and mixing state distributions of BC-containing particles in the PBL were affected by the in-cloud scavenging process.

[revised manuscript text omitted]